# Highly multiplexed 3D profiling of cell states and immune niches in human tumors

Clarence Yapp [1,2,8], Ajit J. Nirmal [1,2,3,8], Felix Zhou [4], Alex Y. H. Wong [1,2], Juliann B. Tefft [1,2], Yi Daniel Lu [1,2], Zhiguo Shang [4], Zoltan Maliga [1], Paula Montero Llopis [5], George F. Murphy [6], Christine G. Lian[6], Gaudenz Danuser [4], Sandro Santagata [1,2,6,7] & Peter K. Sorger [1,2,7] ✉

Diseases such as cancer involve alterations in cell proportions, states and interactions, as well as complex changes in tissue morphology and architecture. Histopathological diagnosis of disease and most multiplexed spatial profiling relies on inspecting thin (4–5 μm) specimens. Here we describe a high-plex cyclic immunofluorescence method for three-dimensional tissue imaging and use it to show that few, if any, cells are intact in conventional thin tissue sections, reducing the accuracy of cell phenotyping and interaction analysis. However, three-dimensional cyclic immunofluorescence of sections eightfold to tenfold thicker enables accurate morphological assessment of diverse protein markers in intact tumor, immune and stromal cells. Moreover, the high resolution of this confocal approach generates images of cells in a preserved tissue environment at a level of detail previously limited to cell culture. Precise imaging of cell membranes also makes it possible to detect and map cell–cell contacts and juxtracrine signaling complexes in immune cell niches.

Rigorous assessment of cell morphology in research settings enables detailed analysis of processes such as organelle dynamics, cell migration and intracellular trafficking[1] and also plays a central role in the histopathological diagnosis of disease[2]. Rapid innovations in optical microscopy have enabled ever more precise three-dimensional (3D) characterization of cultured cells and model organisms[3]. Multiplexed tissue imaging ('spatial proteomics')[4] extends both histopathology and tissue biology by enabling the measurement of dozens of molecular markers in a preserved tissue environment. However, with a few noteworthy exceptions[5–7], most contemporary spatial proteomics is performed using widefield two-dimensional (2D) imaging methods at a resolution (commonly 0.6–2.0 μm laterally) that obscures fine morphological and intracellular details. The current emphasis on assay plex[8] and rapid data acquisition has merit, but the precise distribution of proteins within and outside of cells represents an invaluable

source of information about cell types and states[9,10]. Opportunities therefore exist to marry high-resolution 3D microscopy with spatial proteomics, particularly of the formaldehyde fixed paraffin-embedded (FFPE) specimens universally used for human diagnosis[11] and analysis of murine models[12].

In this study, we extend the public domain cyclic immunofluorescence (CyCIF)[7] method into 3D by using optical sectioning and 3D image analysis. Because this requires high numerical aperture (NA) objectives and confocal microscopy, this type of imaging also has high spatial resolution. In CyCIF and similar methods, high-plex images are generated by repeated rounds of four to six plex antibody staining, imaging, fluorophore inactivation (or antibody stripping) and then incubation with another set of antibodies. Almost all immunohistochemistry and histopathological analysis of hematoxylin and eosin (H&E) stained specimens is performed on ~5-μm-thick sections[13]

[1]Laboratory of Systems Pharmacology, Harvard Medical School, Boston, MA, USA. [2]Ludwig Centre at Harvard, Harvard Medical School, Boston, MA, USA. [3]Department of Dermatology, Brigham and Women's Hospital, Harvard Medical School, Boston, MA, USA. [4]Lyda Hill Department of Bioinformatics, UT Southwestern Medical Center, Dallas, TX, USA. [5]Microscopy Resources on the North Quad, Harvard Medical School, Boston, MA, USA. [6]Department of Pathology, Brigham and Women's Hospital, Harvard Medical School, Boston, MA, USA. [7]Department of Systems Biology, Harvard Medical School, Boston, MA, USA. [8]These authors contributed equally: Clarence Yapp, Ajit J. Nirmal. ✉e-mail: peter_sorger@hms.harvard.edu

because this minimizes interference from out of focus light[14]; existing spatial proteomics methods use similarly thin tissue sections. However, by performing 3D CyCIF on specimens cut at different thicknesses, we found that that nearly all cells (and most nuclei) are incomplete in 5-µm tissue sections, resulting in inaccurate phenotyping and obscuring many cell–cell contacts. Specimens 30–50 µm thick, which can be prepared using conventional sectioning techniques, were found to contain up to two layers of intact cells in which mitochondria, peroxisomes, secretory granules and juxtracrine cell–cell interactions could easily be resolved. Therefore, 3D thick-section CyCIF is ideal for studying cells, their constituents and their local communities in a preserved tissue environment.

## Results

3D CyCIF was used to image five tissue types spanning normal, cancerous and precancerous histologies. Each specimen was subjected to 8–18 rounds of cyclic imaging using a Zeiss LSM980 laser scanning confocal microscope, resulting in 20–54-plex images with 140 nm × 140 nm × 280 nm voxels (200–500 voxels per cell). Extracellular matrix (collagen) was imaged with second harmonic generation (SHG) by fluorescence lifetime imaging microscopy. The datasets averaged ~500 GB mm$^{-2}$ of tissue (the $z$-projections of each tissue can be viewed in Supplementary Figs. 1–11 or at full-resolution online via Minerva[15] and see Supplementary Table 1 for links and metadata and Supplementary Table 10 for protein nomenclature).

To explore tissue thickness as a variable, 5–50-µm-thick sections were cut from FFPE blocks, mounted on glass slides and subjected to dewaxing and antigen retrieval[16]. Alternatively, to enable multiplexed imaging of friable sections, they were mounted on coverslips, held in place with an 'adhesive' coating (Matrigel) or black polyethylene micro-meshes and then placed in 3D-printed carriers made from acrylonitrile butadiene styrene (Methods and Extended Data Fig. 1a–c). Reconstruction of confocal image stacks was performed using Imaris (RRID: SCR_007370) software followed by visualization of primary data slices and 3D surface renderings (Supplementary Note). The 3D segmentation was used to identify individual cells, generate uniform manifold approximation and projection embeddings and distinguish among major immune and tumor cell types[17] (Extended Data Fig. 1d).

### Standard 5 µm histological sections contain few intact cells and nuclei

Dehydration is a component of the paraffin embedding process known to change the volume of FFPE specimens[18]. We found that sections cut at 5 µm on a microtome expanded to ~9 µm following rehydration; similar proportional expansion was observed over a 5–35-µm-thicknesses range (slope ~1.5). Conversely, paraformaldehyde-fixed tissue sections in PBS cut with a vibratome shrank ~1.5-fold upon dehydration but expanded to its original thickness when rehydrated (Extended Data Fig. 1e–f). When tumor cell nuclear aspect ratios were quantified after rehydration, we

did not observe any systematic bias along the imaging (rehydration) axis (Extended Data Fig. 1g,h). Thus, rehydrated sections are likely to be representative of native tissue in three dimensions, at least on a local scale, but sections processed for H&E imaging are ~1.5-fold thinner. Thicknesses reported in this paper are the hydrated thickness as measured during image acquisition; when necessary 'cut@' is used to describe FFPE sectioning thickness (as in 'cut@5 µm').

Multiple layers of intact nuclei were visible in 30–40-µm tissue sections (Fig. 1a), but fewer than 5% of nuclei were intact in sections cut@5 µm (Extended Data Fig. 1i,j). For example, Fig. 1b,c shows a small cell community comprising a dendritic cell (D) and two T cells (T1 and T2, neighboring cells are not show), from a 54-plex CyCIF image of a 35-µm-thick section of invasive (vertical growth phase (VGP)) primary melanoma imaged using 194 optical slices spaced every 280 nm; the cells spanned ~25 µm along the optical axis ($Z$, upper image) and a similar distance in the plane of the specimen ($X,Y$; lower image). Immune cell phenotypes were assigned on the basis of patterns of expression of CD antigens and immune regulatory proteins in 3D images. However, when maximum intensity projections were generated from 9 µm virtual sections (Fig. 1d, labeled I–V) to mimic standard cut@5 µm 2D imaging and phenotyping repeated, many discrepancies were observed. In virtual section III, for example, T1 was incorrectly scored as positive for PD1 due to overlap with cell D along the $Z$ axis. In section I, true positive staining from D (CD11c and MX1) scored as background because the corresponding nucleus was largely absent from the section (segmentation relies on nuclei to locate cells[19]). Overall, 12% of true cytoplasmic signals (judged from 3D reconstructions) lacked a detectable nucleus in a 2D virtual section (Fig. 1e). The impact of such errors on cell type assignment varied with the marker protein: polarized proteins (that is, LAG3 and MX1) resulted in false negative calls 30–40% of the time whereas uniformly distributed proteins such as MART1 resulted in ~5% false negative calls (Fig. 1e). Thus, standard cut@5 µm 2D imaging fragments ~95% of cells as compared with ~20% in a 35 µm reference specimen, resulting in erroneous phenotypes in up to 40% cells, and also reduces the number of cell–cell interactions identified by proximity analysis (Extended Data Fig. 1k–m)[20].

Stereology aims to identify and mitigate biases arising from studying 3D objects in 2D[21,22], and our findings suggest a way to extend stereology to high-plex tissue imaging by generating simulated 2D datasets from ground-truth 3D data and using this information to compensate for 2D bias. However, the simulated 2D images in Fig. 1d do not correctly represent what would be captured by standard widefield slide scanners/microscopes. This arises because confocal microscopy, via a pinhole to reject out-of-focus light, collects photons from a point source approximately threefold to fourfold smaller in $X,Y$ and fivefold smaller in $Z$ than through a 0.5 NA objective lens fitted on a slide scanner. Thus, the 2D images in conventional spatial proteomics studies are less sharp and have lower signal-to-noise ratios than 3D CyCIF images (Extended Data Fig. 1n). Both image collection methods and differences

**Fig. 1 | Demonstrating the need for thick tissue sections using 3D CyCIF. a**, A surface rendering illustrates nuclear volumes in 9 µm (cut@5 µm, left) and 35 µm (cut@24 µm) tissue sections (right). Scale bars, 10 µm. **b**, Immunofluorescence images of six-marker subsets illustrating the microenvironment of the cellular community from the VGP highlighted in **c** to **d** (dotted lines). **c**, A 3D rendering of three selected cells from **b**. A comparison of the point spread functions (PSF) and optical planes (cyan; 280 nm spacing) for laser scanning confocal and widefield microscopy performed with a 40×/1.3 NA objective. Upper: $x,z$ (side) view. Lower: $y–x$ (top) view. **d**, Computed (virtual) 9 µm sections generated from 3D data were used to generate $x–z$ (center) and $x–y$ 2D projections (red boxes to left and right, labeled I–IV). **e**, Top: percentage misclassified cells in a virtual 9 µm section from entire VGP region in dataset 1 when stained with polarized (LAG3, MX1) and diffuse (CD103, MART1) markers, compared with ground-truth data from 35 µm sections ($n = 1,664$ for LAG3$^+$ cells, $n = 1,220$ for MX1$^+$ cells, $n = 85$ for LAG3&MX1$^+$ cells, $n = 980$ for CD103$^+$ cells, $n = 750$ for CD8$^+$ cells, $n = 403$ for CD4$^+$ cells, $n = 239$ for MART1$^+$ cells and $n = 72$ for SOX10$^+$ cells). Bottom: a quantification of the

percent of cells missing nuclei from virtual 9-µm versus 35-µm tissue sections from entire VGP region in dataset 1. A minimum size cut-off of 50 voxels was used to eliminate debris ($n = 1,198$ for cells missing nuclei in 9-µm section, $n = 362$ for cells missing nuclei in 35-µm section). **f**, A multimodal image integrating 3D CyCIF with SHG signal of collagen highlighting the MIS region. Maximum intensity projection of selected channels at lower magnification (left), with additional marker subsets for the indicated ROI (right). Scale bars, 100 µm and 10 µm. **g–i**, Field of views capturing the boundary of a VGP tumor, highlighting densely packed cells at low-resolution (**g**) and in high-resolution renderings of an individual cell (**h**) and illustrating the measurement of nuclear diameter, whole-cell diameter and distance from nucleus membrane to cell membrane (**i**). **j,k**, Examples of cells with extended membrane processes in melanoma: cluster of CD8$^+$ T cells in metastatic melanoma (**j**) and dendritic cell with filopodia extensions in metastatic melanoma (**k**). Two filopodia contacting a T cell and tumor cell, labeled with arrows or arrowheads, respectively. **l**, Langerhans cell in the MIS. Scale bars, 5 µm.

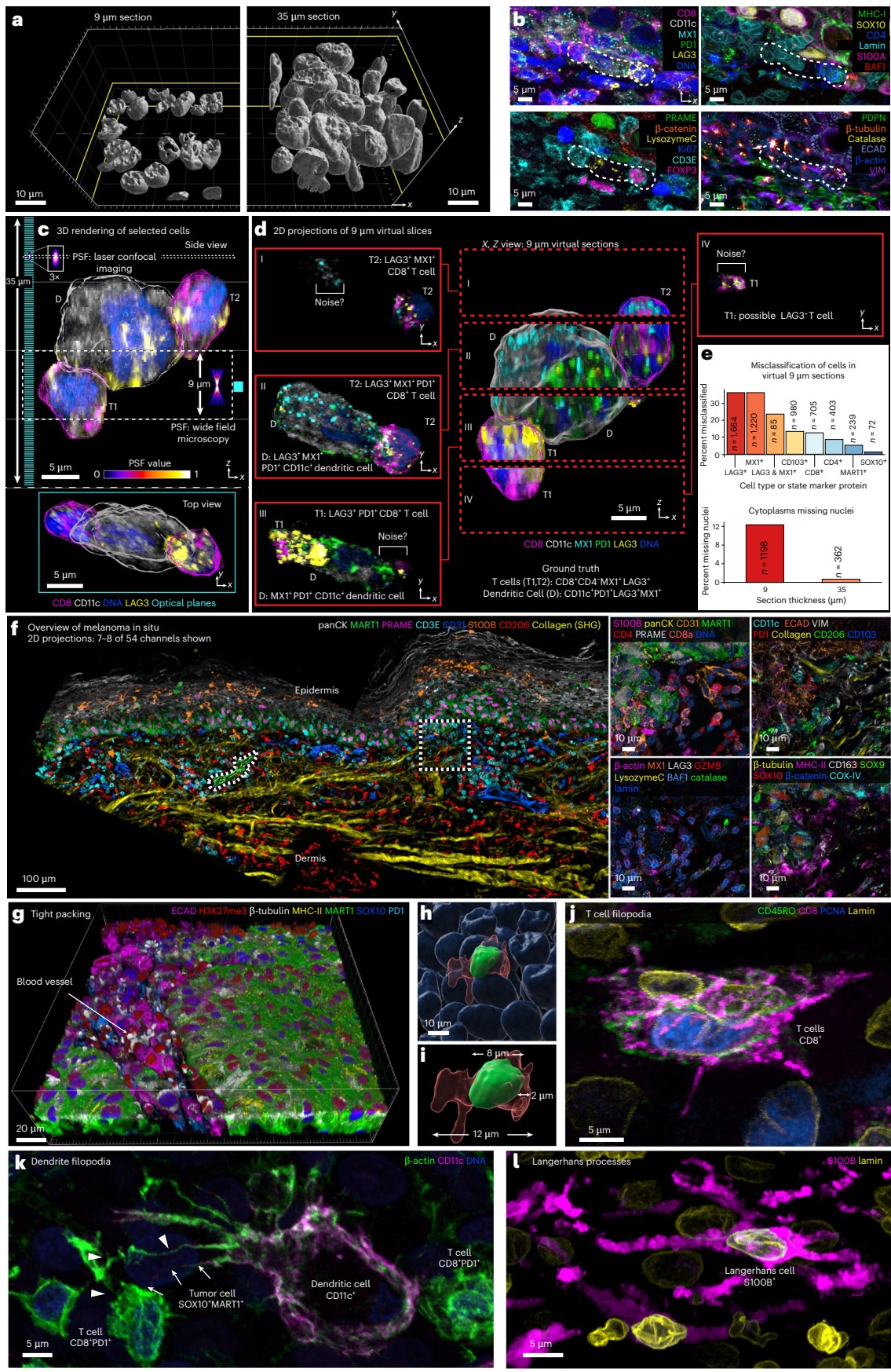

in cell completeness must be accounted for when using 3D high-plex images to better understand 2D images.

## Tumor microarchitecture

For simplicity, we focused biological analysis on data from melanoma specimens that included (1) a preinvasive cutaneous melanoma in situ (MIS) (Fig. 1f), (2) an invasive VGP primary melanoma from the same patient (Extended Data Fig. 2a,b) and (3) a metastatic melanoma to the skin from a different patient; data for other specimens are available in Supplementary Figs. 1–11. We found that cells in both melanomas and adjacent stroma were densely packed, except in areas where blood vessels or extracellular matrix (ECM) filled the voids (Fig. 1g and Supplementary Video 3). In the MIS, for example, nuclei averaged 5.0 μm in diameter (mean 7.2 ± 2.3 μm standard deviation for melanocytes and 4.9 ± 2.8 μm for immune cells). The cells averaged 13 ± 4.3 μm along the major axis and 6.1 ± 1.9 μm along the minor axis, consistent with recent and historical estimates[23]. Thus, the depth of the cytoplasmic compartment, scored as the distance from the plasma membrane to the nuclear lamina, was typically ~1–6 μm, and the membranes of neighboring cells were ~1 μm apart (Fig. 1h,i).

Some cells had highly extended cell bodies and cytoplasmic processes. For example, in Fig. 1j, three CD8+ T cells from metastatic melanoma extended multiple filipodia >5–10 μm from the cell body. A dendritic cell had 20–30 μm filopodia and membrane ruffles that contacted multiple CD8+PD1+ effector T cells (T_EFF); these specialized filopodia enable a switch from antigen sampling to antigen presentation during T cell priming[24] (Fig. 1k and Extended Data Fig. 2c,d). Langerhans cells, skin-resident macrophages[25], also had multiple membrane extensions, with branches extending 30–50 μm across multiple neighboring cell bodies (Fig. 1l). The 3D imaging was essential to identify these morphologies and distinguish changes in cell shape from changes in orientation: 2D views of VGP melanoma suggested concentration of round cells in the tumor center and elongated cells at the tumor margin. In 3D, this was seen to be a result of differences in orientation: cells in the center were more likely to be viewed end-on whereas those in the margin were rotated ~90° (Extended Data Fig. 2e–g). Theoretical studies of cell migration have demonstrated a strong dependency on the geometry of cell packing[26] and accurate 3D representations of tissues are likely to be useful in such studies. Conversely, tight cell packing, extended processes and overlap along the z axis also explain why accurate single-cell segmentation of 2D images and spatial transcriptomic profiles[27] is unlikely to be completely accurate even with optimized algorithms.

## Blood vessels and transendothelial migration

Thick-section 3D imaging made it possible to dissect components of the tissue microarchitecture not generally visible in 2D, such as a 100-μm-long portion of a dermal blood vessel in the MIS (Fig. 2a and Supplementary Video 2). In this vessel, ~10 vimentin and beta-catenin-positive endothelial cells formed a tube enclosing

erythrocytes and a neutrophil. At the distal end, a T_helper and dendritic cell were visible where the vessel appeared to branch. Most remarkable was a B cell (sphericity value ~0.35) flattened against the vessel wall, a morphology consistent with transendothelial migration of immune cells from vessels into tissues[28]. These features were not evident in virtual 9-μm-thick sections (Extended Data Fig. 3a). Elsewhere in the dermis, another B cell had its nucleus traversing a vessel wall while its cell body remained inside the vessel, and in metastatic melanoma, a T cell was visible suspended within a vessel (Extended Data Fig. 3b,c). In the MIS, we found that B cells were the cell type most likely to associate closely with collagen fibers in the dermis[29,30] (Fig. 2b,c) and were often (n = 11 of 14) stretched into irregular shapes. This was not a feature of all B cells; those found in the stroma and VGP were often round (Extended Data Fig. 3d,e). Functions have only recently been ascribed to B cells in the skin, and our images provide direct evidence of B cell recruitment from the vasculature into the dermis, followed by collagen binding, at densities consistent with other reports[31].

## Organelle and cell surface morphologies

High-plex imaging of whole immune and tumor cells in 3D revealed a wide variety of distinctive intracellular and plasma membrane structures (Fig. 2d), including lineage-associated differences in nuclear lamina (for example, a multilobed, hyper-segmented nucleus in neutrophils) (Fig. 2e), microtubule organizing centres (Fig. 2f), peroxisomes (based on catalase staining), secretory granules and/or ER (lysozyme C in neutrophils and granzyme B in T cells), DNA damage foci (γH2AX), mitochondria (COX IV) and biomolecular condensates (MX1[32]) (Fig. 2g–k and Extended Data Fig. 4a). Some features were found in many cell types and others only in selected lineages (for example, catalase foci in dendritic cells and γH2AX foci in keratinocytes and myeloid cells). Thus, intracellular and plasma membrane structures in archival human tissue sections can be characterized at a level of detail hitherto restricted to cultured cells and some model organisms.

Proteins used for immune cell subtyping exhibited a wide range of intracellular distributions. Some proteins were found throughout the plasma membrane, for example, the myeloid cell integrin CD11c, skin-homing T cell integrin CD103 and MHC-II receptor (in tumor and antigen presenting cells). Other proteins were found in discontinuous islands (CD4 and CD8 in T cells) or puncta (the immune checkpoint protein LAG3) (Fig. 2l–o). Some of these distributions have the potential to provide information on activity or cell state. For example, newly synthesized LAG3 localizes to endosomes but can rapidly translocate to the plasma membrane where it is activated by binding to MHC class II on the membranes of apposed cells[33]. Across specimens, we found 1–20 LAG3 puncta per cell, both inside cells and at the plasma membrane (Extended Data Fig. 4b). Granzyme B (GZMB) staining was diffuse and globular in CD4+ T cells and punctate in CD8+ T cells, consistent with localization to cytoplasmic granules. GZMB mediates the cytotoxic activity of T and natural killer cells and globular GZMB can be used to identify activated memory CD4+ T cells (Extended Data Fig. 4c,d and Supplementary Video 4)[34].

---

**Fig. 2 | Visualizing tumor microarchitecture and complex organelle and cell surface morphologies. a,** A surface rendering of a segment of an intact blood vessel (delineated by CD31+ endothelial cells) within the MIS region. The dashed lines demarcate cutting planes for cross-sectional views (lower left and upper right insets), which reveal internal components of the blood vessel including: CD11c+ dendritic cell, a CD4+ T cell, CD11b+CD11c− neutrophil, catalase+ red blood cell and CD20+ B cell undergoing transendothelial migration. Scale bar, 10 μm. **b,** CD20+ B cell (magenta) in the MIS with elongated morphology interacting closely with collagen fibers (yellow), shown as a maximum intensity projection. Scale bars, 5 μm. **c,** A Tukey box plot illustrating the distances between collagen fibers and different cell types in the MIS. Statistical significance was assessed using one sided unpaired Student's t-test. The center line shows the median, the box limits are the upper and lower quartiles and the whiskers are the minimum and maximum after removing outliers. Statistical significance observed between

CD8+ T cells and Mast cells (P = 0.0142) and macrophages and B cells (P = 0.0068). **d–f,** The maximum intensity projections of a neutrophil showing marker subsets for identifying specific organelles (**d**), a multilobed nucleus (**e**) and the cytoskeleton (**f**). **g,** A 3D rendering of the cell shown in **d–f**. **h–j,** A 3D rendering and maximum intensity projection inset (top right) for selected cells, including neutrophils in the MIS (**h** and **i**) and a T cell in metastatic melanoma (**j**). **k,** A 3D rendering of a neutrophil (CD11b+CD11c−) interacting with a MART1+SOX10+ tumor cell (magenta) in the MIS. **l–o,** T cell (CD8+CD103+PD1+TCF1+LAG3+GZMB+PCNA+), showing distribution of intracellular (**l** and **n**) and membranous (**m** and **o**) markers as maximum projections (**l** and **m**) and surface rendering (**n** and **o**). The selected channels of 54-plex 3D CyCIF images used for identifying cell types and organelles. All cells shown were from the dermis of the MIS for **d–i** (see Supplementary Fig. 13 for a full marker assignment). Tick marks on right side of panels **b**, **n** and **o** indicate 1 μm.

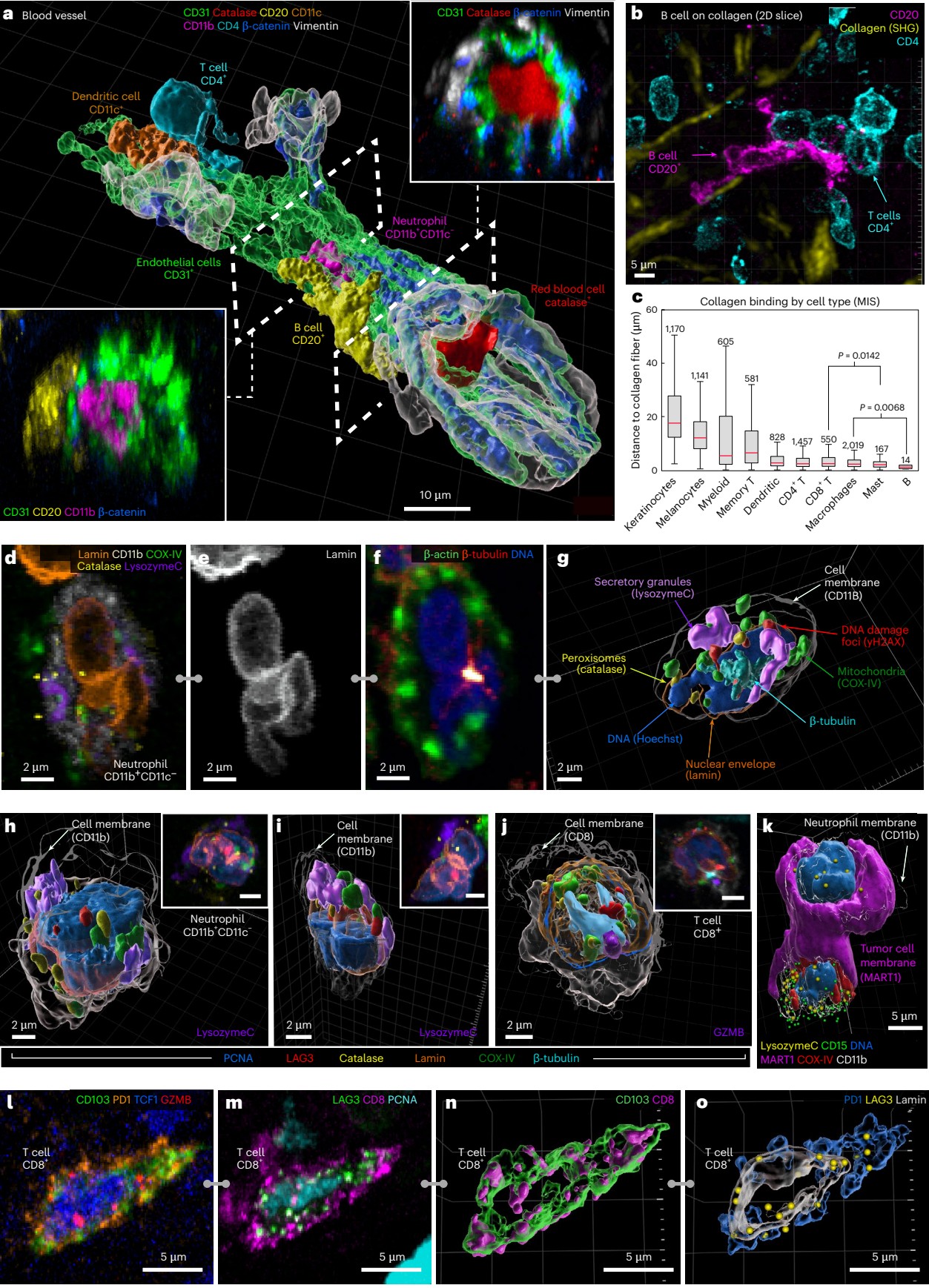

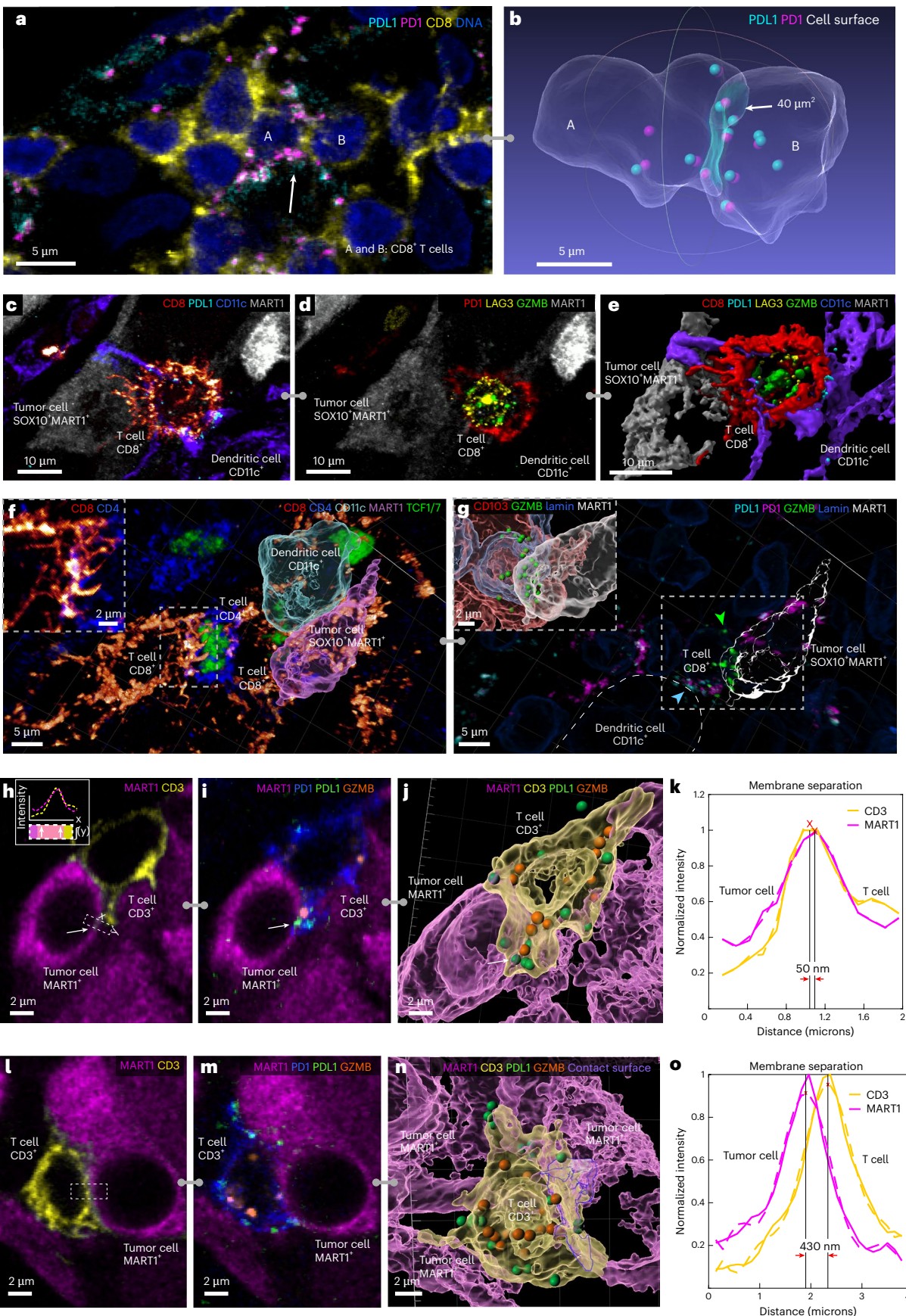

**Fig. 3 | Visualizing functional tumor-immune interactions in native tissue.**
**a**,**b**, Two CD8+ T cells (labelled A and B) expressing PD1 and PDL1 interacting within the VGP, shown as a maximum intensity projection (**a**). White arrow indicates contact surface between cells A and B with high colocalization of PD1 and PDL1. Transparent surface mesh showing contact area and colocalized PD1 and PDL1 as spheres (**b**). **c**, CD8+ T cell and dendritic cell interacting with a tumor cell with filopodia. **d**, The same cells as **c** with GZMB, PD1 and LAG3 are shown; the markers show that the T cell is activated and cytotoxic. **e**, A surface rendering of interactions in **c** and **d**, showing filopodia in greater detail. **f**, A multicellular interaction in metastatic melanoma. A dendritic cell interacting with a tumor cell (with surface rendering overlaid onto immunofluorescence) and a CD4+ T$_{helper}$ cell interacting with a CD8+ T cell through filopodia (inset and Supplementary Video 6). **g**, Same field of view in **f**, showing PD1 and PDL1 colocalization between CD8+ T cell and dendritic cell (blue arrow). In the CD8 cell, GZMB is polarized toward the tumor cell (green arrow and inset). The location of the inset is shown by the box with the dotted line. Scale bars are as indicated. **h**–**j**, The interaction of a MART1+ tumor cell with CD3+ T cell, shown as single optical planes (CD3, MART1 (**h**) and MART1, PD1, PDL1, GZMB (**i**) and as a surface rendering (**j**). The inset in **h** depicts computation of a membrane intensity profile by integrating the fluorescence intensity parallel to the cell membrane (y) and plotting it as a function of distance perpendicular to the membrane (x). **j**, The arrows indicate region of active PD1–PDL1 interaction. **k**, The membrane intensity profile of region indicated in **h**, spanning a point of tumor-immune cell contact demonstrating a type I interaction. **l**–**n**, The interaction of a MART1+ tumor cell with a CD3+ T cell, shown as single optical plane (CD3, MART1 (**l**) and MART1, PD1, PDL1, GZMB (**m**) and as a surface rendering (**n**) and highlighting PD1–PDL1 interactions and proximity to GZMB granules. **o**, The membrane intensity profile shown in **l** demonstrating a type II interaction. For the plots in **k** and **o**, the solid lines are from raw data, and the dashed lines are from a polynomial curve fitting. The red 'X's mark the maximum intensity along the intensity profile, which we defined as the midpoint of the cell membrane for each channel.

## Functional tumor-immune interactions

Changes in the distributions of cell surface proteins also revealed functional contacts. For example, the immune checkpoint receptor PD1 and its transmembrane ligand PDL1 varied from a relatively uniform distribution in the membrane to punctate (Fig. 3a,b, Extended Data Fig. 4e–g and Supplementary Note 1), with the punctate morphology most evident when PD1+ T cells were in contact with PDL1-expressing cells (primarily dendritic cells)[35]. In some cases, many PD1 and PDL1 puncta were visible across an extended domain of membrane–membrane apposition (for example, 13 foci over 40 μm²) (Fig. 3b), in an arrangement consistent with formation of juxtracrine signaling complexes. Several distinctive membrane structures were often visible in a single cell; for example, a PD1+CD8+ T cell contacting a tumor cell with filopodia while also binding PDL1 from a neighboring dendritic cell (Fig. 3c–e and Extended Data Fig. 4h–k). In a different multicellular community, filopodia from a CD8+ T cell contacted a CD4+ T helper cell (Fig. 3f, inset), which in turn contacted another CD8+ T cell that was in contact with a tumor cell and dendritic cell. GZMB in the CD8+ T cell was polarized toward the tumor cell (Fig. 3g, green arrowhead), even though PD1–PDL1 complexes had formed ~1.5 μm away along the T cell membrane (blue arrowhead).

Regulation of TCR signaling is highly localized, so we looked for evidence of T cells experiencing simultaneous and potentially divergent regulatory or functional interactions with more than one neighboring cell. For example, CD8+ T cells with evidence of cytotoxicity (for example, the presence GZMB granules) and cell membranes in close proximity to tumor cells (50 nm separation—consistent with synapse formation) (Fig. 3h–j) were observed to contain PD1–PDL1 complexes along the CD3-expressing membrane. The GZMB granules and PD1–PDL1 complexes in such T cells lay within a few microns of each other, and the cells were also in contact (more distantly) with myeloid cells and potentially regulatory CD4+ T cells. We found multiple examples of such communities, suggesting that a single T cell may be subjected to simultaneous negative and positive regulatory signals from interacting cells in a local niche (Fig. 3h–o).

## Tumor lineage plasticity

Mechanisms of melanoma initiation remain elusive[36], although epigenetic changes (for example, reduced 5-hydroxymethylcytosine (5hmc) levels)[37] have been implicated. The presence of melanoma precursor fields[38] and MIS is scored diagnostically by changes in the morphologies, numbers and positions of melanocytic cells in H&E and immunohistochemistry images[39]. In the MIS region, most melanocytic cells were located at the dermal–epidermal junction (DEJ), interacting with keratinocytes and retaining a dendritic morphology (dendrites are involved in transfer of ultraviolet-protective melanin) (Fig. 4a). By contrast, pagetoid melanocytic cells with an ameboid morphology but lacking dendrites were found at the top of the epidermis (Fig. 4b–h). Pagetoid spread by single and small groups of cells is a hallmark of oncogenic transformation[40]. Nonetheless, the MIS and underlying dermis were not highly proliferative, with only 1% of cells (n = 110) positive for the Ki67+ proliferation marker. Among these Ki67+ cells, 34% were T cells, while the remainder consisted of monocytes (28%) and endothelial cells (2.7%); only a single melanocytic cell was Ki67+ (Extended Data Fig. 5a). By contrast, in the invasive VGP domain from the same specimen, 11% of all cells were Ki67+ with melanoma tumor cells the most proliferative (45% Ki67+), followed by monocytes (44%). Thus, the MIS had the hallmarks of early oncogenic transformation and abnormal cell migration but with limited cell division[41]. The 3D imaging made it possible to unambiguously score combinations of nuclear and cell surface markers on MART1- or SOX10-positive tumor cells in the complex environment of the MIS (n = 875 cells); these markers included 5hmc, PRAME and MART1 (markers used clinically)[42], SOX9, SOX10 and MITF (three transcription factors associated with melanocyte differentiation). In these cells, the six markers were present in many possible combinations (Extended Data Fig. 5b), without evidence of significant spatial correlation (Fig. 4i–r and see Methods for details on per channel gating). The cells undergoing pagetoid spread also expressed many different combinations of lineage markers (Fig. 4c–h) and transcription factors (Fig. 4m,n), but expression

**Fig. 4 | Melanocyte morphologies, lineage marker expression and cellular interactions in the melanocytic intraepidermal compartment. a**, Surface-rendered melanocytes within the MIS, illustrating variations in dendritic (normal) and rounded (transformed) morphologies. Scale bars, 5 μm.
**b**, A representative field of view showcasing the transition of melanocyte morphology from dendritic-like at the DEJ to compact (bottom) and ultimately rounded during pagetoid spread within the epidermis (top). Scale bars, 10 μm.
**c**–**h**, Representative examples of pagetoid spread cells showing different expression levels of compact shaped cells expressing PRAME and MART1 (**c**) and 5hmc, sox10 and MART1 (**d**). **e**, Pagetoid shaped melanocyte expressing PRAME and MART1. **f**, Expression of SOX9 from panel **e**. **g**, Dendritic shaped melanocyte expressing PRAME and MART1. **h**, Expression of SOX10 from panel **g**. Scale bars, 5 μm. **i**–**n**, The images of the MIS. **i**,**k**,**m**, Segmentation masks colored by marker intensity and brightness representing mean expression levels of SOX9, SOX10 and PRAME. The masks in gray denote the positions of keratinocytes; the dashed circles denote IFN-rich domains. The markers are as indicated on each panel.
**j**, **l**, **n**: **j**,**l**,**n**, High-resolution immunofluorescence images of representative cells stained with the same markers per row. **o**–**p**, Images of the MIS as in **i**–**n**, but with colors indicating cells positive for PRAME (magenta) or cells dually positive for SOX9 and SOX10 (green). **q**,**r**, Images as in **i**–**p** but with magenta denoting cells containing nuclear actin rods. **s**, Maximum intensity projections of cells in the MIS showing gradations in SOX9 (red) and SOX10 (green) expression; the red arrows denote differentiated MART1+SOX10+ melanocytes; the green arrows denote MART1+SOX9+ melanocytic cells that have retained dendritic morphology; the yellow arrows denote MART1+ melanocytic cells that coexpress SOX9 and SOX10; the white arrows denote keratinocytes (which are also SOX9+). Scale bar, 30 μm. **t**, Similar data for tumor cells in the VGP regions. Scale bar, 30 μm. **u**, A magnified view of the cells from **s**. Scale bar, 10 μm.

of NGFR (CD271), a marker of melanoma initiating 'stem' cells[43], was not detected. These data imply that melanocytic cells with features of early malignant transformation are subject to frequent changes in cell state (phenotypic plasticity) rather than progressive evolution from a single transformed or progenitor (stem-like) cell, as proposed for advanced invasive melanoma. The degree of plasticity may be greater than suggested above, since we binarized states for easier enumeration even through images demonstrate the presence of intermediate

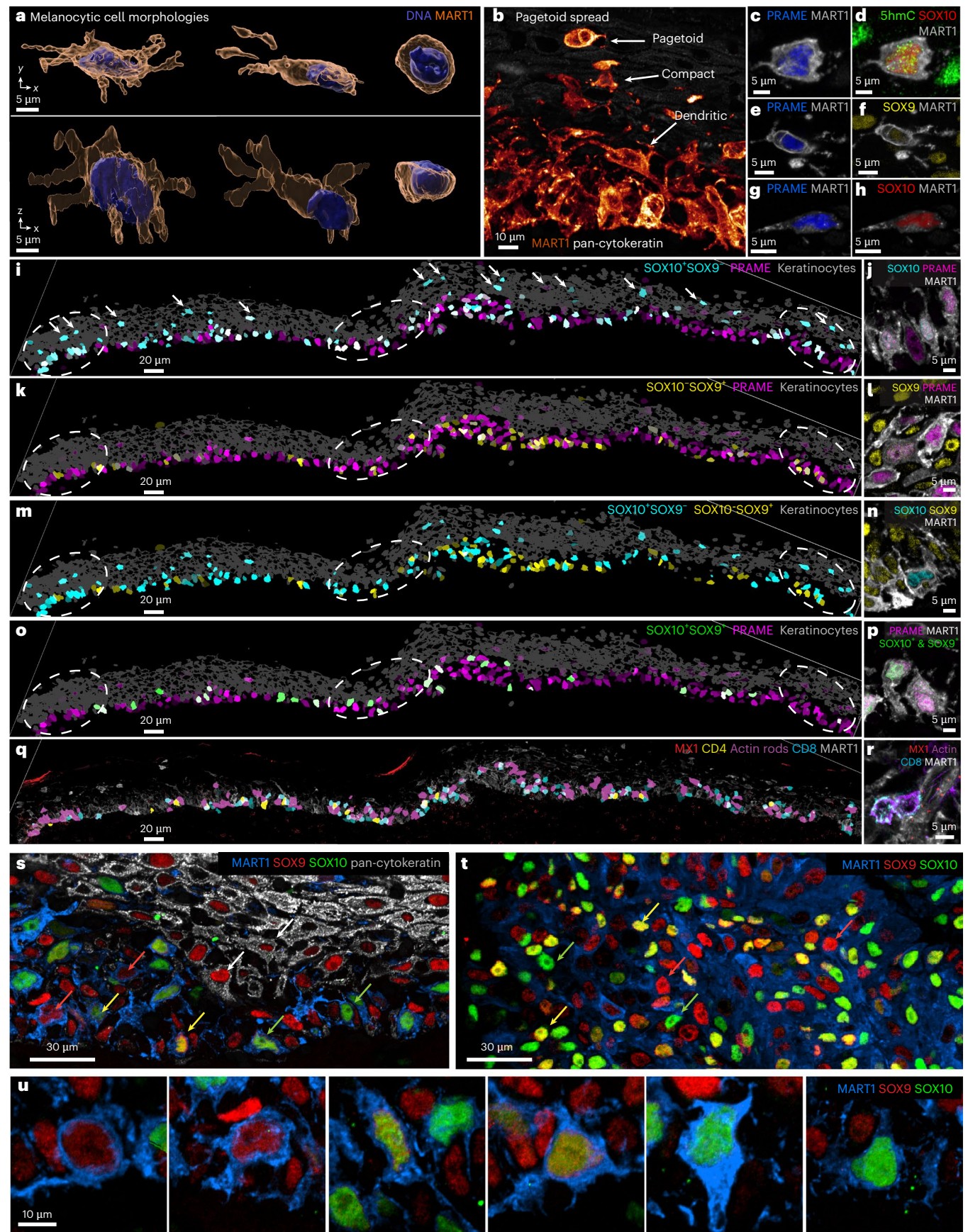

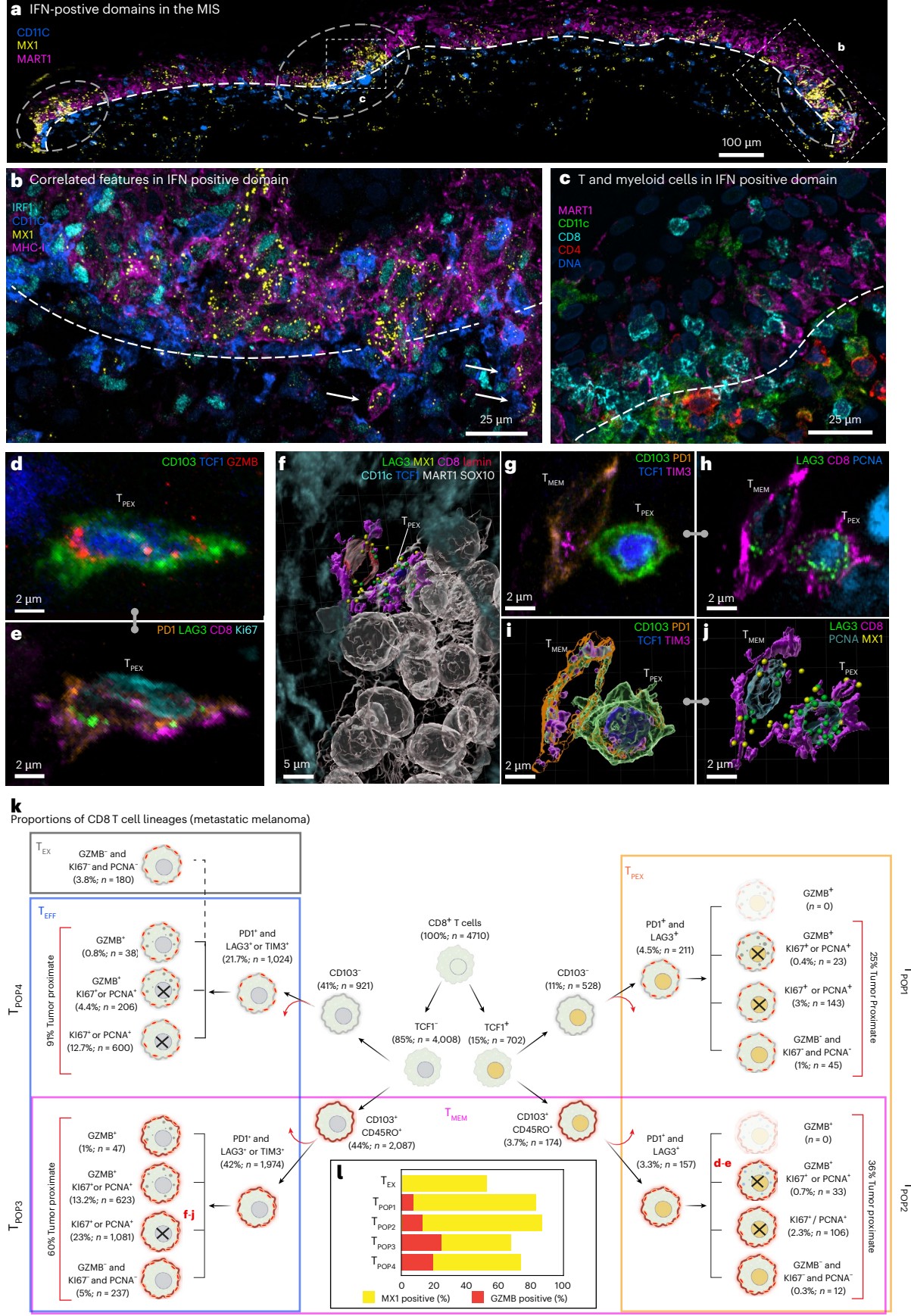

**a** IFN-postive domains in the MIS

CD11C
MX1
MART1

100 μm

**b** Correlated features in IFN positive domain

IRF1
CD11C
MX1
MHC I

25 μm

**c** T and myeloid cells in IFN positive domain

MART1
CD11c
CD8
CD4
DNA

25 μm

**d** CD103 TCF1 GZMB

T_PEX

2 μm

**e** PD1 LAG3 CD8 Ki67

T_PEX

2 μm

**f** LAG3 MX1 CD8 lamin
CD11c TCF1 MART1 SOX10

T_PEX

5 μm

**g** CD103 PD1
TCF1 TIM3

T_MEM

T_PEX

2 μm

**h** LAG3 CD8 PCNA

T_MEM

T_PEX

2 μm

**i** CD103 PD1
TCF1 TIM3

T_MEM

T_PEX

2 μm

**j** LAG3 CD8
PCNA MX1

T_MEM

T_PEX

2 μm

**k** Proportions of CD8 T cell lineages (metastatic melanoma)

**l**

MX1 positive (%)   GZMB positive (%)

**Fig. 5 | Spatial analysis of IFN-rich domains and distinct T cell lineages. a**, Three selected channels of 54-plex 3D CyCIF image of the MIS in dataset 2 (LSP13625). A maximum intensity projection of 116 planes is shown. IFN-rich domains are denoted by dashed circles. DEJ is denoted by a white dashed line. Scale bar, 100 μm. **b**, A magnified view of the inset from **a** showing same cell nuclear localization of IRF1, expression of MX1 and MHC-1 upregulation. DEJ is denoted by a white dashed line. The white arrowheads denote the invasion of melanocytic cells into the dermis. Scale bar, 25 μm. **c**, The enlarged inset from **a**, showing diversity of immune cells crossing the DEJ. DEJ is denoted by the white dashed lines. **d**,**e**, A maximum intensity projection of an activated $T_{PEX}$ cell, showing intracellular organelles such as GZMB (**d**) and membranous proteins such as LAG3 and PD1 (**e**). **f**, A 3D rendering of a $T_{MEM}$ cell interacting with a $T_{PEX}$ cell, which is in turn interacting with a cluster of metastatic melanoma tumor cells. Dendritic cells surround the neighborhood. **g–j**, The $T_{MEM}$ and $T_{PEX}$ cells shown in **f**, as a maximum projection (CD103, PD1, TCF1, TIM3 (**g**) and LAG3, CD8, PCNA (**g**) and 3D rendering (**i** and **j**). **k**, A hierarchical tree diagram showing proportions of CD8+ T sublineages the metastatic melanoma specimen. $T_{MEM}$ (magenta), $T_{PEX}$ (orange) and $T_{EFF}$ (blue) subtypes overlap, giving rise to four hybrid populations ($T_{POP1-4}$) as denoted by vertical labels (see 'Progenitor and $T_{EFF}$ subsets' for details). The red arrows denote additional cell subsets that are not shown on this tree. **l**, The percent of cells positive for GZMB (red) or MX1 (yellow) by population. See Supplementary Fig. 13 for a detailed diagram of which markers were used to define each cell type.

states (see Fig. 4s–u and Extended Data Fig. 5c for SOX9, red, and SOX10, green).

## Inflammatory neighborhoods

The MIS contained multiple spatially distinct domains of inflammatory signaling, which were ~50–100 μm in diameter and exhibited elevated levels or distributions of interferon (IFN) responsive proteins, such as IRF1, MX1 and MHC-I (Fig. 5a,b). IFN domain size was confirmed on a distant serial section (Extended Data Fig. 6). IRF1 translocates from the cytoplasm to the nucleus upon IFN exposure and MX1 and MHC-I are downstream response genes; MX1 forms distinctive biomolecular condensates in the cytoplasm (often multiple condensates in a single cell)[32] and MHC-I is found on the cell surface. Localized IFN-expressing niches have been previously described[44], although the distribution of IFN within the tumor microenvironment is a topic of active investigation[45]. Within these inflammatory domains, melanocytic cells had started to pass through the DEJ and were in contact with immune cells and the opposite was also true (Fig. 5b,c). Thus, our data provide direct evidence for restricted and recurrent spatial niches, defined by the simultaneous presence of an IFN response, melanocyte–immune cell contact and melanocytes crossing the DEJ (the first step in invasion). These IFN-positive spatial niches were coincident with the lineage switching described above but without detectable spatial correlation, despite evidence that IFN can induce melanoma dedifferentiation in cultured melanoma cells[46].

## Progenitor and $T_{EFF}$ subsets

The normal epidermis has an abundance of resident memory T cells ($T_{MEM}$) as a consequence of prior encounters with nontumor antigens (tissue-homing $T_{MEM}$ cells are characterized in our data by expression of the lineage markers CD45RO and CD103)[47]. The presence of tumor leads to additional T cell recruitment and activation. In-depth 3D immunoprofiling of metastatic melanoma using ten T cell lineage and state markers ($n$ = 4,710 CD8 and $n$ = 2,820 CD4 cells) revealed a diversity of populations and states (Fig. 5d–k and Extended Data Fig. 6d,e). Among these, progenitor exhausted T ($T_{PEX}$) cells[48–50] (15% of CD8 cells) (Fig. 5k, orange box) are of particular interest because they can be reactivated by immune checkpoint inhibitors, and their presence is associated with improved patient outcomes[51]. These cells are commonly defined as CD8+CD3+ T cells coexpressing the master transcriptional regulator T cell factor 1 (TCF1)[52] and checkpoint proteins (exhaustion markers) PD1 and LAG3. In our data, $T_{PEX}$ cells could be divided into two subpopulations based on expression of the CD45RO and CD103 resident memory markers ($T_{MEM}$, magenta box). Thus, $T_{PEX}$ and $T_{MEM}$ populations overlapped (giving rise to the hybrid populations $T_{POP1}$ and $T_{POP2}$) (Fig. 5k). $T_{MEM}$ cells also overlapped with $T_{EFF}$ (blue box; defined as CD8+ TCF1−PD1+ [LAG3 or TIM3]+ [Ki67, PCNA and/or GZMB]+) and gave rise to hybrid populations $T_{POP3}$ and $T_{POP4}$. Terminally exhausted cells ($T_{EX}$; gray box; defined as CD8+ PD1+ [LAG3 or TIM3]+ [Ki67−, PCNA− and GZMB−]) were distinct from the four hybrid populations. LAG3 puncta were observed in all hybrid populations with $T_{MEM}$ and CD103+ $T_{PEX}$ having the most (Extended Data Fig. 4b). Thus, in-depth phenotyping made possible by 3D imaging shows that $T_{PEX}$, $T_{MEM}$ and $T_{EFF}$ CD8+ T cells have overlapping transitional states, consistent with a growing body of single-cell RNA (scRNA) sequencing data[48] (Extended Data Fig. 6f).

Cytotoxic T cells were distinguished in our data by the presence of 1–20 GZMB puncta per cell (Extended Data Fig. 4c,d). Unexpectedly, GZMB+ T cells (5% of all CD8+ T cells) were found to be both positive or negative for TCF1, CD45RO and CD103 (Fig. 5l, red bars). TCF1− cytotoxic T cells were more likely to be GZMB+ and in contact with tumor cells than any other subtype, but visual review confirmed that all four populations included cells with GZMB polarized toward closely apposed tumor cells, implying active cell killing. In addition, 40–80% of $T_{PEX}$, $T_{MEM}$ and $T_{EFF}$ cells were PCNA or Ki67 positive, consistent with recent (or ingoing) cell proliferation[53,54]. Moreover, greater than 60% of all CD8+ T cells contained multiple MX1 puncta, indicating an active response

**Fig. 6 | Cell–cell interactions and multivalent immune cell niches. a**, A single image plane of a PD1+CD8+ T cell interacting with two MART1+ tumor cells. The white arrow denotes the juxtacrine PD1–PDL1 interaction. **b**, The membrane intensity profile perpendicular to axis of interaction of membranes from cells 1 and 3 in **a** and demonstrating a type I interaction. **c**,**d**, A dendritic cell interacting with a CD8+ T cell as a maximum projection (**c**) and membrane intensity profile (type I interaction) (**d**). The box indicates the region of line membrane intensity profile shown in **d**. **e**,**f**, Representations of a $T_{PEX}$ cell from the invasive margin as a maximum projection (**e**) and membrane intensity profile (type I interaction) (**f**), for the region marked by the white dashed box in **l**. **g**,**h**, A CD4+ $T_{helper}$ cell (magenta) interacting with a dendritic cell (yellow), as a maximum intensity projection (**g**). Membrane intensity profiles are representative of type II interactions (**h**). The selected fields of view and intensity profiles for selected membrane–membrane interactions in dataset 1 (LSP13626) are shown for **a–h**. Scale bars, 5 μm. **i–k**, Examples of type III interactions (lymphonets), involving CD4+CD8+ dendritic cells in the MIS. These networks are characterized by loosely packed cells with cell–cell interactions via relatively small membrane domains. **l**, The stroma in the vicinity of the VGP melanoma showing neighborhoods rich in CD20+ B cells, CD11C+ dendritic cells and CD3E+ T cells but without the clusters of proliferating Ki67+ cells that are characteristic of mature germinal centers. **m–o**, A total of 11 cells from the MIS lying in proximity to the DEJ, shown as primary data (**m**), 3D surface renderings (**n**) and as a schematic representation of three type I, four type II and two type III interactions inferred from membrane intensity profiles in single-plane images (**o**). Community in **m** and **n**, also shown in Supplementary Video 5. Scale bars, 10 μm. **p**, A cross-sectional slice of two CD8+ and two CD4+ T cells depicting the direct engagement of cell membranes (Supplementary Video 4). Scale bar, 3 μm. **q**,**r**, A membrane intensity profile depicting CD4 and CD8 average expression within the bottom boxed region of **p** (**q**) or right boxed region of **r** (**s**). **s–u**, A type II interaction between a MART1+ tumor cell and CD4+ T cells, as a representative image (**s**) and membrane intensity profiles for the interaction between cells 1–2 at region A (**t**) and B (**u**). Scale bars, 3 μm. For membrane intensity profiles, the solid lines are from raw data, dashed lines from polynomial curve fitting. The red 'X's mark the maximum intensity along membrane intensity profile and thus, the centroid of the cell membrane for each channel. See Supplementary Fig. 13 for a detailed diagram of which markers were used to define each cell type.

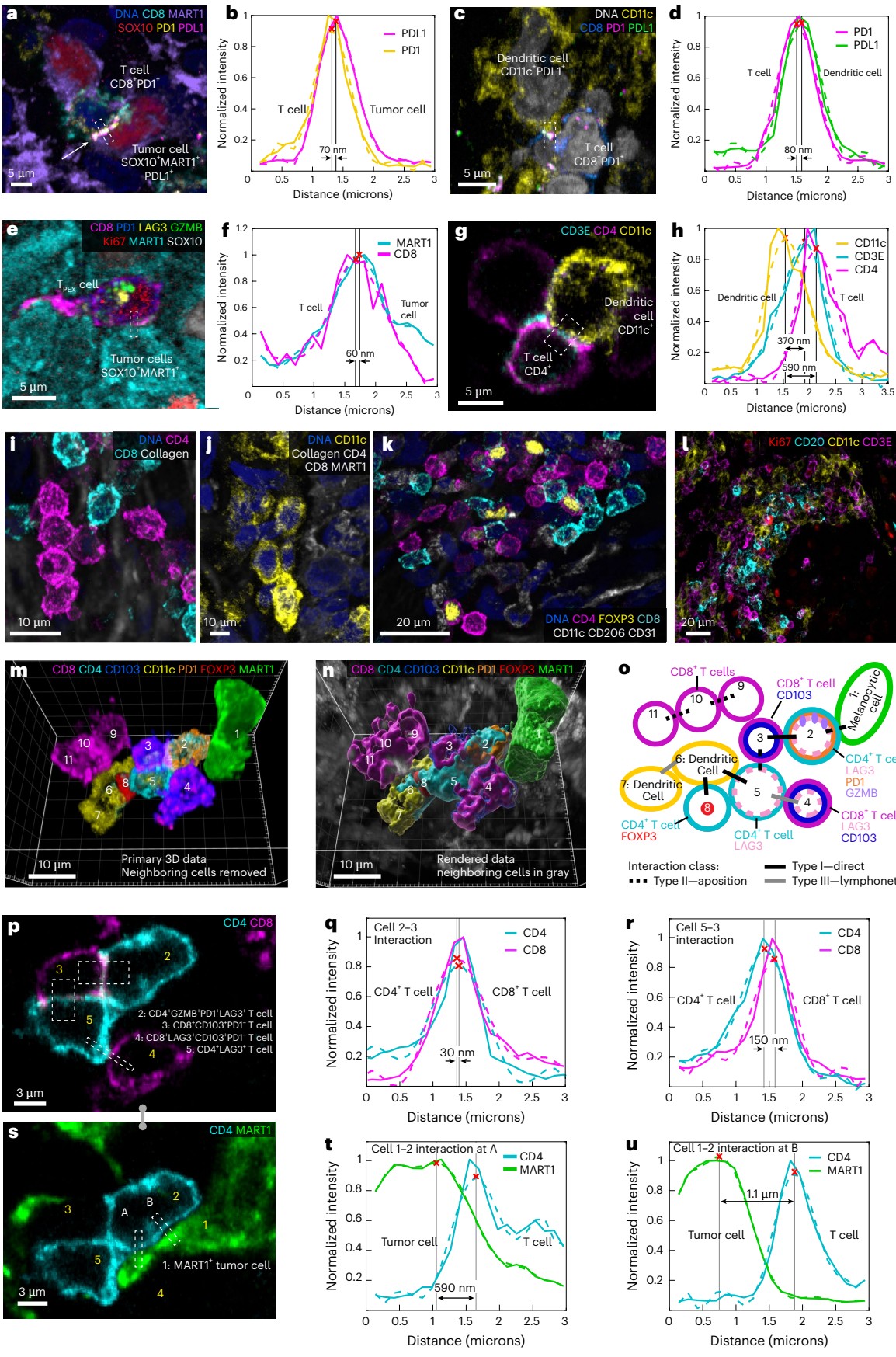

to IFN (Fig. 5l, yellow bars). These data are consistent with the known effects of IFN on T cell proliferation and suggest that this extends to all major T cell subtypes.

Spatial analysis showed that $T_{PEX}$ cells were significantly closer to $T_{MEM}$ and $T_{EFF}$ cells than to $T_{EX}$ or other $T_{PEX}$ cells and that $T_{MEM}$ cells were closer to tumor cells (Fig. 5k and Extended Data Fig. 6g–i). These data suggest evolution of $T_{PEX}$ cells from a TCF1+ to a TCF1− effector status in both memory ($T_{POP2}$ CD45RO+CD103+) and classical ($T_{POP1}$) populations (Extended Data Fig. 6j,k). Thus, the $T_{PEX}$ cells we detected are probably involved in (1) self-renewal, (2) formation of TCF1+ cytotoxic cells and (3) differentiation into classic $T_{EFF}$ cells.

### Membrane–membrane interactions

Existing approaches to proximity analysis use nuclear positions as an approximate means of identifying cell–cell interactions[35], but high-resolution imaging makes it possible to study membrane–membrane interactions directly. We observed three arrangements on the basis of the separation and extent of membrane–membrane proximity: (1) direct binding (type I interaction), (2) membrane apposition (type II) and (3) neighborhood clustering (type III). Direct binding involved pixel-level overlap in proteins from neighboring cells and, thus, colocalization at the resolution limit of the microscope. These type I interactions were most obvious in the case of CD8+PD1+ T cells interacting with tumor and myeloid cells; Fig. 6a shows this for a rare PDL1+ melanocytic cell. A membrane intensity profile spanning a cell–cell junction on apposed membranes followed by polynomial curve fitting, revealed a membrane-to-membrane spacing of ~70 nm (Fig. 6b) versus an average intermembrane spacing of ~1.5 µm among all cells. PDL1-expressing dendritic cells bound to CD8+PD1+ T cells had a similar membrane spacing (Fig. 6c,d), as did a T cell interacting with a PDL1 negative melanocytic cell in the VGP (Fig. 6e,f and Extended Data Fig. 7a). Given the resolution limits of optical microscopes, these type I spacings are consistent with integrin-stabilized immunological synapses formation, which EM shows to involve ~30 nm membrane separation[55].

Type II cell–cell interactions were characterized by neighboring cells with extensive membrane apposition but without evidence of pixel-level overlap in protein staining; in this case, a membrane–membrane spacing of 300–600 nm was typical (Fig. 6g,h). Type II interactions between CD4+ and CD8+ T cells were common across the MIS. In the conventional mode, antigen presenting cells (APCs) present antigens to both CD4+ and CD8+ T cells, with the CD4 helper cells enhancing the cytotoxicity of CD8 cells via cytokine production. Speculatively, type II interactions may facilitate paracrine signaling.

Type III cell–cell interactions involved an intermembrane spacing of ~500 nm involving a smaller area of the membrane (1–2 µm²); such interactions may correspond to the tightly packed (jammed) cell arrangement[56]. In some cases, 100 or more immune cells were observed to make type III interactions, generating lymphocyte networks ('lymphonets'). We observed lymphonets comprised primarily of CD4+ T cells or CD8+ T cells, dendritic cells and mixtures thereof (Fig. 6i–l and Extended Data Fig. 7b). 'Lymphonets' did not contain CD4+FOXP3+ regulatory T cells, which were most commonly involved in type I interactions (Fig. 6k), or tissue-resident macrophages, which were uniformly distributed across the dermis. Type III interactions among T, B and dendritic cells were also observed in VGP melanoma (Fig. 6l) and may represent nascent tertiary lymphoid structures (TLS), which play a role in responsiveness to immunotherapy[57].

Different classes of membrane–membrane interactions frequently co-occurred. Figure 6m,n shows a community of one melanocytic cell at the DEJ (cell 1) and ten immune cells (cells 2–11) (Fig. 6o and Supplementary Video 5) in which a CD4+GZMB+ $T_{MEM}$ (cell 2) formed a type I contact with a CD8+LAG3+CD103+PD1− $T_{MEM}$ (cell 3) (Fig. 6p) with an estimated membrane–membrane spacing of 30 nm over a 20–30 um² area (Fig. 6q). Cell 3 made an extended type II contact with a CD4+LAG3+ T cell (cell 5) (150 nm spacing Fig. 6r). Cell 4 and 5 engaged in a spatially

restricted type III interaction (640 nm spacing) (Extended Data Fig. 7c). Cell 2 (CD4+ T cell) also engaged in type II contact with a melanocytic cell (1; 690 nm spacing) (Fig. 6s,t). The two cells were proximate over a much larger area, but we judged the 1.1 µm spacing to be a consequence of tissue packing rather than interaction (Fig. 6u). Elsewhere in the network, type I interactions were observed between CD4+ T cell (cell 5) and a dendritic cell (cell 6) and cell 6 and a CD4+ $T_{REG}$ cell (cell 8); finally, a CD8+ network (cells 9–11) extended in an epidermal direction (Fig. 6o). Thus, a single immune or tumor cell can be in intimate contacts with multiple other cells of different types.

## Discussion

Our data show that 3D high-plex imaging data of tissues and tumors has the potential to substantially improve how we study the physical organization of tissues, assign single-cell phenotypes and infer regulatory interactions in multicellular communities. Visualizing the precise shapes of whole cells makes it possible to study cell–cell contact from the perspective of juxtaposed membrane–membrane contacts rather than mere proximity of cell (nuclear) centroids. High-resolution 3D imaging also overcomes errors in 2D image segmentation and corrects for misassignment of marker expression states due to protein polarization within cells or overlap of cells along the imaging axis.

We find that conventional 5-µm tissue sections contain few if any intact cells (or even nuclei) and that cell fragmentation substantially interferes with cell type assignment. However, by increasing section thickness only fourfold to fivefold (to a hydrated thickness of 30–40 µm), and performing optical sectioning, it is possible to overcome this problem and study organelles, condensates, cytoskeletal structures, receptor–ligand complexes, networks of filipodia and dendrites, and other subcellular structures in diverse cell types and tissues. For example, T cells in tissues are also revealed by 3D imaging to have a wide range of morphologies and states, consistent with phenotypic plasticity and branching developmental trajectories[58]. Immune cells are also components of multicellular communities with tightly apposed membranes that appear to involve both complementary and opposing regulatory signals: for example, a single cytotoxic GZMB+CD8+ T cell can be polarized toward a tumor cell, enveloped by filipodia from a CD4 helper cell and repressed by a PDL1-expressing myeloid cell. Precise membrane-level descriptions of tumor architecture are expected to assist in the development of computational models of cell–cell signaling, cell migration and tissue architecture[59].

The thick-section approach described probably represents an optimum for analysis of cells and local communities because it strikes a good balance between resolution, number of possible markers and sparing of tissue (an important consideration with clinical samples). However, a description of mammalian tissues must account for an ~10⁶ range of length scales from intracellular organelles to entire organs; ultimately, this will require integrating methods including super resolution optical microscopy, light-sheet fluorescence microscopy[60,61], microscale computed tomography[20] and 3D transcript profiling. Across this continuum, as length scales increase, achievable resolution usually falls, as does the number of simultaneously addressable markers, but visualization of vessels, nerves and other extended structures greatly improves (Supplementary Note). Thus, the current work is best understood as an approach to the first part of the 3D continuum involving high-resolution imaging of cells, organelles and local neighborhoods.

The 3D tissue imaging is unlikely to be a replacement for 2D approaches, particularly in translational research involving large sets of specimens, because 3D imaging is harder to perform, generates larger datasets and is not directly compatible with existing histopathology workflows. Instead, 3D data are likely to be most useful for detailed and precise study of a more limited number of samples. As in stereology, however[21], even a limited number of accurate 3D datasets on specific tissues and tumors will make it possible to correct for limitations in 2D images and identify new features of tissue not readily discernible in 2D images alone.

## Online content

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

## Methods

Detailed protocols for performing 3D thick-section tissue imaging are available at https://www.protocols.io/view/3d-tissue-cyclic-immunofluorescence-3d-cycif-261ge59m7g47/v1 and will be updated as the methods improve.

### Specimen collection

See Supplementary Table 1 for clinical metadata for all specimens. Specimens for melanoma (MIS and VGP), glioblastoma, lung metastasis and tonsil were retrieved from the archives of the Department of Pathology at Brigham and Women's Hospital and collected under Institutional Review Board approval (FWA00007071, protocol IRB18-1363) under a waiver of consent. Serous tubal intraepithelial carcinoma (STIC) samples were obtained from University of Pennsylvania after institutional review board approval[62]. Three datasets were used for the studies described in the body of the text: two 35-μm serial sections of melanoma (referred to as dataset 1 (LSP13626) and dataset 2 (LSP13625)) and a 35-μm section of metastatic melanoma obtained from the NIH Cooperative Human Tissue Network (referred to as metastatic melanoma or dataset 3; LSP22409/WD-100476). The quantifications are based on dataset 1. Deep immune cell phenotyping was based on features computed from dataset 3. The histopathological regions of interest for each specimen were annotated as described previously[35] by a board-certified pathologist using standard melanoma diagnostic criteria.

### 3D CyCIF

The procedure for 3D CyCIF is modified from the standard CyCIF[7] protocol, with additional care taken during staining steps. Staining plans containing lists of antibodies used with different specimens can be found in Supplementary Tables 3–10. Antigen retrieval, staining and bleaching was performed as described previously[35]. Due to the fragile nature of thicker samples, extra care was taken during washes, bleaching and removing coverslips. Antibodies were diluted in 400 μl of blocking buffer and each section stained for 8–10 h at room temperature to encourage penetration of antibodies but permit same-day imaging. See Supplementary Figs. 1–11 for the whole slide images of the full dataset for all samples.

We found that most tissues held up well to these procedures, but that a subset of melanoma samples disintegrated during antigen retrieval. We have observed this previously with standard section skin and primary melanoma specimens, but the reason why some specimens are more fragile than others remains unknown (such 'preanalytical variables' are common in histology). However, 3D imaging of several specimens (for example, Supplementary Fig. 12) revealed that the tissue had not fully adhered to the slide, instead exhibiting a series of corrugations that touched the slide in only some locations. Further research will be required to overcome this 'corrugation' problem.

### Preparation of fragile samples

For fragile tissue specimens that adhere poorly to microscope slides following dewaxing and antigen retrieval, we developed alternative approaches that did not require removing the coverslip between cycles, which we identified as one contributor to tissue degradation (Extended Data Fig. 1a). In this procedure, FFPE tissue sections were laid directly onto no. 1.5H grade glass coverslips (Extended Data Fig. 1b,c) and then stained, bleached and imaged. This is ideal for the use of high NA objective lenses on inverted microscopes since such lenses are sensitive to coverslip thickness. We found that coating coverslips with poly-L-lysine overnight significantly reduced tissues from lifting off during dewaxing. We then glued a 1-mm-thick spacer (cat. no. IS003, SUNJin lab) around the tissue using cyanoacrylate glue thereby forming a well for the antibody solution or mounting media. Antibody incubation, imaging, bleaching and washing were performed using the standard thick tissue CyCIF approach. During imaging, the well spacer was filled with 70% glycerol and covered with a second coverslip to reduce evaporation. We also explored the use of overlaying 400–500 μl volume of Matrigel (mixed with PBS at 1:1 volume ratio) or a black polyethylene mesh (with seven square holes per inch) on the tissue within the spacer. Both helped to protect the specimen while not interfering with antibody staining. The entire assembly can either be inserted into standard microscope stage inserts or fitted into 3D-printed slide-shaped holders for more convenient handling. Although this second approach to sample preparation requires some optimization, user training and the use of preprinted components, the additional setup time is insignificant compared with the per cycle 3D image acquisition time.

### Optimizing sample thickness

To determine an ideal tissue thickness for CyCIF imaging we used tonsil tissue. Based on the maximum working distance of most water and oil-immersion lenses (~200 μm) and the thickness of a no. 1.5H coverslip (170 μm), we selected tonsil sections that were cut@10, cut@20, cut@30, cut@35 and cut@40 μm thick (Supplementary Fig. 16). These were stained with Hoechst and gamma-tubulin conjugated in Alexafluor 555. Gamma-tubulin is punctate and serves as a useful stain for assessing antibody penetration and image aberrations. Z-stacks were acquired from each stained tissue sampled at 103 nm laterally and 230 nm axially using a 40×/1.2W C-Apochromat water immersion objective lens on a Zeiss LSM980 confocal microscope. We observed punctate gamma-tubulin in all thicknesses up to 35-μm tissue thickness, with uniform intensity along the axial axis (Supplementary Fig. 16a–d). However, at 40 μm thickness, gamma-tubulin intensity significantly diminished along the axial axis (Supplementary Fig. 16e), and contrast (even along the top surface) was poorer than with thinner samples. We speculate that standard dewaxing and antigen retrieval protocols were not working well at tissue thicknesses greater than cut@35 μm. Moreover, we also observed signal attenuation in the Hoechst channel in cut@40 μm specimens. In this case, poor penetration of short wavelength light is probably an issue. Based on these considerations, we concluded that confocal imaging of thick sections is probably to be most effective with samples thinner than cut@30–35 μm.

### Variables in antibody staining

We sought to identify the shortest antibody incubation time needed to homogenously stain a thickness of 35 μm. Unlike 2D widefield imaging where overnight incubation times are often conveniently used, 3D imaging takes significantly longer and we want to accelerate the overall process (Supplementary Fig. 17). Four 35-μm human colorectal cancer tissue specimens were stained with a cocktail of primary conjugated antibodies for 1, 2, 4 and 8 h at room temperature. Z-stacks were acquired with voxel size (400 nm × 400 nm × 290 nm) to reduce file sizes, and the depth of antibody penetration was assessed using orthogonal views in Imaris software. After 8 h of staining, we observed that E-cadherin, CD11c, CD3E and MX1 had homogenously penetrated the entire thickness of the tissue. E-cadherin staining was complete within 2 h. However, vimentin staining was limited to the top and bottom surfaces of the tissue and penetration did not improve with longer incubation times. We suspect this could have to do with the relative distribution of protein across the tissue thickness. It is expected that vimentin (which is expressed by many diverse cell types) would be present in high concentrations compared with other markers (for example, MX1 and CD3E). A high-degree of antibody binding at the outer surfaces may lower the effective concentration of antibody within the center of the tissue, as has been described elsewhere[63].

We also evaluated whether certain fluorophores impacted antibody penetration. This is important for CyCIF where the ability to choose different antibody fluorophore combinations is essential. We obtained a primary melanoma and costained MART1 conjugated to Alexafluor 647 with other secondary antibodies (Alexafluor 488,

Alexafluor 555 and Alexafluor 750) (Supplementary Fig. 18a) for 8–10 h at room temperature. We bleached MART1-647 and restained with Alexafluor 647 in a subsequent cycle. Supplementary Fig. 18b shows that the MART1 primary conjugate (magenta) penetrated the full thickness of the tissue, as judged by Hoechst staining (turquoise). Supplementary Fig. 18c–f shows that all secondary antibodies (magenta) penetrated equally well and showed a similar staining pattern to the MART1 primary conjugate. This demonstrates the ability for secondary antibodies to be used for thick tissue CyCIF. We noted that Alexafluor 750 had lower contrast, which can be attributed to the lower sensitivity of detectors in the near infrared spectrum.

While testing multiple primary conjugated antibodies, we observed antibody penetration issues with some antibody conjugates. Although many immune markers (PD1, CD11c, CD8a, MHC-1 and MHC-II; green) exhibited full depth staining, several tumor and stromal markers (αSMA, PCNA and SOX10; red) only stained the top layer of tissue (Supplementary Fig. 19). To determine whether the fluorophore played a role in this, we repeated staining with the same PCNA clone conjugated to Alexafluor 488 or Alexafluor 750. We noticed there that was a difference in staining pattern; the Alexafluor 488 conjugate-stained fewer cells (Supplementary Fig. 20a) but showed improved staining penetration (Supplementary Fig. 20b). For αSMA, we tried a similar strategy, but using a different fluorophore required a different antibody clone. Unlike PCNA, we did not see an improvement in staining penetration of a blood vessel (Supplementary Fig. 21). From these data we concluded that antibody penetration is not uniquely dependent on fluorophore or clone but is influenced by multiple factors and that each antibody must therefore be evaluated for its ability to stain a thick section using Z-stacks.

### 3D image acquisition

All image data was collected on a LSM980 Airyscan 2 (Carl Zeiss) equipped with a 405, 488, 561, 647 and 750 nm laser lines and 5×/0.16 NA air, 10×/0.45 NA air and 40×/1.3 NA oil-immersion objective lenses. Microscope slides were secured in a slide holder fitted with a spring-loaded clamp, which correspondingly was secured onto the microscope stage in a plateholder. In ZEN 3.9, a 2D overview scan of Hoechst using the 5× objective lens was used to identify regions of interest for higher resolution imaging at 40× in three dimensions. The images were sampled at 16-bit at 0.14 μm per pixel in X and Y and 0.28 μm per pixel in Z for approximately 170 or more optical planes. The pinhole size was set to 35 μm. A focus surface was used to maintain focus. To increase throughput, bidirectional and fast frame scanning was used. The channels were separated into two tracks: track 1: Hoechst, Alexafluor 555 and Alexafluor 750 (if present); track 2: Alexafluor 488 and Alexafluor 647. The emission range for Hoechst, Alexafluor 488, Alexafluor 555, Alexafluor 647 and Alexafluor 750 were 380–489 nm, 499–544 nm, 579–640 nm, 660–705 nm and 755–900 nm, respectively. We note that the current work does not fully exploit the spectral unmixing capabilities of the LSM980[64–66] due to a requirement for additional panel optimization. However, better spectral unmixing in the future is expected to reduce the number of cycles required to collect high-plex data in the future.

Type I and II collagen were imaged using SHG in a Stellaris 8 DIVE coupled to an Insight X3 multiphoton laser and running LasX. The images were acquired with a 20×/0.75 NA multi-immersion lens and sampled at 0.36 μm laterally and 0.95 μm axially. The SHG signal was detected using 4Tune Spectra nondescanned HyD detectors and separated from that of Hoechst 33342 using fluorescence lifetime imaging microscopy.

### 3D image processing and registration

To improve signal-to-noise, all data acquired on the Zeiss LSM980 were processed using Zeiss ZEN LSM Plus Processing. Channels were background subtracted by removing a fixed constant gray-level from the background. The first cycle was stitched in ZEN using the Hoechst channel as a reference, and all subsequent cycles were registered to this first stitched cycle. Single-field and stitched 3D datasets were imported using Bioformats in MATLAB (Mathworks). First, the X and Y translations were obtained using max projections of the Hoechst nuclei channel. Following this transformation, subsequent cycles were registered in Z. We found that separating the lateral from axial transformations was more accurate than registering X, Y and Z in one optimization step. We then performed histogram equalization with MATLAB's histeq() function and fine-tuned image alignment with elastic deformations using MATLAB's imregdemons() function. Lastly, all transformations for each cycle were applied to their corresponding channels. Each channel was saved and appended to a TIFF file and visualized in Meshlab, ChimeraX or Imaris 10.0 (Bitplane) as .ims files. We regard these as interim methods that will benefit from additional automation and refinement in the future.

### Single-cell phenotyping

Manual gating was performed for each marker to differentiate background from true signal. All antibodies had been validated in our laboratory; true signal was determined by comparing signal from a positive control tissue. The gates identified for each marker were subsequently used to normalize the single-cell data within a range of 0–1, wherein values above 0.5 indicated cells expressing the marker. The scaled data was subsequently used for phenotyping the cells based on known lineage markers as described previously using the SCIMAP Python package (scimap.xyz)[35]. See Supplementary Fig. 13 for the detailed marker combinations used to define cell types.

### RCN analysis to identify microenvironmental communities

The latent Dirichlet allocation based recurrent cellular neighborhood (RCN) was performed using SCIMAP (scimap.xyz)[35] using a k value of 10 (Extended Data Fig. 8). The clusters were manually organized into meta-clusters (seven clusters), based on the cellular composition of the clusters. The meta-clusters were also overlaid on the H&E and CyCIF images to validate their characteristics. For instance, RCN1 typically aligned with areas known to be tumor domains, while RCN2 was more closely associated with the epidermis, thereby highlighting the structural elements within the dataset.

### Statistics and reproducibility

The total number of patients in this study is insufficient to make comparison between them meaningful. However, within-specimen single-cell measurements were calculated from at least 120,000 cells (representing biological replicates) across three datasets from two patients with melanoma. With respect to variability from one field of view to the next, we note that there is no uniform definition in a stitched multispectral dataset. In the case of unstitched data, a field of view corresponded to a single image tile of length and width 210 μm × 210 μm and consisted of approximately >1,000 cells. However, we believe that the most relevant statistic for analyzing cell communities comprising less than or equal to ten cells is the number of unique communities as calculated in Extended Data Fig. 1. We estimate this to be >10,000 per specimen. All statistical tests on these niches were performed using MATLAB's ttest2 implementation of the two-sample t-test without assuming equal variances and a significance value of P < 0.05.

### Cell type calling in virtual thin sections

Three sections of varying thicknesses (cut@5, cut@10, cut@20 and cut@35 μm) were cut from a FFPE block, processed, stained with Hoechst and imaged as described above. The mean thickness of random regions of interest were measured using orthogonal views in Imaris. We observed that if a FFPE section was cut at 5 μm, it will inflate to 9 μm after rehydration. Therefore, from our datasets and corresponding cell masks, we created virtual 9-μm-thick serial sections and compared

them with the entire 35-µm-thick section. To enumerate the percentage of incomplete cells in these thinner sections, we used MATLAB's regionprops3 bounding box to find cells that had top or bottom faces coinciding with the top or bottom of the virtual section respectively. The volume property was used to compare the average cell volume that would be missing from each virtual section from its corresponding whole volume in the 35-µm section. To determine degree of cell type miscalling in thin sections, we performed spot counts of LAG3, GranzymeB and MX1 and compared with the corresponding thick section. A cut-off of >2 spots was used to identify positivity in both thin and thick sections. We calculated the percentage of positive cells that were misclassified as negative cells due to spot exclusion in a virtual thin section. Graphs were plotted in R showing mean and interquartile range.

### Cell interaction analysis in virtual thin sections

A densely packed volume of 56 µm × 56 µm × 35 µm was selected and cropped from the epidermis region of dataset 1−melanoma in situ for cell interaction analysis and 3D segmentation. From each cell, we measured the cell centroid and the distance to the nearest unobstructed neighboring cell edge/membrane to identify neighbors. We further used a minimum distance cut-off of 2 µm to identify neighbors that involved membrane contact. The number of missing neighbors in all virtual sections was normalized to the number of neighbors in the full tissue thickness, which was taken as ground truth ((number of neighboring cells in 2D)/(number of neighboring cells in 3D)).

To test the ability of detecting neighbor cells in 2D optical planes, we extracted single $Z$-planes spaced three planes apart. We filtered out cells with an area of less than 2 µm² (25 pixels). For thicker sections, we started from the middle $Z$-plane and systematically increased the thickness by 12 planes. We filtered out cells with a volume less than 2.7 µm³ (125 voxels). Finally, we also tested a maximum projection of a 9 µm virtual section to simulate widefield imaging of a traditional 5-µm section (adjusted for hydration). Graphs were plotted in MATLAB (Mathworks).

### B cell sphericity analysis

B cells in the melanoma in situ were segmented in 3D in Imaris using the Surfaces module with a smoothing of 0.14 µm per pixel. Noncellular bright objects were manually removed. Since B cells were more compact in the VGP, we used Arivis' Cellpose Cyto2 model implementation based on the membrane marker MHC-1 and a cell diameter of 8 µm. B cells were manually selected by having a CD20 mean intensity above 750 gray-level units and a minimum volume of 25 µm². In both software, the calculation for sphericity is the same and was based on each cell's mask instead of voxels.

### Calibration curve of FFPE tissue thickness

FFPE mouse thymus tissue was sectioned at different thicknesses of 5, 10, 20 and 35 µm. We cut three sections for each thickness and stained each with 4′,6-diamidino-2-phenylindole (DAPI) and LDS751 (ThermoFisher Scientific) overnight in PBST (0.1%). A 3D stack of each tissue section was imaged in 70% glycerol ($n = 3$). The thickness was measured in the $XZ$ orthogonal view in ImageJ with the merged channels. The regression analysis was performed using measured thicknesses as the dependent variable and set nominal thicknesses on the microtome as the independent variable. Mean values were calculated for each nominal thickness, with standard deviation used to construct error bars on the calibration plot. The analysis was carried out using Python (v3.9.16), and the linear regression model was implemented from the scikit-learn package (v1.4.1.post1). The plot was generated using the matplotlib package (v3.4.3).

### Segmentation of individual 3D cells with Cellpose

Individual 3D cells were segmented from the dense tissue volumes using a new modification of Cellpose designed for 3D data[67] (see Zhou et al.

for a detailed description of the software package developed for this novel segmentation approach). The original 2D Cellpose model (https://github.com/MouseLand/cellpose)[68], is a custom gradient tracking approach that aggregates $x−y$, $y−z$, $x−z$ 2D slice cell probability and gradient maps predicted by pretrained 2D segmentation models. The full Cellpose segmentation framework, suitable for a wide range of 3D cell imaging data along with in-depth validation and determination of method applicability specific to this project, is described below.

**Image preprocessing for Cellpose.** The 3D volumes were acquired at voxel resolution of 140 nm × 140 nm × 280 nm. For each 3D channel image, we resized the $x−y$ slices by half to obtain isotropic voxels. The raw image intensity, $I_{raw}^{ch}$, was then corrected for uneven illumination, $I_{correct}^{ch} = I_{raw}^{ch} \frac{I_{raw}^{\bar{h}}}{I_{bg}^{ch}}$, where $I_{raw}^{\bar{h}}$ is the mean image intensity and $I_{bg}^{ch}$ an estimation of the background illumination obtained by downsampling the image by a factor of 8, Gaussian smoothing with sigma = 5 and resizing back to the original image dimensions. $I_{correct}^{ch}$ was then contrast-stretched to a range of 0–1, clipping any intensities less than the 2nd percentile to 0 and any greater than the 99.8th percentile to 1. Cellpose uses a single channel cytoplasmic and nuclear signal for two-color based cell segmentation. The mean of the intensity-normalized, background-corrected HLA-AB, CD3E, CD11b and β-actin channels was used as the cytoplasmic signal. DAPI was used as the nucleus signal. Both cytoplasmic and nucleus signals underwent a further round of background correction and contrast stretching as described above before being concatenated to form the input RGB volume image.

**Initial Cellpose 2D segmentation.** The RGB volume was input slice-by-slice to Cellpose 2D in three different orientations: $x−y$, $x−z$ and $y−z$ to obtain three stacks of cell probability and 2D gradients. The performance of Cellpose depends on appropriate setting of the diameter parameter which relates to the size of the cells to be segmented. As the appearance of the cells may vary depending on orientation, we conduct a parameter screen with diameter = [10,100] at increments of 5 using the mid-slice for each orientation. At each diameter we compute the 'sharpness' of the predicted gradient map as the mean of the image variance evaluated over a local 5 × 5 pixel window in both '$x$' and '$y$' gradient directions. The diameter maximizing the variance after a moving average smoothing with window size of 3 was used to run Cellpose 2D on the remaining slices in the orientation. The raw cell probability output, $P$ from Cellpose are the inputs to a sigmoid centered at zero, $1/(1 + e^{-P})$. This means the probabilities vary predominantly linearly in the range −6 to +6, and this reduces the distinction between foreground and background. Thus, we clip the probabilities to the range [−88.72, 88.72] (to prevent overflow or underflow in float32) and convert back to a normalized probability value in the range 0–1 by evaluating the sigmoid, $1/(1 + e^{-P})$. The probabilities from all three orientations are combined into one by averaging. Similarly, the 2D gradients are Gaussian smoothed with sigma = 1 voxel and combined into a single 3D gradient map. Gradients are then normalized to be unit length. Lastly, we perform 3-level Otsu thresholding on the combined probability map and use the lower threshold to define the foreground binary voxels for gradient tracking.

**Aggregating Cellpose 2D predictions.** The volume was divided into subvolumes of (256, 512, 512) with 25% overlap. Within each subvolume we run gradient descent with momentum for 200 iterations, momenta, $\mu = 0.98$, step size $\delta = 1$ to propagate the position of foreground pixels toward its final attractor in the 3D gradient map

$$\left( x_i^{t+1}, y_i^{t+1}, z_i^{t+1} \right) \leftarrow \left( x_i^t, y_i^t, z_i^t \right) + \frac{1}{\delta + \mu} \left( \delta \nabla \left( x_i^t, y_i^t, z_i^t \right) \right.$$
$$\left. + \mu \nabla \left( x_i^{t-1}, y_i^{t-1}, z_i^{t-1} \right) \right),$$

where $(x_i^t, y_i^t, z_i^t)$ denotes the coordinate of foreground voxel $i$ at iteration number $t$, $\mu$ the momentum ranging from 0–1, $\delta$ the step size and $\nabla$ is the gradient map. Nearest neighbor interpolation is used, thus $(x_i^t, y_i^t, z_i^t)$ is always integer valued. Gradient tracking of all subvolumes are conducted in parallel using multiprocessing. The final coordinate positions from all subvolumes are compiled. We then build a volume count map where voxels mapping to the same final coordinate adds +1 to the count. The count map is Gaussian smoothed with sigma = 1 and binarized using the mean value as the threshold. Connected component analysis identifies the unique cell as clusters where foreground voxels have been mapped to the same cell. Transferring this labeling to initial voxel positions $(x_i^{t=0}, y_i^{t=0}, z_i^{t=0})$ generates the individual 3D cell segmentations.

**Postprocessing 3D cell segmentations.** Small individual cell masks (<1,000 voxels$^3$ ≈ 20µm$^3$) were first removed because they corresponded to debris. We also removed all cell masks that do not agree with the Cellpose predicted 3D gradient map. This is done by computing the 3D heat diffusion gradient map given the computed 3D cell segmentations and computing the mean squared error with the input combined Cellpose 3D gradient map for each cell. Cells with mean squared error >0.8 were discarded. Cells that are implausibly large, with volume greater than the mean volume ±5 standard deviations were also discarded.

For the remainder cells, we run a label propagation[69] to enforce that each segmented cell mask comprises only a single connected component and to denoise the masks. This is done for each cell mask, $M_i$, by cropping a subvolume, $V_i$, the size of its bounding box padded isotropically by 25 voxels. Each unique cell region is represented as a positive integer label. Every label in $V_i$ is encoded using a one-hot encoding scheme to create a binary column for each unique label. This generates a label matrix, $L_i \in \mathbb{R}^{N\times(p+1)}$ for $V_i$, where $N$ is the total number of voxels and $p$ the number of unique labels in $V_i$ and one additional label for background. We then construct the affinity matrix, $A$, as a weighted sum ($\alpha = 0.25$) of an affinity matrix based on the intensity difference in the cytoplasmic signal between 8-connected voxel neighbors, $A_{\text{intensity}}$, and one based on the connectivity alone, $A_{\text{laplacian}}$; $A = \alpha A_{\text{intensity}} + (1-\alpha) A_{\text{laplacian}}$. $A_{\text{intensity}} = \begin{cases} e^{-D_{\text{intensity}}^2/(2\mu(D_{\text{intensity}})^2)}, & i \neq j \\ 1, & i = j \end{cases}$, where $D_{\text{intensity}}$ is the pairwise absolute difference matrix between two neighboring voxels $i$ and $j$. $A_{\text{laplacian}} = \begin{cases} e^{-D_{\text{laplacian}}^2/(2\mu(D_{\text{laplacian}})^2)}, & i \neq j \\ 1, & i = j \end{cases}$, where $D_{laplacian}$ is the graph Laplacian with a value of 1 if a voxel $i$ is a neighbor of voxel $j$, and 0 otherwise. $\mu(D)$ denotes the mean value of the entries of matrix $D$. The iterative label propagation is

$$z \in \mathbb{R}^{N \times p}$$

$$z^{t=0} = \mathbf{0}$$

$$z^{t+1} \leftarrow (1-\gamma)Az^t + (\gamma)L,$$

where $t$ is the interation number, $\mathbf{0}$ denotes the empty vector and $\gamma$ is a 'clamping' factor that controls the extent the original labeling is preserved. We set $\gamma = 0.01$. We run the propagation for 25 iterations. The final $z$ is normalized using the softmax operation, and argmax is used to obtain the final labels. The refined cell mask, $M_i^{\text{refine}}$ is defined by all voxels where $z$ has the same cell label $i$. All postprocessing steps were implemented using parallel multiprocessing iterating over individual cells.

**Comparison of tissue thickness before and after hydration, dehydration and rehydration**

**Tissue preparation and 3D image acquisition.** Kaede mouse thymus tissue was fixed in 4% paraformaldehyde and stored in PBS at 4 °C. The tissue was embedded in 8% agarose for vibratome sectioning (Leica VT1000 S) at a thickness of 35 µm and mounted on frosted glass slides. The section was stained with Hoechst 33342 and imaged in the hydrated state. The tissue was then mounted with 70% glycerol and imaged with a high-precision coverslip no. 1.5H (ThorLabs) on a Zeiss LSM980 confocal microscope with an EC Plan-Neofluar 40×/1.30 oil DIC M27 objective.

For tissue dehydration, the same sections were dehydrated through a series of ethanol solutions (50%, 70%, 95%, two changes of 100%) and xylene (two changes). The dehydrated sections were then mounted in a toluene based mounting media (Permount, Fisher Scientific) and imaged again. Subsequently, the sections were decoverslipped in xylene for 2 h and then rehydrated through two changes of xylene and graded ethanol back to PBS (as previously described). The rehydrated section was imaged in the same condition as the hydrated state in 70% glycerol as previously described.

**Image processing, tissue thickness measurements and statistical analysis.** $Z$-stacks were observed in ImageJ (version 1.54f) as $XZ$ orthogonal views to measure tissue thickness. We selected an orthogonal plane from each of the three conditions (hydrated, dehydrated and rehydrated) and measured the tissue top and bottom in three locations. These were selected to be approximately at the same locations between the tissue sections for consistency.

Tissue thickness measurements were compared across three hydration states of the same sample: hydrated, dehydrated and rehydrated. Paired $t$-tests was conducted in Python (v.3.9.16) using the scipy.stats package to assess the effect of hydration state on tissue thickness. The error bars in the accompanying figure represent the standard deviation for each treatment group. The plot was generated using the matplotlib package (v.3.7.1), and data processing was performed using pandas (v.2.2.1) and NumPy (v.1.24.3) packages.

## Reporting summary

Further information on research design is available in the Nature Portfolio Reporting Summary linked to this article.

## Data availability

All primary images and derived data (-5 TB) will available via AWS transfer at the time of publication. Instructions for accessing the primary and derived data are available on a data index page via Zenodo at https://doi.org/10.5281/zenodo.10055593 (ref. 70). These images can be viewed using the free ImarisViewer (https://imaris.oxinst.com/imaris-viewer). The 2D maximum projections of each dataset can be viewed in the MINERVA viewer (no download required), please visit https://www.tissue-atlas.org/atlas-datasets/yapp-nirmal-2023 or see Supplementary Table 1 for links. A subset of data will be available for 3D interactive viewing within the browser-based tool Vitessce (http://vitessce.io/)[71]. This effort is a work in progress and will be available in the future.

## Code availability

Original code associated with this Article is available via GitHub at https://github.com/labsyspharm/mel-3d-mis and via Zenodo at https://doi.org/10.5281/zenodo.10055593 (ref. 70).

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

## Acknowledgements

We thank T. Kupper, D. Liu and J. Agudo for scientific advice, A. Chen, S. Chan, J. Muhlich and J. Hoffer for help with data analysis; N. Ghelenborg and E. Moerth for access to the 3D Vitesse data viewer; J. Lian for model training; and J. Appelt for tissue integrity studies and the MicRoN core facility at HMS for assistance with microscopy. We also thank G. Guimaraes, T. Desai and S. Fore from Carl Zeiss Inc. for providing extended access to the LSM980 Airyscan 2 microscope used to collect data in this proposal. We thank M. Baym for access to a Dremel 3D45 3D printer used to make the slide holders. This work was supported by the Ludwig Center at Harvard (P.K.S. and S.S.), a CCBIR grant U54-CA268072 (G.D., P.K.S. and S.S.), NCI grants R00CA256497 (A.J.N.), a Research Specialist Award R50-CA252138 (Z.M.), a Team Science Grant from the Gray Foundation (P.K.S. and S.S.) and the Mark Foundation for Cancer Research (P.K.S. and S.S.). Histopathology was supported by P30-CA06516. S.S. is supported by the BWH President's Scholars Award. The funders had no role in study design, data collection and analysis, decision to publish or preparation of the manuscript.

## Author contributions

C.Y., P.K.S and A.J.N. developed the concept for the study. C.Y., A.W., Z.M. and P.M.L. collected image data. Y.L., Z.S., F.Z. and G.D. developed software and performed data analysis in collaboration with C.Y., A.J.N. and J.B.T. S.S., G.F.M. and C.G.L. provided specimens and pathology expertise. C.Y., P.K.S. and J.B.T. wrote the manuscript. All authors provided edits and approved the final manuscript. G.D., G.F.M., C.G.L., S.S. and P.K.S. provided supervision.

## Competing interests

P.K.S. is a cofounder and member of the BOD of Glencoe Software, member of the BOD for Applied Biomath and member of the SAB for RareCyte, NanoString, Reverb Therapeutics and Montai Health; he holds equity in Glencoe, Applied Biomath and RareCyte. P.K.S. consults for Merck, and the Sorger lab has received research funding from Novartis and Merck in the past 5 years. The other authors declare no competing interests.

## Additional information

**Extended data** is available for this paper at https://doi.org/10.1038/s41592-025-02824-x.

**Correspondence and requests for materials** should be addressed to Peter K. Sorger.

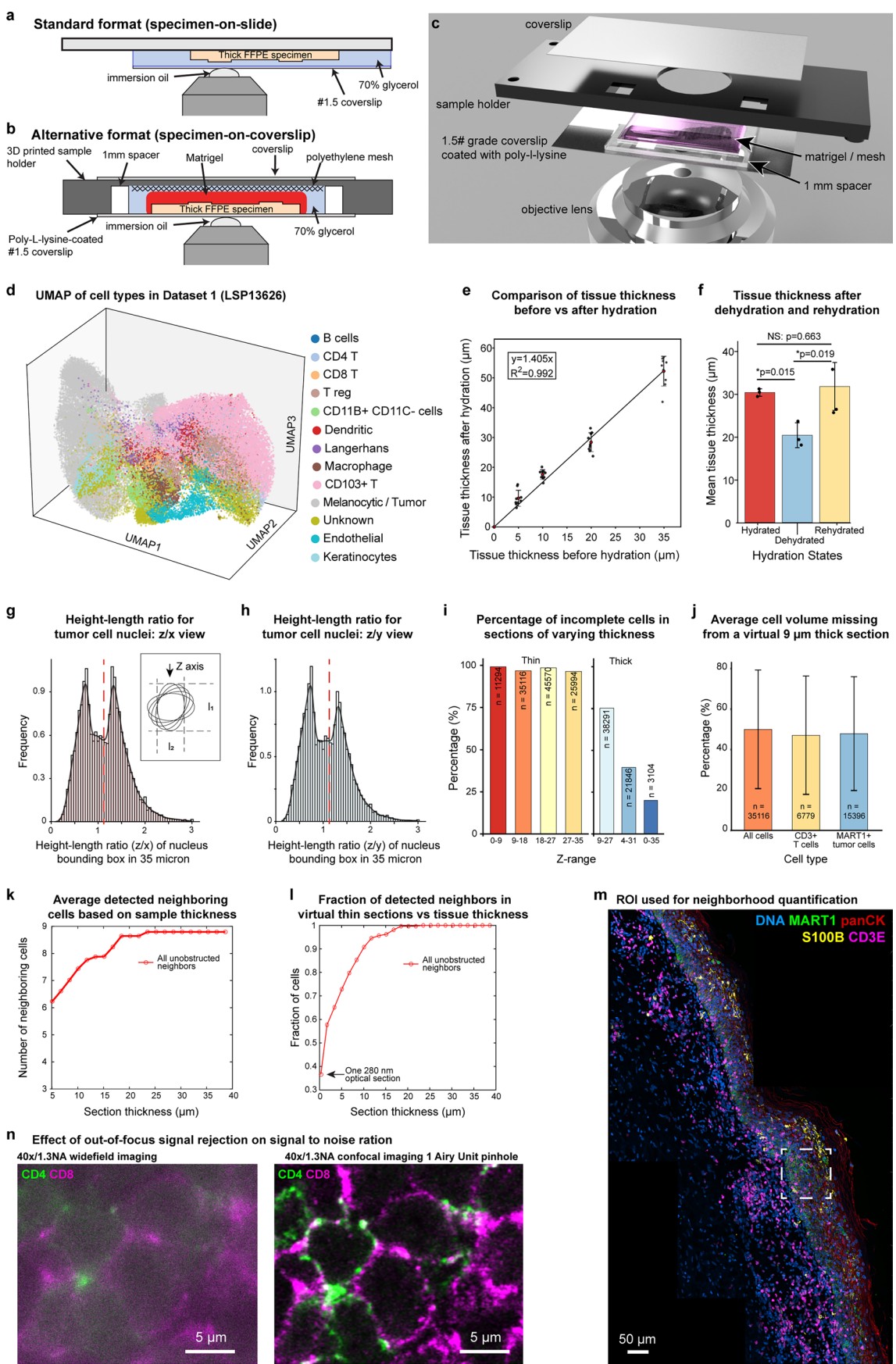

**Extended Data Fig. 1 | See next page for caption.**

**Extended Data Fig. 1 | Imaging of thick tissue sections using 3D CyCIF.** Two methods for 3D imaging of thick tissue sections using high NA (high-resolution) oil immersion objectives. **a**, The standard arrangement in which a stained specimen is mounted to a glass slide and overlaid with a coverslip in 70% glycerol mounting medium. The coverslip is removed after each imaging cycle for fluorophore inactivation and another round of staining[7]. This approach is often satisfactory but can result in damage to some tissues as cycle number increases. All data shown in the main figures of this paper was collected using this method. **b**, An alternative arrangement developed for imaging tissues prone to damage. The specimen is mounted to the coverslip and overlaid with Matrigel and a polyethylene mesh to hold the tissue in place, the assembly is fitted into a 3D printed holder and covered with a second coverslip to enable imaging with standard microscope slides. Use of this approach is demonstrated for a 35-µm section of colorectal cancer across 6 CyCIF cycles in Supplementary Fig. 12a and a melanoma precancer sample across 3 cycles in Supplementary Fig. 12b. **c**, Rendering of the holder used for specimen-on-coverslip imaging shown in **b**. **d**, UMAP rendering of all cell types analysed in Dataset 1 (LSP13626) as generated using 3D image segmentation algorithms. See Supplementary Fig. 13 for flow chart of cell type classifications. **e**, Comparison of tissue thickness before and after hydration. Tissue was sectioned at nominal thicknesses of 5, 10, 20, and 35 µm. For each thickness, measurements were taken at three different positions from each of the three samples, pooled for analysis ($n = 9$ per group). Individual datapoints are overlaid. Error bars represent standard deviation of pooled measurements. Mean hydrated thickness ± SD.: 9.01 ± 2.53 µm (5 µm), 16.48 ± 1.52 µm (10 µm), 26.67 ± 2.98 µm (20 µm), and 49.03 ± 4.68 µm (35 µm). A linear regression constrained through the origin was fitted to the data ($y = 1.405x$, $R^2 = 0.992$). **f**, Tissue thickness under three conditions: PFA fixed mouse thymus tissue were sectioned to a thickness of 35 µm with a vibratome and mounted on glass slide under three conditions, mean thickness ± SD: hydrated (30.17 ± 0.88 µm), dehydrated (20.25 ± 2.88 µm), and rehydrated (31.58 ± 5.57 µm). Hydrated samples were mounted in 70% glycerol; dehydrated samples were processed through ethanol/xylene and mounted in Permount; rehydrated samples were previously dehydrated sections returned to PBS. Thickness was measured at three ROIs per sample ($n = 3$, technical replicates). Individual measurements are overlaid. Significance was assessed using paired two-tailed t-tests. Asterisks indicate P < 0.05; ns, not significant. **g**, Height vs length ratio for computed bounding boxes covering ellipsoidal tumour cell nuclei in the VGP (having major and minor axes $l_1$ and $l_2$) as viewed in the Z,X plane. As depicted in the inset, top-down views (along the Z axis) would produce a distribution of apparent lengths representing $l_1$ and $l_2$; the fact that these are nearly equivalent in number in the data suggests no significant distortion along the imaging axis. **h**. Same analysis as **j** but viewed from the Z,Y plane. **i**. Percentage of cells from 35-µm Dataset (LSP13626) that would be incomplete in 9-µm virtual sections (positioned along the optical axis) compared to sections 18-35 µm thick. n represents the number of cells analysed. **j**, Percentage of cell volume missing when Dataset 1 was sectioned into virtual 9 µm thick sections for all cells, CD3+ T cells, and MART1+ tumour cells. Error bars represent the interquartile range. n value is the number of cells analysed ($n = 35{,}116$ for all cells, $n = 6{,}779$ for CD3 + T cells, $n = 15{,}396$ for MART1+ tumour cells). Note that tumour cell nuclei in this specimen are not much larger than immune cell nuclei; truncation of tumour cells would be more severe with large, pleomorphic nuclei. **k**, Average number of observed cell interactions between nearby cells as thickness of virtual section increases. **l**, Fraction of observed cell interactions with increasing virtual section thickness normalized to known number of cell interactions identified from full tissue thickness. **m**. Maximum intensity projection of Dataset 1 showing the cropped region used for analysis in **k** and **l**. **n**, Images of the same cut@5µm tissue section acquired with traditional widefield microscopy (left) and laser scanning confocal microscopy (right) with the same objective lens and microscope (40x/1.3NA, LSM980). Note that existing slide scanners operate at lower NA than the image shown here (typically 0.5 to 0.95 non-oil immersion objectives), resulting in higher signal-to-noise ratio in the widefield image. These data are quantified in Supplementary Note 1 to facilitate conversion of 3D confocal data into 2D representations.

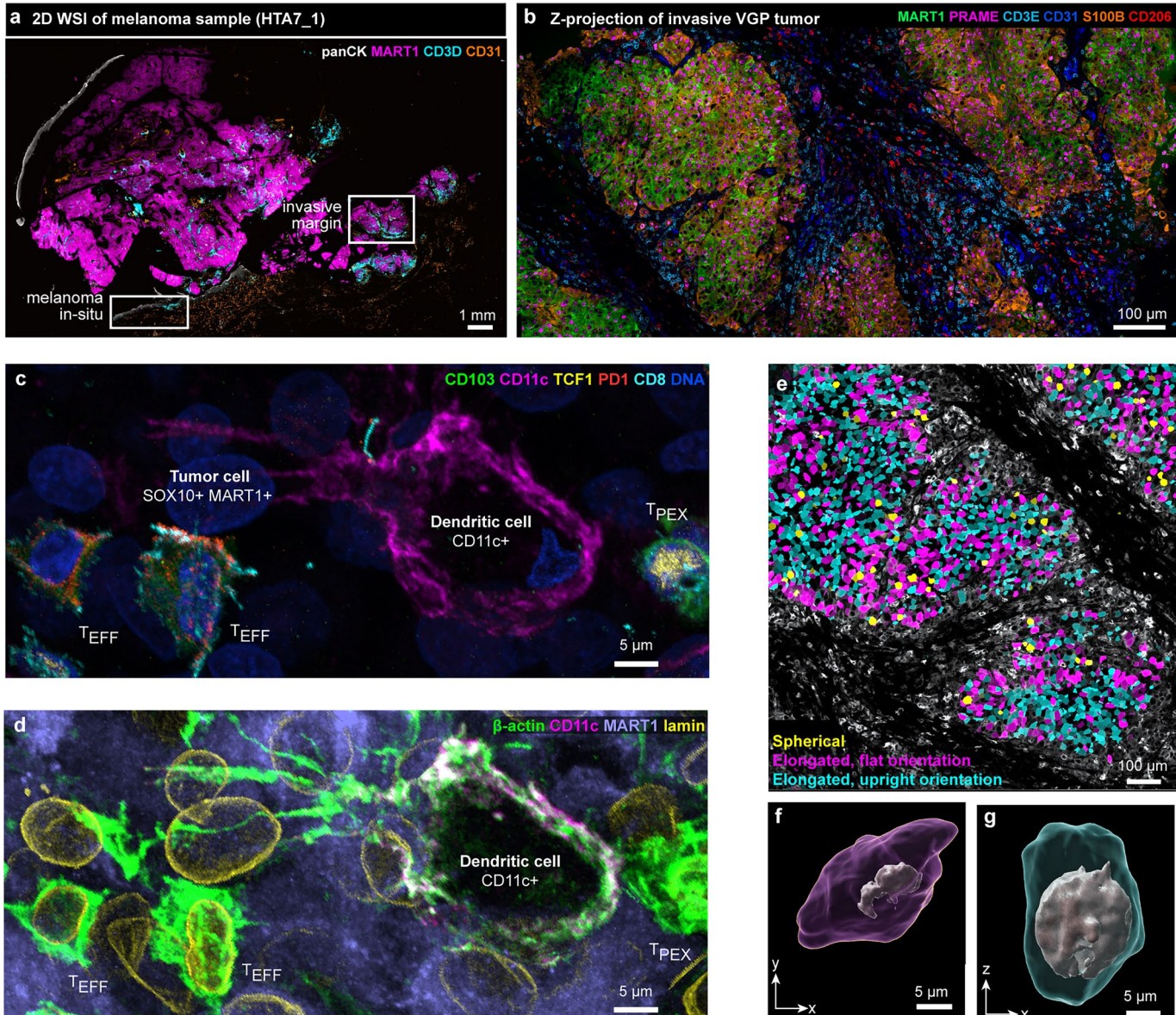

**Extended Data Fig. 2 | Visualizing elongated immune and tumour cells in dense region from vertical growth phase. a**, 2D CyCIF whole slide image of adjacent section of the primary melanoma sample HTA7_1. White squares indicate the regions of melanoma in-situ (MIS) and the invasive vertical growth phase (VGP) melanoma where high-resolution 3D CyCIF was performed. Marker colours as indicated. Scale bar 1 mm. **b**, Maximum projection of the region of the invasive margin imaged with 3D CyCIF of Dataset 1 (LSP13626), showing a subset (6) of the total 54 markers. Image corresponds to right ROI indicated in (**a**).

Scale bar 100 μm. **c-d**, Dendritic cell from Fig. 1j with additional markers highlighting T cell subtypes (**c**) and the dense neighbourhood of tumour cells (**d**). See Supplementary Fig. 13 for detailed T cell subtype calling. **e**, Colour-coded segmentation masks of tumour cells from vertical growth phase of melanoma region. Colour encodes for orientation of tumour cells (spherical cells in yellow; elongated cells in magenta and cyan). Scale bar, 100 μm. **f, g**, Surface renderings of selected cells from (**e**). These cells differ in orientation in the tissue rather than aspect ratio.

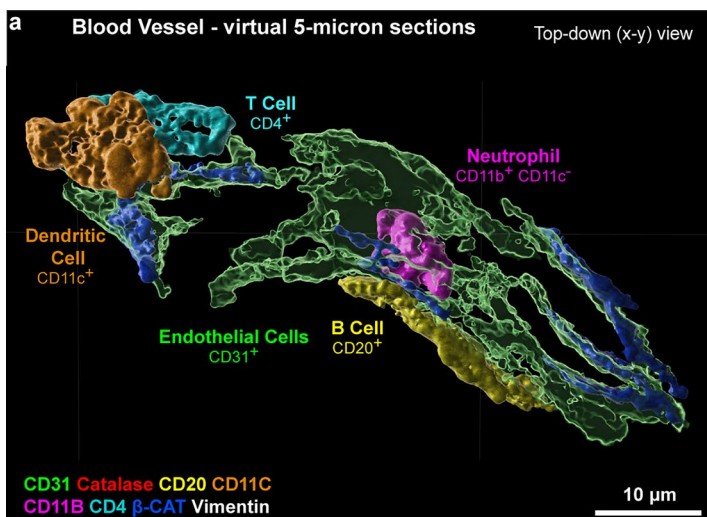

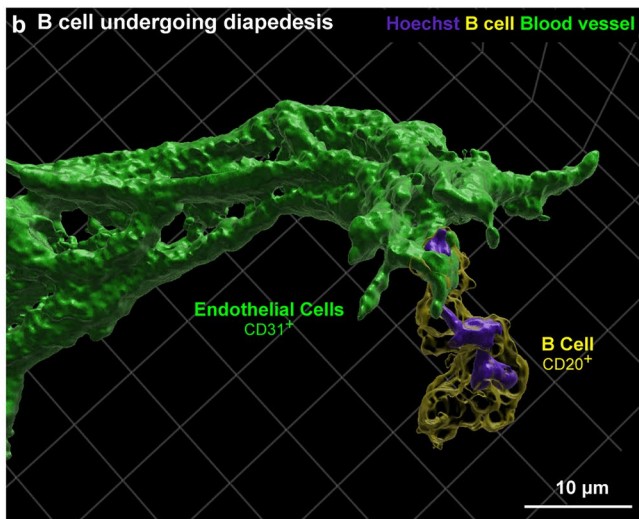

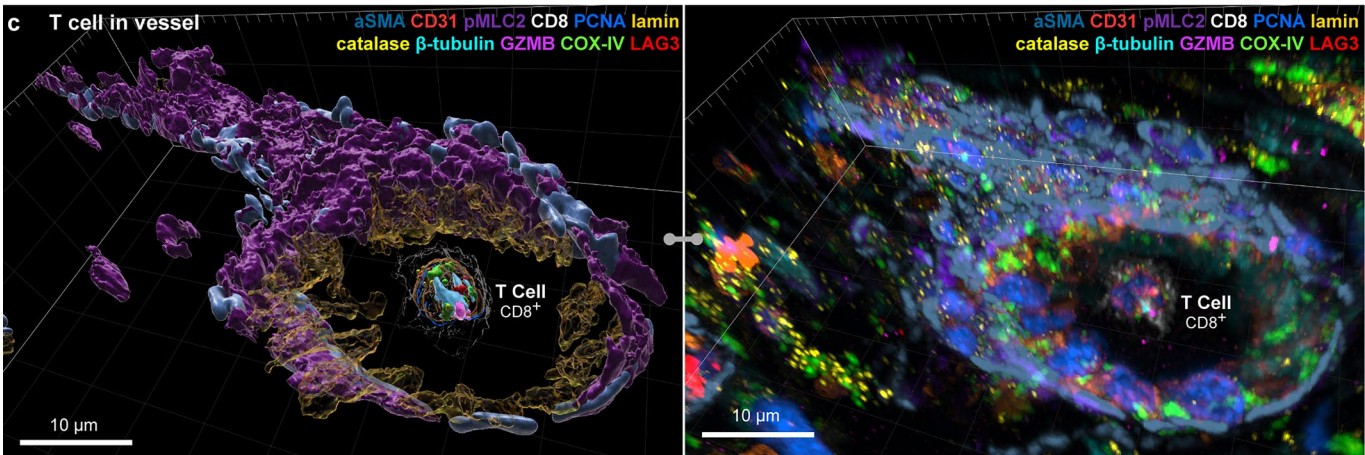

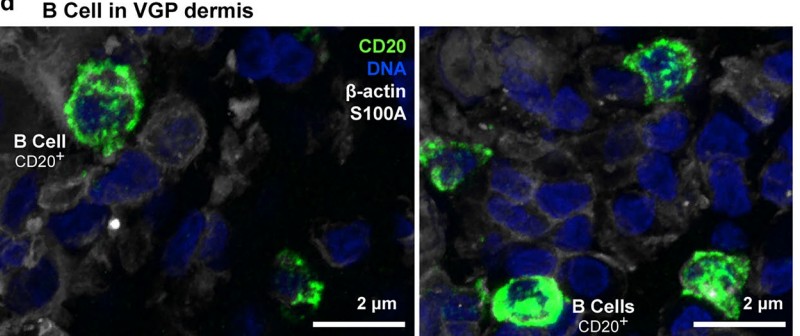

**Extended Data Fig. 3 | Visualizing multi-cellular structures, cell shape, and motility in native tissue. a**, Surface rendering of 5-µm virtual section of the blood vessel from Fig. 2a, viewed from above, showing B cell undergoing trans-endothelial migration (diapedesis). Scale bar 10 µm. **b**, Surface rendering of a different B cell (yellow) elsewhere in dermis of melanoma in-situ migrating through vessel wall (green) Scale bar 10 µm. **c**, T cell inside vessel, shown as a surface rendering (left) and as volumetric rendering (right). **d**, B cells in the VGP dermis with a characteristic round morphology. **e**, Sphericity (left) and volume (right) of B cells in the entire MIS (n = 15) and VGP (n = 352) regions from 1 biological replicate from 1 specimen (Dataset 1 - LSP13626). Data is represented as mean +/-SD.

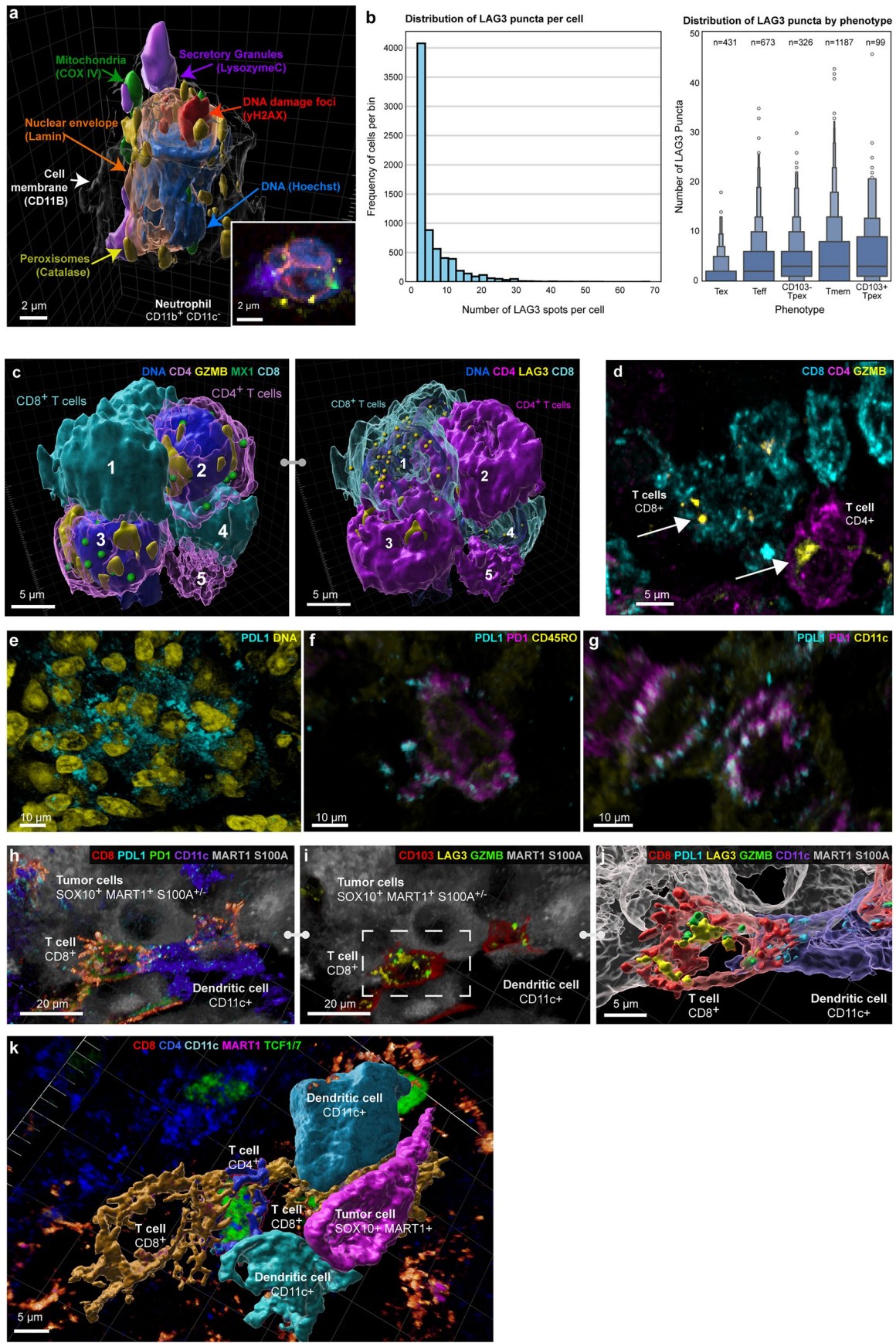

**Extended Data Fig. 4 | See next page for caption.**

**Extended Data Fig. 4 | Visualizing complex organelle and cell-surface morphologies. a**, 3D rendering of a neutrophil with organelles labelled. **b**, Left: Histogram displaying the frequency distribution of LAG3 spots per cell. The x-axis represents the number of LAG3 spots identified within individual cells, and the y-axis indicates the frequency of cells corresponding to each LAG3 spot count. Right: Distribution of LAG3 puncta by cell type. Shown as a boxen plot, where each box represents a quantile range, progressively detailing the distribution from the median outwards to the extremes. Open circles indicate outliers. **c**, Surface rendering of five interacting immune cells in the MIS, including three CD4$^+$ helper T cells (magenta) and two CD8$^+$ T cells (cyan). Left: MX1 biomolecular condensates (green), globular GZMB$^+$ (yellow) in CD4$^+$ T cells. Spacing between opposed membranes is <1.5 μm and contact area is ~20 μm$^2$. Scale bars 5 μm. Right: Reversed opacity of left image, showing punctate LAG3 (yellow) on the membranes of CD8$^+$ T cells. **d**, Comparison of GZMB morphology (arrows) within a CD8 and a CD4 T cell in the MIS. **e-g**, Volumetric renderings showing PD1 and PDL1 can manifest as different morphologies within the same sample (Dataset 3 - LSP22409). **e**, Diffuse PDL1 on dendritic and CD8$^+$ T cells. **f**, Diffuse PD1 and punctate PDL1 within the same dendritic cell. **g**, colocalization of punctate PD1 and PDL1 within the same dendritic cells. **h**, Activated LAG3$^+$ GZMB$^+$ CD103$^+$ PD1$^+$ CD8$^+$ T cell with long filopodia (red) and dendritic cell (purple) interacting with a tumour cell. Note: This is a different cell community from that in Fig. 2p-r. **i**, Same cells as (**h**) with GZMB and LAG3 shown, highlighting that the T cell is activated and cytotoxic. **j**, surface rendering of interactions in (**h**) and (**i**), showing the filopodia in greater detail. **k**, 3D rendering of cells highlighted in the multicellular interactions in Fig. 3f-g and Supplementary Video 6.

**a**

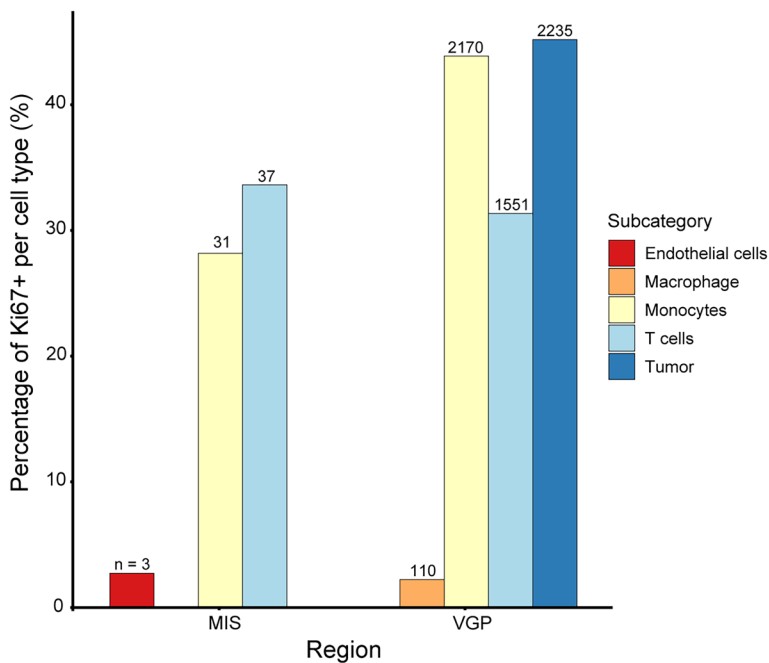

**The cell type compositioin of Ki67+ cells in melanoma in-situ and invasive margin**

**b**

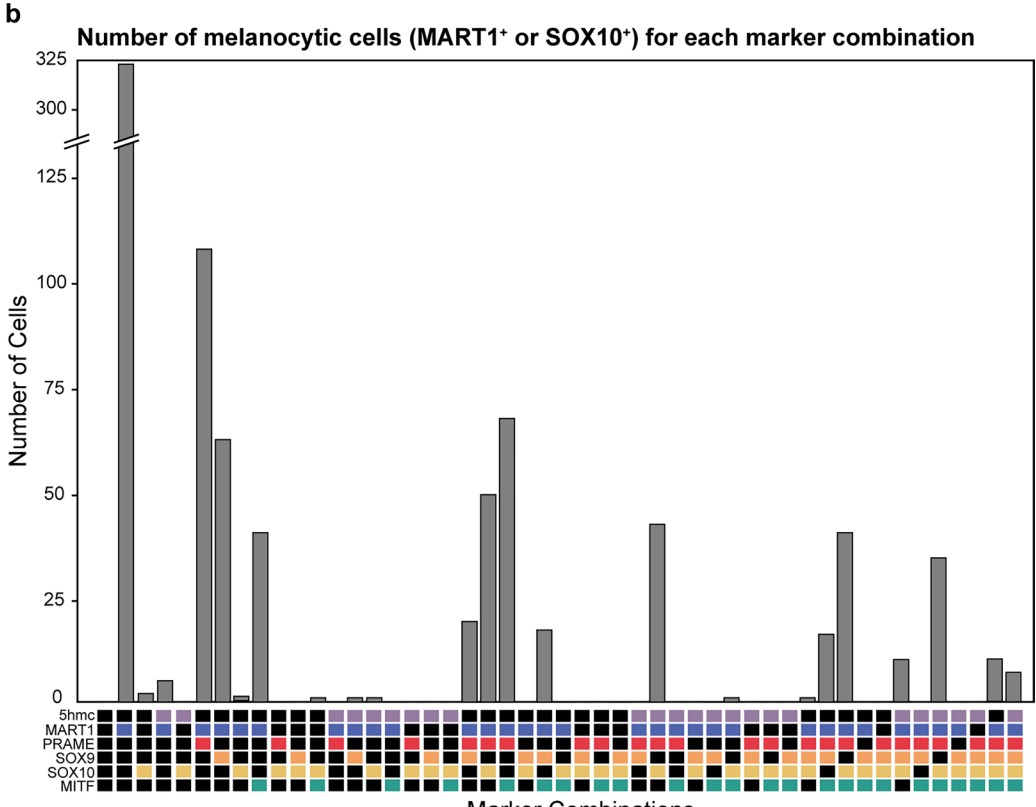

**Number of melanocytic cells (MART1⁺ or SOX10⁺) for each marker combination**

**Extended Data Fig. 5 | Distribution of KI67 cells by phenotype in melanoma tissues. a**, Quantification of the Ki67+ cells within the MIS and VGP regions. Sample size is indicated above each bar. **b**, Bar graph showing the number of melanocytic cells in the MIS positive for each combination of six proteins (63 possible combinations): MART1 (green), PRAME (yellow), MITF (orange), SOX10 (red), SOX9 (blue), and 5hmc (violet). n = 857. All categories with fewer than 15 cells were verified by manual inspection. Note different y-axes.

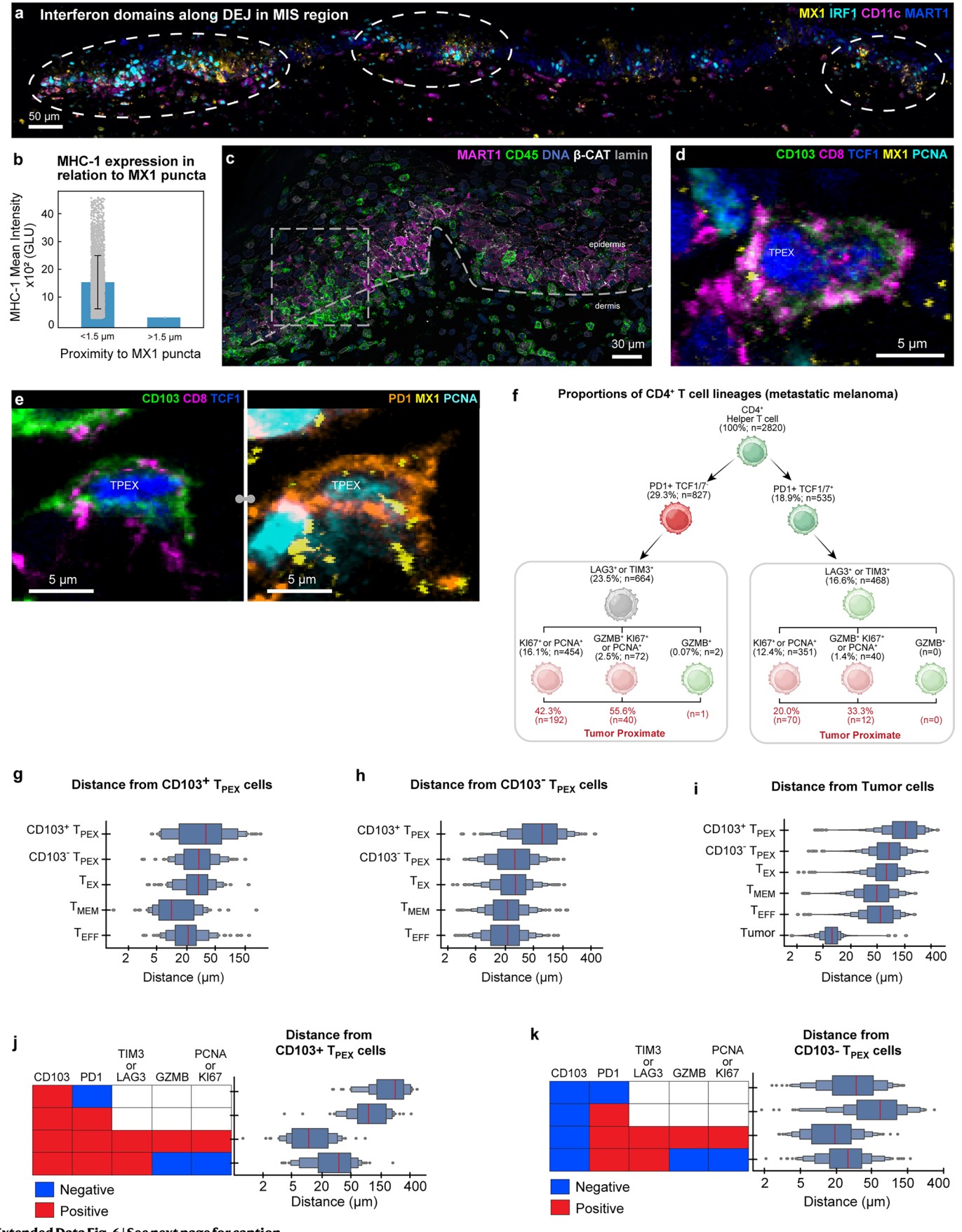

**Extended Data Fig. 6 | See next page for caption.**

**Extended Data Fig. 6 | Spatial analysis of IFN-rich domains and distinct T cell lineages. a**, 2D CyCIF image of the MIS showing the correlation between pockets of MX1 (yellow) and IRF1 (cyan) along the dermal epidermal junction (dark blue). CD11c+ Dendritic cells shown in purple. Scale bar 50 μm. **b**, Quantification of MHC-1 expression in Gray Level Units (GLU) < 1.5 μm (proximal) or >1.5 μm (distant) from 3921 MX1 punctum along the entire length of the DEJ in one independent dataset (entire MIS and VGP regions of dataset 1; LSP13626). Data is represented as mean value +/- SD after removing top and bottom 5% outliers. Sample size (n) is 1 biological replicate from 1 specimen. **c**, Five selected channels from 42-plex CyCIF image of MIS in dataset 2. DEJ denoted by white dashed lines. White dashed rectangle is enlarged in Fig. 5c and exemplifies CD45+ immune cells (green) breaking through the DEJ into the epidermis. Scale bar 30 μm. **d-e**, Two examples of maximum projections of TCF1+ T$_{PEX}$ cells, showing non-proliferation

(**d**) and proliferation via PCNA staining (**e**). Scale bar 5 μm. **f**, Hierarchical tree diagram showing proportions of CD4 + T cell sub-lineages in the metastatic melanoma sample. Red values indicate the percentage of cells in the associated state that are next to tumour cells. **g-k** Boxen plots of cell proximity in metastatic melanoma. Each box in a boxen plot represents a quantile range, progressively detailing the distribution from the median outwards to the extremes. Red boxes indicate cells positive for the given marker, blue indicate cells negative for the given marker, white indicate markers that were not used for selection. **g-h**, The distribution of the shortest distance between CD103+ (**g**) and CD103- (**h**) T$_{PEX}$ cells with other T cell subpopulations. **i**, The distribution of the shortest distance between tumour cells and T cell subpopulations. **j-k**, The distribution of the shortest distance between CD103+ (**j**) and CD103- (**k**) T$_{PEX}$ cells with subclasses of T$_{MEM}$ populations, with activity states indicated by the marker patterns on the left.

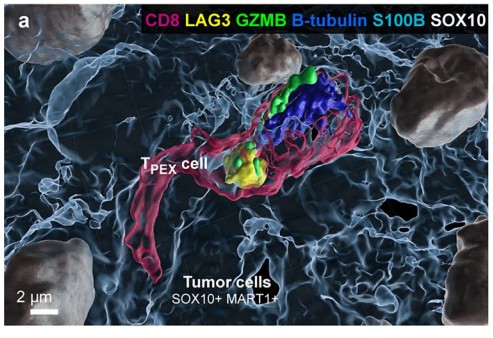

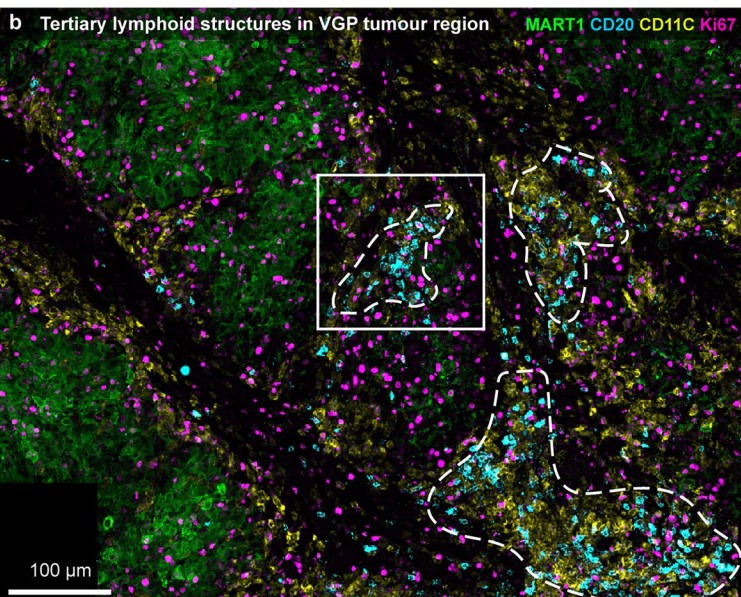

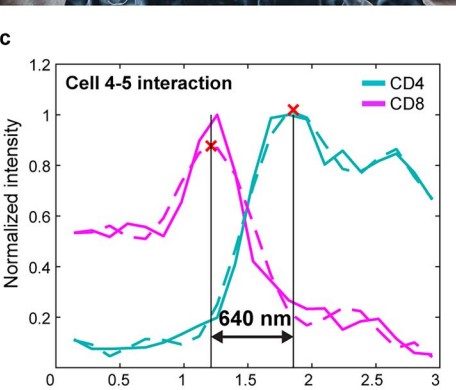

**Extended Data Fig. 7 | Cell-cell interactions and multivalent immune cell niches. a**, Surface rendering of Fig. 6e. **b**, Maximum projection of tertiary lymphoid structures in the VGP tumour region. Dashed lines demarcate colonies of CD20 + B cells (cyan) mixed with CD11c+ dendritic cells (yellow). White box indicates zoom in version found in Fig. 6l. Scale bar 100 μm. **c**, A l membrane intensity profile depicting CD4 (cyan) and CD8 (magenta) average expression across membranes of cells 4 (CD8 T cell) and 5 (CD4 T cell) from Fig. 6p. Solid lines represent raw data and dashed lines polynomial curve fitting. Red 'X's mark the maximum intensity along membrane intensity profiles and denote the midpoint of the cell membrane for each channel.

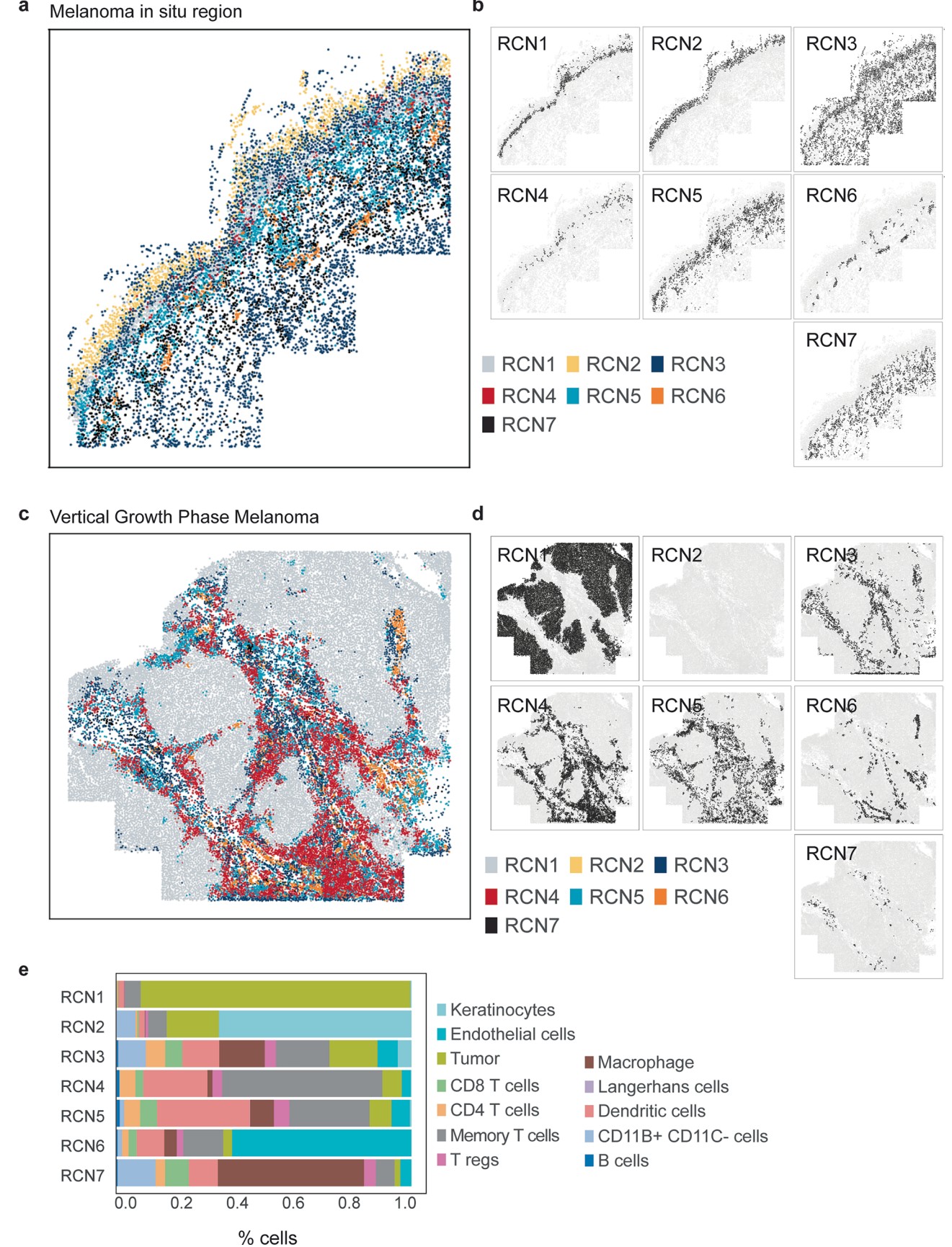

**Extended Data Fig. 8 | See next page for caption.**

**Extended Data Fig. 8 | Recurrent cellular neighbourhoods, identified by applying spatial latent Dirichlet allocation (LDA) to 2D projections of 3D data. a**, Scatter plot showing the MIS region. Cells are coloured based on their recurrent cellular neighbourhoods (RCN1–7). **b**, Scatter plot highlighting the distribution of each RCN in the MIS region of tissue. **c**, Scatter plot showing the invasive region of vertical growth phase melanoma. Cells are coloured based on the RCN to which they belong. **d**, Scatter plot highlighting the distribution of each RCN in the invasive region of tissue. **e**, Bar plot depicting the proportion of different cell-types within each RCN.

# Reporting Summary

## Statistics

For all statistical analyses, confirm that the following items are present in the figure legend, table legend, main text, or Methods section.

| n/a | Confirmed | |
|---|---|---|
| ☐ | ☒ | The exact sample size (*n*) for each experimental group/condition, given as a discrete number and unit of measurement |
| ☐ | ☒ | A statement on whether measurements were taken from distinct samples or whether the same sample was measured repeatedly |
| ☐ | ☒ | The statistical test(s) used AND whether they are one- or two-sided<br>*Only common tests should be described solely by name; describe more complex techniques in the Methods section.* |
| ☒ | ☐ | A description of all covariates tested |
| ☒ | ☐ | A description of any assumptions or corrections, such as tests of normality and adjustment for multiple comparisons |
| ☐ | ☒ | A full description of the statistical parameters including central tendency (e.g. means) or other basic estimates (e.g. regression coefficient) AND variation (e.g. standard deviation) or associated estimates of uncertainty (e.g. confidence intervals) |
| ☒ | ☐ | For null hypothesis testing, the test statistic (e.g. *F*, *t*, *r*) with confidence intervals, effect sizes, degrees of freedom and *P* value noted<br>*Give P values as exact values whenever suitable.* |
| ☒ | ☐ | For Bayesian analysis, information on the choice of priors and Markov chain Monte Carlo settings |
| ☒ | ☐ | For hierarchical and complex designs, identification of the appropriate level for tests and full reporting of outcomes |
| ☒ | ☐ | Estimates of effect sizes (e.g. Cohen's *d*, Pearson's *r*), indicating how they were calculated |

*Our web collection on statistics for biologists contains articles on many of the points above.*

## Software and code

Policy information about availability of computer code

| Data collection | Images acquired with Zeiss ZEN 3.7 with LSM Plus Processing. |
|---|---|
| Data analysis | Image registration and data analysis was performed in Mathworks MATLAB 2021b with Image Processing Toolbox, Curve Fitting Toolbox, Statistics and Machine Learning toolbox, and parallel computing toolbox, Python v3.9 and SCIMAP 1.3.1 (scimap.xyz; https://github.com/labsyspharm/scimap). Cell segmentation performed in Cellpose (https://github.com/MouseLand/cellpose). 3D visualization done in Bitplane Imaris 10.0. |

For manuscripts utilizing custom algorithms or software that are central to the research but not yet described in published literature, software must be made available to editors and reviewers. We strongly encourage code deposition in a community repository (e.g. GitHub). See the Nature Portfolio guidelines for submitting code & software for further information.

## Data

Policy information about availability of data

All manuscripts must include a data availability statement. This statement should provide the following information, where applicable:
- Accession codes, unique identifiers, or web links for publicly available datasets
- A description of any restrictions on data availability
- For clinical datasets or third party data, please ensure that the statement adheres to our policy

Data Availability (At time of publication)

Code and working demo can be found at https://github.com/labsyspharm/mel-3d-mis. All primary images and derived data (~5 TB) will available via AWS transfer at the time of publication. Instructions for accessing the primary and derived data is available via a data index page on Zenodo (doi.org/10.5281/zenodo.10055593). These images can be viewed using the free Imaris viewer (https://imaris.oxinst.com/imaris-viewer). 2D maximum projections of each dataset can be viewed in the MINERVA viewer (no download required), please see Supplementary Table 1 for links. A subset of data will be available for 3D interactive viewing within the browser-based tool Vitessce (http://vitessce.io/). This effort is a work in progress and will be available in the future.

# Research involving human participants, their data, or biological material

Policy information about studies with human participants or human data. See also policy information about sex, gender (identity/presentation), and sexual orientation and race, ethnicity and racism.

| | |
|---|---|
| Reporting on sex and gender | This study is not human subjects research. Supplementary Table 1 lists patient demographics for the 3 melanoma patients. |
| Reporting on race, ethnicity, or other socially relevant groupings | Listed in Supplementary Table 1. |
| Population characteristics | N/A |
| Recruitment | N/A |
| Ethics oversight | Specimens were retrieved from the archives of the Department of Pathology at Brigham and Women's Hospital and collected under Institutional Review Board approval (FWA00007071, Protocol IRB18-1363) under a waiver of consent. |

Note that full information on the approval of the study protocol must also be provided in the manuscript.

# Field-specific reporting

Please select the one below that is the best fit for your research. If you are not sure, read the appropriate sections before making your selection.

☒ Life sciences ☐ Behavioural & social sciences ☐ Ecological, evolutionary & environmental sciences

For a reference copy of the document with all sections, see nature.com/documents/nr-reporting-summary-flat.pdf

# Life sciences study design

All studies must disclose on these points even when the disclosure is negative.

| | |
|---|---|
| Sample size | Dataset 1 and 2 are serial adjacent sections from the same patient. All measurements were taken from regions of interests from unique patient-derived samples, which limits large sample size. Regions encompass the majority of the lesion as identified by board-certified pathologist. |
| Data exclusions | Images of antibody stains that did not stain/wash/register properly were excluded from analysis but are still included in imaris files for viewing and labeled as such. |
| Replication | Dataset 1 and 2 are serial adjacent sections from the same patient. Measurements were taken from unique patient-derived samples, and therefore, the possibility of running replication studies is very limited. |
| Randomization | N/A |
| Blinding | Conventional blinding is not relevant to this type of retrospective non-interventional study. Details of the model development are described in detail in the text and methods. |

# Reporting for specific materials, systems and methods

We require information from authors about some types of materials, experimental systems and methods used in many studies. Here, indicate whether each material, system or method listed is relevant to your study. If you are not sure if a list item applies to your research, read the appropriate section before selecting a response.

## Materials & experimental systems

| n/a | Involved in the study |
|---|---|
| ☐ | ☒ Antibodies |
| ☒ | ☐ Eukaryotic cell lines |
| ☒ | ☐ Palaeontology and archaeology |
| ☒ | ☐ Animals and other organisms |
| ☒ | ☐ Clinical data |
| ☒ | ☐ Dual use research of concern |
| ☒ | ☐ Plants |

## Methods

| n/a | Involved in the study |
|---|---|
| ☒ | ☐ ChIP-seq |
| ☒ | ☐ Flow cytometry |
| ☒ | ☐ MRI-based neuroimaging |

# Antibodies

| | |
|---|---|
| Antibodies used | Clone information and RRID of primary and secondary antibodies used in the manuscript can be found in Supplementary Table 3-10. |

1 , Target: Alpha-actin-2 , Label: Alexa Fluor 750 , R&D Systems , Cat: IC1420S-025 , Clone: 1A4 , RRID: AB_2868436
1 , Target: CD11b , Label: Alexa Fluor 488 , Thermo Fisher Scientific (eBioscience) , Cat: 53-0196-80, 53-0196-82 , Clone: C67F154 , RRID: AB_2637195
1 , Target: CD3E , Label: Alexa Fluor 488 , Cell Signaling Technology , Cat: 86936BC , Clone: D7A6E , RRID:
1 , Target: CD45 , Label: Alexa Fluor 647 , BioLegend , Cat: 304020, 304056 , Clone: HI30 , RRID: AB_493034
1 , Target: CD66b , Label: phycoerythrin , BioLegend , Cat: 392903 , Clone: 6/40c , RRID: AB_2750201
1 , Target: COX4 , Label: Alexa Fluor 555 , Abcam , Cat: ab210675 , Clone: EPR9442(ABC) , RRID: AB_2857975
1 , Target: CPT1A , Label: Alexa Fluor 488 , Abcam , Cat: ab171449 , Clone: 8F6AE9 , RRID: AB_2714024
1 , Target: Cytokeratin (pan) , Label: Alexa Fluor 750 , Novus Biologicals , Cat: NBP2-33200AF750 , Clone: AE-1/AE-3 , RRID: AB_2868569
1 , Target: ECP , Label:  , Abcam , Cat: ab207429 , Clone: EPR20357 , RRID: AB_2943114
1 , Target: Gamma Tubulin , Label: Alexa Fluor 647 , Abcam , Cat: ab191114 , Clone: TU-30 , RRID: AB_2889219
1 , Target: Goat IgG , Label: Alexa Fluor 555 , Thermo Fisher Scientific , Cat: A-21432 , Clone:  , RRID: AB_2535853
1 , Target: Langerin , Label:  , R&D Systems , Cat: AF2088-SP , Clone:  , RRID: AB_355143
1 , Target: Mouse IgG , Label: Alexa Fluor 647 , Thermo Fisher Scientific , Cat: A-21237 , Clone:  , RRID: AB_2535806
1 , Target: MPO , Label: Alexa Fluor 647 , Santa Cruz Biotechnology , Cat: sc-365436-AF647 , Clone: A-5 , RRID: AB_2943296
1 , Target: PCNA , Label: Alexa Fluor 750 , Cell Signaling Technology , Cat: 24114BC , Clone: PC10 , RRID:
1 , Target: PD-L1 , Label: Alexa Fluor 647 , Cell Signaling Technology , Cat: 62813BC , Clone: E1L3N , RRID:
1 , Target: phospho-Histone 3 , Label: Alexa Fluor 750 , Cell Signaling Technology , Cat: 43185BC , Clone: D2C8 , RRID:
1 , Target: Rabbit IgG , Label: Alexa Fluor 488 , Invitrogen , Cat: A-11070 , Clone:  , RRID: AB_2534114
1 , Target: S100A1 , Label:  , Abcam , Cat: ab183979 , Clone: EPR19013 , RRID: AB_2894716
1 , Target: S6 (Ser235/236) , Label: Alexa Fluor 750 , Cell Signaling Technology , Cat: 62788BC , Clone: D57.2.2E , RRID:
1 , Target: SOX10 , Label:  , Abcam , Cat: ab216020 , Clone: SOX10/1074 , RRID: AB_2847913
1 , Target: Stat1 (pY701) , Label: Alexa Fluor 555 , Cell Signaling Technology , Cat: 8183S , Clone: 58D6 , RRID: AB_10860600
1 , Target: TCF1/TCF7 , Label: Alexa Fluor 488 , Cell Signaling Technology , Cat: 6444 , Clone: C63D9 , RRID: AB_2797627
1 , Target: Vinculin , Label: eFluor 570 , Thermo Fisher Scientific , Cat: 41-9777-80 , Clone: 7F9 , RRID: AB_2573646
2 , Target: 5'-HMC , Label:  , Active Motif , Cat: 39769 , Clone:  , RRID: AB_10013602
2 , Target: Actin, cytoplasmic 1 , Label: Alexa Fluor 555 , Cell Signaling Technology , Cat: 8046S , Clone: 13E5 , RRID: AB_11179208
2 , Target: BANF1 , Label: Alexa Fluor 568 , Abcam , Cat: ab208534 , Clone: EPR7668 , RRID: AB_2868492
2 , Target: Catalase , Label: Alexa Fluor 488 , Abcam , Cat: ab185041 , Clone: EP1929Y , RRID: AB_2884892
2 , Target: CD103 , Label: Alexa Fluor 647 , BioLegend , Cat: 350209 , Clone: Ber-ACT8 , RRID: AB_10640870
2 , Target: CD15 , Label: Alexa Fluor 488 , BioLegend , Cat: 301910 , Clone: HI98 , RRID: AB_493257
2 , Target: CD8a , Label: eFluor 660 , eBioscience , Cat: 50-0008-82 , Clone: AMC908 , RRID: AB_2574149
2 , Target: COX4-1 , Label: Alexa Fluor 647 , Cell Signaling Technology , Cat: 7561S , Clone: 3E11 , RRID: AB_10994876
2 , Target: Cyclin-D1 , Label: Alexa Fluor 488 , Abcam , Cat: AB190194 , Clone: EPR2241 , RRID: AB_2728784
2 , Target: Cytokeratin (pan) , Label: eFluor 570 , Thermo Fisher Scientific (eBioscience) , Cat: 41-9003-80, 41-9003-82 , Clone: AE1/AE3 , RRID: AB_11218704
2 , Target: E-cadherin , Label: Alexa Fluor 555 , Cell Signaling Technology , Cat: 4295 , Clone: 24E10 , RRID: AB_2728822
2 , Target: H2AX , Label: Alexa Fluor 488 , BioLegend , Cat: 613406 , Clone:  2F3 , RRID: AB_2248011
2 , Target: Histone H3-Lys27-Trimethyl , Label: Alexa Fluor 750 , Cell Signaling Technology , Cat: 98316 , Clone: C36B11 , RRID: AB_2943245
2 , Target: Ki-67 , Label: Alexa Fluor 488 , Cell Signaling Technology , Cat: 11882S , Clone: D3B5 , RRID: AB_2687824
2 , Target: Lysozyme C , Label: Alexa Fluor 790 , Santa Cruz Biotechnology , Cat: sc-518012 AF790 , Clone: E-5 , RRID: AB_2943318
2 , Target: Mast Cell Tryptase , Label: Alexa Fluor 790 , Santa Cruz Biotechnology , Cat: sc-59587 AF790 , Clone: AA1 , RRID: AB_2943323
2 , Target: MITF , Label: Alexa Fluor 488 , Abcam , Cat: ab201675 , Clone: D5 , RRID: AB_2728787
2 , Target: N/A , Label: Alexa Fluor 488 , Invitrogen , Cat: A-11070 , Clone:  , RRID: AB_2534114
2 , Target: Nestin , Label: eFluor 570 , Thermo Fisher Scientific , Cat: 41-9843-80 , Clone: 10C2 , RRID: AB_2573652
2 , Target: PD-L1 , Label: Alexa Fluor 647 , Cell Signaling Technology , Cat: 15005 , Clone: E1L3N , RRID: AB_2728832
2 , Target: PD1 , Label: phycoerythrin , Cell Signaling Technology , Cat: 60333S , Clone: D4W2J , RRID: AB_2943233
2 , Target: PDPN , Label: Alexa Fluor 647 , BioLegend , Cat: 916610 , Clone:  , RRID: AB_2810816
2 , Target: pMLC2 , Label: Alexa Fluor 488 , Cell Signaling Technology , Cat: 35145BC , Clone: E2J8F , RRID:
2 , Target: PRAME , Label: Alexa Fluor 488 , Cell Signaling Technology , Cat: 39509 , Clone: E7I1B , RRID: AB_2943228
2 , Target: S100B , Label: Alexa Fluor 555 , Abcam , Cat: ab274881 , Clone: EP1576Y , RRID: AB_2890062
2 , Target: SOX10 , Label: Alexa Fluor 647 , Abcam , Cat: ab270151 , Clone: SP267 , RRID: AB_2927700
1&2 , Target: Catenin beta-1 , Label: Alexa Fluor 488 , Cell Signaling Technology , Cat: 2849 , Clone: L54E2 , RRID: AB_10693296

1&2 , Target: CD11b , Label: Alexa Fluor 647 , Abcam , Cat: ab204471 , Clone: EPR1344 , RRID: AB_204471
1&2 , Target: CD11c , Label: Alexa Fluor 555 , Cell Signaling Technology , Cat: 77882BC , Clone: D3V1E , RRID:
1&2 , Target: CD163 , Label: Alexa Fluor 555 , Abcam , Cat: ab281746 , Clone: EPR19518 , RRID: AB_2940922
1&2 , Target: CD20 , Label: Alexa Fluor 488 , eBioscience , Cat: 53-0202-80, 53-0202-82 , Clone: L26 , RRID: AB_10734357
1&2 , Target: CD206 , Label: Alexa Fluor 555 , Cell Signaling Technology , Cat: 48352BC , Clone: E2L9N , RRID:
1&2 , Target: CD31 , Label: Alexa Fluor 647 , Abcam , Cat: ab218582 , Clone: EPR3094 , RRID: AB_2857973
1&2 , Target: CD3E , Label: Alexa Fluor 555 , Cell Signaling Technology , Cat: 57869BC , Clone: D7A6E , RRID:
1&2 , Target: CD4 , Label: Alexa Fluor 488 , R&D Systems , Cat: FAB8165G , Clone:  , RRID: AB_2728839
1&2 , Target: CD8a , Label: eFluor 660 , eBioscience , Cat: 50-0008-82 , Clone: AMC908 , RRID: AB_2574149
1&2 , Target: FOXP3 , Label: eFluor 570 , eBioscience , Cat: 41-4777-80, 41-4777-82 , Clone: 236A/E7 , RRID: AB_2573609
1&2 , Target: Granzyme B , Label: Alexa Fluor 647 , Santa Cruz Biotechnology , Cat: sc-8022 AF647 , Clone: 2C5 , RRID: AB_2232723
1&2 , Target: HLA-A and HLA-B , Label: Alexa Fluor 488 , Abcam , Cat: ab198376 , Clone: EPR1394Y , RRID: AB_2943099
1&2 , Target: HLA-DPB1 , Label: Alexa Fluor 647 , abcam , Cat: ab201347 , Clone: EPR11226 , RRID: AB_2861375
1&2 , Target: Hoechst 33342 , Label:  , Life Technologies , Cat: H3570 , Clone:  , RRID: AB_2651135
1&2 , Target: IRF1 , Label: Alexa Fluor 647 , Cell Signaling Technology , Cat: 14105 , Clone: D5E4 , RRID: AB_2798393
1&2 , Target: LAG3 , Label: Alexa Fluor 555 , Cell Signaling Technology , Cat: 56141BC , Clone: D2G4O™ , RRID: AB_2798739
1&2 , Target: Lamin-A/C , Label: Alexa Fluor 488 , Cell Signaling Technology , Cat: 8617S , Clone: 4C11 , RRID: AB_10997529
1&2 , Target: Lamin-B1 , Label: Alexa Fluor 488 , Abcam , Cat: ab194106 , Clone: EPR8985(B) , RRID: AB_2728786
1&2 , Target: MART-1 , Label:  , Abcam , Cat: ab210546 , Clone: EPR20380 , RRID: AB_2889292
1&2 , Target: MX1 , Label: Alexa Fluor 488 , Cell Signaling Technology , Cat: 7937BC , Clone: D3W7I , RRID: AB_2799122
1&2 , Target: Neurofilament L , Label: Alexa Fluor 488 , Cell Signaling Technology , Cat: 8024 , Clone: C28E10 , RRID: AB_10860421
1&2 , Target: PD-1 , Label: Alexa Fluor 647 , Abcam , Cat: ab201825 , Clone: EPR4877(2) , RRID: AB_2728811
1&2 , Target: PMEL , Label: phycoerythrin , Abcam , Cat: ab246731 , Clone: EP4863(2) , RRID: AB_2890052
1&2 , Target: S100 alpha , Label: Alexa Fluor 488 , Abcam , Cat: ab207367 , Clone: EPR5251 , RRID: AB_2728788
1&2 , Target: SOX9 , Label: Alexa Fluor 488 , Abcam , Cat: ab196450 , Clone: EPR14335 , RRID: AB_2665383
1&2 , Target: Tubulin beta chain , Label: Alexa Fluor 555 , Cell Signaling Technology , Cat: 2116S , Clone: 9F3 , RRID: AB_10695881
1&2 , Target: Vimentin , Label: Alexa Fluor 750 , Cell Signaling Technology , Cat: 69227BC , Clone: D21H3 , RRID:

Validation

Each antibody was tested on clinical discard and tonsil samples to verify expected and unexpected staining patterns in positive and negative controls. Staining patterns were visually certified by Board Certified Pathologists.

# Plants

Seed stocks

N/A

Novel plant genotypes

N/A

Authentication

N/A

