## [Peer Review File · Nature Methods]

Highly Multiplexed 3D Profiling of Cell States and Immune Niches in Human Tumours

Corresponding Author: Professor Peter Sorger

This manuscript has been previously reviewed at another journal that is not operating a transparent peer review scheme. The manuscript was considered suitable for publication without further review at Nature Methods.

Version 1:

Reviewer comments:

Reviewer #1

(Remarks to the Author)

(Remarks on code availability)

Reviewer #2

(Remarks to the Author)

I think the authors did a great job to answer all of the major concerns from the initial review in Nature. I appreciate they included a lot more information on the methodic details and shared protocols and I sincerely think that Nature Methods is the right home for the paper.

I do think that even seemingly simple and rather technical figures like Supp. Figures 12-22 are helpful for readers to understand potential limitations and know what to watch out for to perform these kind of experiments. The level of complexity in creating well-vetted panels for 3D multiplexed imaging is very high, yet is often underestimated by many scientists who are just starting with multiplexed antibody stainings or rely on commercial platforms/products without performing important technical controls and optimization. I hope seeing the hurdles documented in the papers provides a more critical thinking perspective to the readers and users of these methods. The last set of supplementary figures are great to evaluate the affect of the imaging settings on the analyses and to get a sense what is adequate and what is overkill for particular questions.

I have just two minor notes:

The new observation and the associated quantification of the tissue shrinkage/expansion due to the hydration conditions are interesting. I was just curious if the observed change is isotropic or is just happening in the z dimension because the tissue is sticking to the support at the bottom at the time of hydration (or could be happening unevenly from the bottom to the top, where the tissue would expand not into a rectangle prism but to a trapezoid one). The reason I was asking is because I do not think that this is typically taken into account by most when doing registration of H&E stained sections with multiplexed IF or imaging-based spatial transcriptomics assays, but the effect might be minor laterally on thin sections. If the authors have relevant observations or data/quantifications, some comments on this would be useful for that readers that also perform other types of tissue assays.

For the alternative mounting with the matrigel, can the authors clarify - is the top coverslip fixed in place somehow (if so how?) and is this then manually displaced to do the washes and put back on each imaging cycle?

(Remarks on code availability)

I have browsed through the deposited code, but was not able to do an extensive review to see how it runs as I do not have MATLAB.

As far as I can see the readme only covers the cell interaction analysis and demo. A bit more documentation/instructions on

the other parts could be helpful (some useful information is fortunately provided in the Methods section).

RESPONSE TO REVIEW 4/7/2025

NMETH-A57303-T

Highly Multiplexed 3D Profiling of Cell States and Immune Niches in Human Tumours
Yapp... Sorger et al.

OVERVIEW OF THE RESPONSE

We thank the reviewers for their thorough and detailed review of our manuscript. We believe that we have addressed all their concerns through revisions to the text, additional experiments and analysis, and the inclusion of expensive new data. To fit this new data we have removed several results relating to actin biology and the cytoskeleton; we do not believe that these are central to the story. We provide an overview of our response below, followed by a detailed point-by-point response.

1. New results and methods. As we worked through the quantitative analysis requested by the reviewers, we made several unexpected “discoveries” about tissue imaging that resulted in major improvements to the approach. Most notable among these is the fact that tissue sections processed for immunofluorescence expand ~ 1.5 fold in thickness when subjected to antigen retrieval and antibody staining. This hydration-dependent expansion is consistent with earlier literature, but to our knowledge, has not been quantified and incorporated into models of 3D tissue architecture. We have added to the manuscript a series of experiments that argue that hydrated tissue dimensions, and not the dimensions encountered in standard H&E images (which involve dehydrated specimens), are representative of actual tissue, at least at the level of cells and cell communities.

We have also undertaken a major effort to make thick section multiplexing more reliable across tissue types (as requested by reviewer 3). This has led to the development of two different approaches to imaging: a conventional one in which tissue is mounted to glass slides and an alternative one in which tissue is mounted to coverslips and overlaid with adhesive substrates (Matrigel) and physical grids to decrease tissue damage during cycling. We now demonstrate that these methods work well across a wide range of tissues, increasing the usability of our approach. All of these updates and new results are described in Figure 1 and Extended Figure 1.

2. Relationship to (and citation of) earlier work. Reviewer 1 pointed out a number of citations that we had missed in our first draft. We sincerely apologise, particularly since several of the missed manuscripts come from one of the leaders in multiplex imaging (the Germain group at NIH). We have corrected these omissions. At the same time, we submit that the reviewer has not fully recognized the difference between working with specimens from model organisms that can be frozen, or alcohol fixed and human specimens, which are almost always formaldehyde fixed and paraffin embedded (FFPE). The methods developed for tissues from model organisms in a research setting cannot be transferred directly to imaging FFPE specimens. This is an important aspect of our work that we underemphasized in the original draft.

In addition, the reviewer describes as standard, microscopes and imaging conditions (e.g. 1.4NA oil immersion objectives) that are commonly used by cell biologists working with cultured cells (and animal models); but are very different from the high-speed scanning microscopes used for tissue profiling in all of the recently popularised methods such as MIBI, IMC, MxIF, CODEX (Akoya), COMET (Lunaphore) and CyCIF. These latter approaches use lower resolution optics and single

optical sections because the specimens are large, and the emphasis is on data collection efficiency: a whole-slide tissue image commonly comprises over 1,000 conventional fields of view (e.g. collected with a 0.5 to 0.75 NA dry (air) objectives). The resolution we achieve in this paper, and thus the ability to discriminate very fine features and cell membranes, is unprecedented for high-plex images of human specimens using either thick or thin sections.

3. 2D vs 3D Imaging. Reviewer 1 makes the assertion that our data do not constitute true 3D imaging since the “3D” label only applies when specimens are thicker or imaged using fluorescence light sheet microscopy (including methods described in Germain lab papers). This does not seem to us a defensible position: a 2D image has a single optical plane and a 3D image has multiple planes (in the case of our images, >120). The size of the specimen, and whether light sheet or confocal methods are used to acquire the data, is an entirely separate issue. Moreover, we believe one of the most interesting findings of our study is that after 75 years of immunofluorescence imaging of 5 μ m tissue sections, few if any intact cells have been examined.

We have nonetheless written an extended supplementary note on different approaches to 2D and 3D optical microscopy that we believe fully addresses the concerns of reviewer 1. We have also modified Figure 1 to better illustrate key features of 2D v 3D imaging. These will be familiar to many experienced microscopists (like Reviewer 1) but we believe that the succinct “explanatory” text in the body of the manuscript is important for a spatial proteomics community that thinks less about imaging technology.

Reviewer 2 takes an opposite approach from Reviewer 1 and suggests that we examine the implications of 2D v 3D imaging more carefully and better quantify the impact of using thin tissue sections. This is an excellent idea that we have thoroughly explored, as described below.

4. Organization of the manuscript. Both Reviewer 1 and 2 were concerned that our manuscript did not conform to the expected layout of a *Nature* manuscript. They correctly point out that our manuscript has a substantial methodological component followed by a series of illustrative findings that can be made using high resolution 3D images. They suggest that a more focused look at one of these topics with more specimens, or additional tissue types would make for a stronger “science” story. We have indeed better focused the current study and we have modified the formatting to fit the *Nature Methods* style. .

5. Request for additional detail of the method. Both reviewers ask for additional details with respect to the performance of the method. We have therefore prepared a detailed protocol for the on-line methods and in a protocols.io description. We have also built a web resource for the data and methods and we will keep this continuously updated as methods improve and additional reagents or tissue types are tested.

DETAILED RESPONSE TO REVIEW

REFEREE 1

(Remarks to the Author)

In this submission from Yapp et al. entitled “Multiplexed 3D Analysis of Immune States and Niches in Human Tissue” the authors use 35 μ M thick sections of human melanomas to conduct what they

term a “3D” analysis. This follows on from prior work of the Sorger lab involving 3D imaging using serial thin sections. The central claim in the present manuscript is that by extending the thickness of a section to 30-40 μM , it is possible to capture entire cells rather than cutting through most cells as would be the case for the more typical sections of 5-10 μM used for multiplex analysis. With this more complete image of intact cells, they conduct an analysis of the phenotype, surface protein distribution, intracellular organelle content and structure, and spatial relationships within the TME. As would be expected from two senior investigators with many key contributions to the generation and detailed analysis of optical imaging data (Sorger and Danuser), this study is replete with the use of advanced algorithms for image dissection and the rendered surface interpretations of the primary immunofluorescence (IF) data are elegant. Beyond the imaging itself, the authors draw a series of conclusions about the distribution of immune subsets and cancer cells in two different types of melanoma, as well as providing interpretations of the signaling events occurring in cells based on the patterns of surface protein distribution as well as the distance between opposing membranes.

Evaluation of this manuscript is difficult, as it represents a combination of biological claims and technical developments. There is substantial value in the technical details of how the authors both collected the primary data and in how it has been processed to provide multiscale information ranging from intracellular organization to local interactions, to overall representation of cell types within the TME. At the same time, many of the biological interpretations are not supported by adequate replicate experiments, either from the same tumor sample or from additional patient material. In addition, a number of the interpretations are of concern, several of the novelty claims on the technical side fail to properly credit published work from other groups, and some of the discussion in the supplement about achieving ‘real’ multiplex 3D imaging is incorrect based on recent findings from other laboratories.

As described in the overview above, we believe that our manuscript’s combination of “biological claims and technical developments” is a better fit to *Nature Methods*. We have extensively modified the manuscript throughout to address concerns raised in these two paragraphs regarding credit, biological interpretations, and novelty. These modifications are described more completely below in response to the reviewer’s numbered comments.

Technical issues:

1. The major premise of the current paper is that multiplex IF has not been applied to sections thicker than 5-10 μM and that such data even from these sections are at low resolution (“Nonetheless, almost all contemporary high-plex image-based profiling emphasizes relatively low resolution 2D imaging (0.6 to 2.0 μm lateral resolution) of conventional 5 μm thick tissue sections to increase the speed and convenience of data acquisition.). This is not accurate. Radtke et al. (Proc Natl Acad Sci U S A. 2020 Dec 29;117(52):33455-33465; Nat Protoc. 2022 Feb;17(2):378-401) have reported high multiplex IF using a method called IBEX with sections from diverse mouse and human tissues sectioned at 30 μM and imaged at a resolution comparable to that reported here (~250 x 250nm in XY). It is true that these investigators did not use the data for the deep study of individual cell features such as granule content or precise distribution of membrane proteins, but the imaging data needed for such analyses has been collected by others and the implication here that

this is a new approach in multiplex imaging is not really true. It should also be noted that many of the observations about cell-cell interactions, T cell contact with dendrites, morphology of invading tumor cells and so on, have been published in papers using 2-photon intravital imaging by multiple groups (Cahalan, von Andrian, Mempel, Friedl, Germain, Bousso, Krummel, and others). These are not high-plex images, being constrained by detector number to 4-6 parameters, but many of the observations made in static images here correspond to what is in the work from these other groups and it seems proper to credit such prior publications when discussing the present observations.

We thank the reviewer for drawing our attention to these papers and to a series of review articles by the same senior author; we have added references to the revised manuscript. We agree that we should have mentioned the IBEX method (and now do - prominently) but we do not agree that it substantially overlaps our paper. IBEX of course has been primarily applied to flash-frozen tissue, and only cursorily applied to FFPE tissues, which are more challenging to stain because of their low epitope abundance (as is mentioned in the Radtke 2020 paper). Additionally, we sample in higher plex, at finer resolution in both lateral and axial dimensions (2x and 3x more respectively), and at twice the bit depth. However, the IBEX method should have been mentioned, and we have corrected this error in our revised manuscript.

More significantly, we spend the great majority of the paper not on technical details, but rather on illustrating what can be achieved using high resolution, high sensitivity, 3D imaging. 2-photon intravital imaging is an important method, but it is restricted to 2-3 channels and performed exclusively in animal models. We do not believe that the current paper is the right place to review the totality of imaging methods applicable to tissue (although we make a stab at this in our new Supplementary Note), but we have tried to accommodate the reviewer's concern by adding a section on high-plex fixed cell and low-plex live cell intravital imaging to the conclusion. Finally, we submit that while the reviewer is correct that microscopists image tissues at a wide range of resolutions (particularly with model organisms), our statement that "*almost all contemporary high-plex [spatial proteomics] emphasizes [efficient but] relatively low resolution 2D imaging (0.6 to 2.0 μm lateral resolution*" remains true.

2. While I appreciate the difficulty of conducting >50 plex IF imaging with minimal noise and high reliability of all antibodies in all cycles, the current study only uses 3 sections of tissue for all the analyses. While this is adequate to show that they can imaging at 35 uM and obtain full cell shapes in such a data set, it is inadequate to provide quantitatively representative data even from the specific tumor samples giving rise to these images, no less as considering the data to be representative of the two melanoma disease states. Indeed, there are now a substantial number of high multiplex analyses of tumor tissue in the literature and these range from several dozen to several hundred samples each (see work from Angelo, Bodemiller, Nolan, and others). I think the authors should temper all their conclusions about tumor biology in the absence of further data; even better, they should conduct lower plex IF analysis of additional samples focused on their major conclusions about stem cells, invasion, cell interactions, and so on to see if their conclusions can be validated with these additional samples.

We respectfully disagree; comparison with Angelo, Bodemiller, and Nolan cannot be made due to dramatic difference in resolution and dimension of the specimens. Moreover, it is not true that the current study describes only 3 tissue sections; for simplicity, our analysis focuses on these tissues, but we provide many more in supplementary materials. The studies from Angelo, Bodemiller, Nolan, and others typically involve a similar number of 2D samples at 5-10 fold lower resolution, generating

datasets 100-1000 fold smaller. Imaging of “several hundred samples” as described by the reviewer, has only been achieved with tissue microarrays, which represent very small pieces of tissue (typically <1 mm in diameter). The authors cited by the reviewer work almost exclusively with TMAs, which creates the impression of a rich data set, despite severe limitations in spatial power when dealing with such small specimens (see DOI: 10.1016/j.cell.2022.12.028). Imaging hundreds of whole slide samples at high resolution on a laser scanning confocal is currently not feasible, but we are working to develop faster methods that balance throughput with resolution.

Overall, our manuscript includes 11 datasets, not only the three the reviewer identified. To reduce the complexity of our manuscript, we focus on 3 melanoma datasets for the body of the text. In addition to the existing overview of each dataset in the supplement, we have now also made a high-resolution 2-D projection of each dataset using Minerva (an online visualisation software), where the reviewer can browse our images. Importantly, Minerva will allow readers to view full-resolution images and toggle channels on/off as they please. Links to the Minerva story for each dataset are available in **Supplementary Table 1**. We think this adds a significant value to readers, who can now browse each dataset easily online at full resolution - without download; however, the data are also downloadable for able to manage such large image files.

3. With respect to the “3D” nature of the study, as they note, confocal imaging is inherently “3D” and even thin sections of 5-10uM contain ‘3D’ information on the contents of cells. 3D microscopic optical imaging has more conventionally been considered to be imaging at 100s of uM or even several mm dimensions that reveal the paths of tissue elements with ramifying distributions such as nerves, blood and lymphatic vessels, small airways, and ducts across sufficient tissue volume to understand their organization in space. The present work is far from such 3D analyses and it seem misleading to headline such a claim.

As described in the introduction to this response, there is no objective basis on which to make this claim. The 2D v 3D distinction has nothing to do with the thickness of the specimen. Although there are important observations to glean from macroscale 3D analysis, our work provides an unprecedented 3-dimensional view of cellular interactions within native tumour context, which we believe to be a substantial advance from the current state of the art. By way of illustration, 3D optical deconvolution and superresolution microscopy of cultured cells, which are rarely thicker than ~10 um, has transformed cell biology over the past two decades. We also submit that the reviewer is ignoring the fact that nearly all of the high-profile spatial profiling papers reported over the last five years involve a single optical plane. Our new supplementary note discusses the merits of LSFM vs. confocal microscopy. The reviewer might be interested to know that we have developed an approach to cyclic light sheet fluorescence microscopy (LSFM) of mm thick FFPE human tissues and can indeed visualize “*nerves, blood and lymphatic vessels, small airways, and ducts.*” Of note, we also remount and resection these thick tissues for 3D confocal imaging. We are therefore confident that LSFM is not a substitute for the high resolution imaging shown here.

The authors might argue that with respect to multiplex IF, they are extending the usual work-flow from sections that lack most intact cells at 5-10uM to whole cells in 35uM sections and they have a point here. However, even at 35 uM, they will be missing portions of larger cells such as macrophages, dendritic cells, and stromal cells that often extend over 40-70uM. In addition, while they do capture and show a portion of a blood vessel with cells within the lumen, there are no data on the general organization of vessels within and across the tumor sample and no visualization of nerves.

The reviewer is correct. At any specific spatial scale it is difficult to discern features that have a larger characteristic dimension. The branching of blood vessels and nerves is better discerned using light sheet (LSFM) than confocal microscopy. However, this is not the purpose of our paper. Instead, we focus on showing that all or nearly all cells are incomplete in a typical 5 um tissue section but that 35-40 um, >75% or more of all cells are intact (see new **Extended Data Figure 1i**). Some cell types will still be incomplete in 40 um sections, but the immune cells that are the focus of this paper are largely intact (excluding some dendritic cells). We carefully show why the switch to thick sections substantially impacts cell type calling, and the identification of intracellular structures. The reviewer does not mention the fact that LSFM is substantially (5-fold or more) lower in resolution than what we report, generally precluding a precise analysis of cell morphology (or processes such as extravasation - which we show for a blood vessel) and that it does not currently extend beyond 5 channels (more typically 3). In sum then, the reviewer's points are generally correct, but we submit that they are not germane to our manuscript.

Furthermore, the speculation in this paper that confocal imaging cannot be used for high multiplex IF of thick tissue that does provide true 3D data is incorrect. Li et al (Nat Protoc. 2019 Jun;14(6):1708-1733; Proc Natl Acad Sci U S A. 2017 Aug 29;114(35):E7321-E7330) showed an entire mouse LN in 5-6 parameters and that same laboratory has stated in a review (Immunol Rev 2022 Mar;306(1):8-24) that they have achieved > 15-20 plex in > 300uM samples using confocal imaging. While the latter claim in a review in no way pre-empts this present submission, especially as the actual data are not available for evaluation and comparison, it does seem the authors should be more nuanced in how they describe their present data sets and also take into consideration the work of others doing larger scale multiplex 3D imaging.

With respect to these concerns, let us first state that we should have cited the C3D paper from the Germain laboratory and have now mentioned it in the discussion. We also prominently cite the IBEX work from the same lab and sincerely apologize that we missed these citations in our initial draft.

However, there is a very substantial difference between a single imaging cycle (which gives rise to 5-6 channels in Li et al) on a non-embedded murine tissue and high-plex (>50-channel) imaging of a human FFPE tissue. The latter approach introduces a host of problems with image registration, antibody staining, etc.

Importantly, the high-plex tissue imaging methods that have driven interest in spatial profiling enable a completely different approach to understanding tissue and tumour biology. These methods make it possible to distinguish cell types from each other and simultaneously assay cell state and cell-cell interactions. We were not aware of the Germain & Radtke 2021 review article, but our views are in line with the author's discussion about the in-progress 3D-IBEX method and we too believe that our manuscript highlights the value of high-resolution, highly-multiplexed confocal imaging for clarifying rare immune subtypes and revealing novel interactions within the tumour microenvironment. There is no precise threshold that distinguishes high-plex from conventional imaging (or transcript profiling) but high-dimensional data analysis methods generally become most effective over 20 channels, which is greater than what is claimed by the review article.

Unfortunately, we cannot compare our method to 3D-IBEX given that it is unpublished. However, we tried to accommodate the reviewer's concern by citing a 2024 meeting abstract from the Germain lab on 3D-IBEX (*Ce3D-IBEX: Achieving Multiplex 3-dimensional Imaging for Deep Phenotyping of Cells in Tissues*),

4. The authors largely ignore the entire field of stereology (e.g., METHODS 18, 493–507 (1999)) that has for some time worked on how thin 2D images mis-represent true tissue content. Such work should be cited as it provides methods for correcting what is seen in the thin sections. Of course, stereology does NOT provide data on the cell contents as nicely reported here, but the thin section-thick section issue is a key one of concern to this long-standing field and seems worthwhile to make readers aware of this literature.

This is a good point, thank you for the suggestion. You are correct; our rationale in **Figure 1** illustrating the need for thick section confocal imaging for accurate phenotyping is directly in line with the observations of the stereology field. We have now added a brief section on stereology and suggested that our work should be integrated with previously developed methods for inferring 3D structures from 2D images.

5. The authors state "...cell segmentation methods rely on nuclei. Thus, accurate phenotyping of these three immune cells requires imaging whole cells to account for non-uniform protein distributions in cells and their overlap in Z." This is incorrect. In Li et al., Proc Natl Acad Sci U S A . 2017 Aug 29;114(35):E7321-E7330, the authors discuss segmentation in volumetric multiplex imaging and write "To circumvent these issues, we explored various image processing pipelines to accurately segment all imaged cells independent of nuclear staining.", followed by description of a pipeline for achieving such nucleus-free segmentation in 3D images.

Once again, we apologise for missing this paper from the Germain lab and specifically for missing the point that segmentation can be performed using by summing multiple membrane markers together and applying watershed in mouse lymph nodes. We have tried such an approach ourselves and found that it is generally difficult to find such markers in the complex setting of a human tissue. A quick examination of the 50 or so papers published in the last five years to tackle the problem of cell segmentation in high-plex spatial profiles all use nuclei as the basis for the first step in segmentation. To our knowledge, this includes all of the commercial packages that make up the bulk of the published data in the field.

More importantly, we submit that the reviewer misses the point of this part of our manuscript in which we show that what appears to be background signal in a thin section in fact represents foreground signal from a cell that lies above or below, but whose nucleus is not in the imaging plane (see **Figure 1d** and new quantification in **Figure 1e**). It seems highly improbable that any extant nuclei or *non-nuclei* based segmentation approach could accurately discern such signals in the complex environment of a human tissue or tumour. Whether to-be-developed machine learning approaches can address the issue remains to be determined, but for now we regard the non-assignable intensity to be a fundamental limitation of thin-section imaging.

6. The authors then discuss slide scanner data, but most academic laboratories employing multiplex imaging conduct their data collection at high resolution; 40x 1.3 NA objectives are typical along with a high pixel count per image to yield data of ~250 x 250uM lateral resolution. Clinical labs using slide scanners are not conducting highly multiplex imaging of the type discussed here, so these comments are off the mark.

With respect, this is simply not true. As mentioned above, conventional low-plex tissue imaging (of model organisms and tissue culture cells) has long used high NA objectives but all of the work that has recently been published using CODEX, IMC, MxIF, MIBI, CyCIF involves single optical planes acquired at resolutions closed to 600 nm laterally; in the case of MIBI and IMC, the resolution is as

coarse as 1-5 um per pixel, whereas our study samples at a lateral resolution of 0.14 microns per pixel.

We submit however, that the reviewer has missed the point in these comments. The issue is not who has the better microscope, or smaller pixels, but rather what can be done at higher resolutions using 3D reconstruction. We believe that we have convincingly shown that with high-plex, high-resolution 3D imaging it is possible to discern subcellular structures (including those that drive tumour phenotypes and tumour-immune interaction) and juxtacrine signalling complexes that have not previously been described in either model organisms or human specimens.

7. For Fig 4i-s, they discuss the combinatorial representation of the proteins they interrogate and state that all possible mixtures were observed. As stated, this implies that many of the molecules studied are absent from many of the cells. This is of course dependent on the signal to noise ratio in the image, the extent of expression, and the thresholding they apply to the imaging data. It is quite possible that cells they categorize as negative for a given protein have a low amount of the molecule(s) in question; there is an important biological difference between a complete lack of protein in a cell and a low level of that molecule, perhaps only at the particular moment of fixation of the sample. The authors should be more cautious in their claims about expression data of this type, especially when it is the lack of detection that is driving the interpretation.

The reviewer is of course correct. We have added a note in this section that points readers to our methods section for more details about how channels were gated (see '*Single Cell Phenotyping*'). As stated above, we have also made Z projections of all images available online (see links in **Supplementary Table 1**) and have added a new **Supplementary Note 1** (described below) to address the role of signal to noise.

All immunophenotyping, including standard methods performed using flow cytometry, rely on positive and negative gating. Immune cells in particular are characterised based on a set of binary criteria such as CD3-positive, CD8-positive, CD4-negative, PD1-positive etc. Thus, the reviewer's comments pertain to the entire field of immunology. Also relevant to this discussion is the fact that high-resolution imaging has substantially better signal to noise than low resolution imaging (and all extant approaches to high-plex spatial profiling) and that punctate signals can be much more accurately quantified. Thus, gating problems are much less severe in the images we acquire than in most other methods.

8. They describe poor antibody penetration in S fig 15 and speculate about the origin of the problem. It is worth noting that they are seeing this problem with highly abundant targets such as SMA. Others doing volume imaging at varying specimen thickness have reported such a problem and it can be ascribed in at least some cases to rapid depletion of free antibody by the targets in the more superficial regions of the sample, leaving too little antibody to give adequate staining in deeper areas. This gradient effect is enhanced by the high intensity of the surface layer staining that leads the imager to attenuate the laser power to avoid signal saturation, thus reducing signal from whatever antibody actually makes it to deeper regions. There can be other explanations of course, including the fluorochrome issue discussed, but this removal of antibody from the staining mix is worth considering, especially if the antibodies are pre-titrated and diluted according to thin section controls.

We thank the reviewer for raising this point (which is also raised by reviewer 2) and agree completely. We have added this to the methods section, “ Optimization of sample thickness and antibody staining protocol.” We have also added a new **Supplementary Figure 14** that demonstrates how antibodies penetrate the tissue over time. As the reviewer may also have experienced, it is possible to improve penetration using various denaturing reagents. We are exploring these reagents and anticipate that it will be possible to use an increasingly wide range of antibodies with thick tissue sections.

9. In Fig 4o, I am confused – the magenta label says PRAME, but the cells in magenta largely correspond to those with SOX9 and SOX10 staining in panel m yet they do not appear in the conjoint green color assigned to these proteins in panel o. Perhaps I misunderstand how they have processed the data since some cells have all 3 markers.

We thank the reviewer for pointing this out. In retrospect, we realise that our labelling convention was confusing. The magenta is indeed PRAME and is consistent with the PRAME channel in 4i and 4k. However, the green label indicates cells dually positive for SOX10 and SOX9. Likewise, the data in 4i and 4k indicate cells that are positive for only SOX9 or SOX10. We apologise for the confusion and have clarified this in the label on 4i, 4k, 4m, 4o, and 4p, and the corresponding figure legends.

Biology:

1. They describe Langerhans cells as “...the dendritic cells of skin”. This is no longer considered to be true. These are embryologically derived macrophages – see Nat Rev Immunol 2014 Jun;14(6):392-404.

This was sloppy writing on our part. We have corrected this and thank the reviewer for pointing out this error.

2. In discussing the data in Figure 3, they make claims or suggestions about the direction of migration of T cells based on static data. In truth, the imaged cells could have been moving in any of many different directions within the TME and have just been caught in one place and one polarized orientation at the moment of fixation. Live imaging shows such diverse directions of movement of T cells within inflamed tissues and to a lesser degree tumors. I urge the authors to be more cautious. As to the points about tension, the tumor core, and the margins, the section is oriented in only one direction across the full tumor volume and hence, it is unknown if these patterns would be seen in the orthogonal extremes of the tumor; further, this is a single sample from this particular tumor, so how representative the features are cannot be evaluated. Finally, details of tumor invasion have been visualized by Friedl and colleagues and should be cited (one recent review is Trends Cell Biol 2023 May;33(5):388-402; technical matters for such imaging are in Elife. 2022 Feb 15;11:e63776).

The reviewer is objecting to our use of migration to describe lymphocytes in tumour (as in “migrated into the tumour core”) when we should have used a more neutral and passive term such as “were infiltrated within.” We did not mean to imply that we could discern knowledge about migration patterns from a single image.

We have elected to remove these panels from Figure 3 in order to make room for more methodological detail and meet the word count limit of Nature Methods.

3. In the section on inflammatory neighborhoods, they describe responses to what they infer is local interferon production. Such measurements have been made in ref. 47, noted later in the text, but also in Nat Cancer 2020 Mar;1(3):302-314 and Cancer 2023 Jul;4(7):968-983 among other studies from Bousso and colleagues.

Thank you for the suggestion. We were aware of these papers and have collaborated directly with these authors; we have now added references to the Oyler-Yaniv paper (prior ref. 47) and Nat Cancer 2020 Mar;1(3):302-314 at this point in the text.

4. In the next section on T cell subsets, they conclude based on staining that there is substantial overlap at least in phenotype among Tpex, Tmem, and Teff CD8 T cells and that this is different from what is generally believed, but such overlapping marker protein expression is commonly seen and noted in scRNAseq data and especially flow cytometry in both tumor studies and the classic clone 13 LCMV model.

This is an interesting point, and we have revised our text and added some references. The clone 13 LCMV model is of course an acute infection model, and we agree with the general thrust of the reviewer's comments here and below that tumour biologists are insufficiently aware of some of what is known about infection. More specifically, we have cited <https://www.nature.com/articles/s41590-020-0760-z> in this regard. Of note however, this paper and the others we have found on the LCMV model use flow cytometry and dissociative scRNAseq, not spatial immunoprofiling. Regardless, the reviewer is correct that we should have described these data without making any claim to novelty.

5. They state "Tpex, Tmem, and Teff cells were also highly proliferative (40-80% PCNA or Ki67 positive) demonstrating a high degree of activation and self-renewal." This is an over-interpretation of the data. Other work has shown that the cells they define in these categories can/do have a lineage relationship and hence, if the Tpex divide, their progeny will have the PCNA or Ki67 staining they see even if these progeny cells are not continuing to divide. Suggesting that Teff have a capacity for self-renewal also runs against many observations in the literature that show these Teff to behave as terminally differentiated cells. Better phrasing and a more cautious interpretation of the data are called for here.

This is an interesting point and we have revised the text to state that cells expressing PCNA or Ki67 may have recently divided (as opposed to being about to divide). Of course, we agree entirely with the reviewer that the presence of a proliferation marker in a cell does not imply that it is poised for long-term proliferation as part of a self-renewing immune compartment. However, work has shown that Tpex can be highly proliferative, and we now cite such work (see doi: 10.1016/j.cell.2023.02.021).

6. Likewise for the suggestions of three branches involved in Tpex giving rise to Tmem and Teff cells. The data they show can easily fit the commonly accepted model of Tpex giving rise a stem-like daughter and what is called here a Tmem cell that can further differentiate into a Teff. It is hard to see how the very limited data they provide on phenotype and position provide a more definitive answer to the lineage issue than scRNAseq velocity or pseudotime analyses or experiments with cells transfers of isolated T cell populations.

We agree completely with the reviewer on this point and did not intend to imply otherwise. We have reworded this section of the paper to emphasize interactions and spatial positioning rather than implying a novel discovery about lineage.

7. In the membrane to membrane section, they suggest they have observed relationships distinct from what is in the literature - "Type II interactions between CD4 and CD8 T cells were common across the MIS, without evidence of a nearby antigen presenting cell (e.g., dendritic cell), making this an atypical interaction with respect to current understanding of T cell biology." It is unclear to me why they make this statement. T cells can contact each other in a complex tissue environment but in the absence of a pMHC ligand for the TCR on at least one of the interacting T cells, upregulation of integrin affinity needed to make a firm (that is, tight) membrane-membrane association will not occur. Transient cell-cell contact can occur but the negative charge of a typical cell membrane will prevent close leaflet binding unless strong bonds are formed that for T cells typically involve LFA-1 and ICAM or VLA and VCAM. They also ignore the substantial evidence of antigen presentation by T cells to each other in such an environment, so considering this to be an interaction devoid of an antigen-presenting cell is simply incorrect. This is especially cogent with human tissues where the activated T cells can express MHC class II molecules in addition to MHC class I molecules.

Again, we have no disagreement with the reviewer on this point and are not sure what has raised their ire. We do see many tight CD4-CD8 associations without a proximate APC (as a statement of fact) but we **do not** mean to imply that this changes our view on well-established mechanisms of antigen presentation. We will have modified the manuscript to make clear that we are describing the organisation of the TME, not a mechanism of immune regulation.

8. The statement "One state missing from our primary melanomas but present in metastases is an NGFR-high state, which has been described as neural crest (stem-like) and tumour-initiating. Thus, high cell plasticity may precede the appearance of stem-like states rather than derive from them." is not supported by the data, given that only two sections were studied from a single tumor of this type. Without more data on additional sections and additional cases with the same tumor, such a conclusion cannot be drawn.

We have previously established this feature of early melanomas and have now cited that earlier evidence (see doi: 10.1158/2159-8290.CD-21-1357).

9. The authors write - "Moreover, we find that individual immune cells are often subject to opposing regulatory signals: a single cytotoxic GZMB+ CD8 T cell can be polarized toward a tumour cell, enveloped by filipodia from a CD4 helper cell, and repressed by a PDL1-expressing myeloid cell." This statement involves a misunderstanding of how PD-1 works. If a T cell has its immunological synapse containing the TCR polarized towards an antigen-bearing tumor cell, PD-1:PD-L1 interactions with a DC away from that synaptic region will actually decrease the inhibitory effect of PD-1 because such inhibition requires colocalization with the TCR to mediate a repressive effect on local phosphorylation events via SHP phosphatase recruitment. The cell interactions described would thus NOT include a repressive interaction with the DC for this reason.

We have collected new data on this topic and now illustrate cases in which CD8 T cells are directing granzyme B particles to tumour cells, with simultaneous PD1 PDL1 interactions occurring within a few microns of the point of membrane-membrane interaction. The CD8 T cells are also interacting

with helper T cells and dendritic cells. We think these types of multi-dentate interactions make our point, although we obviously cannot demonstrate the functional consequences of opposing regulatory interactions. In multiple cases, these interactions co-occur within 3-5% of the membrane circumference. No our knowledge, there does not exist sufficient in vitro literature to determine whether this constitutes a local interaction or not. Regardless, we have rephrased this section of the manuscript to make the conclusions more speculative.

Referee #2

(Remarks to the Author):

The authors use cyclic imaging (long established) in a tissue-clearing method using confocal microscopy. They use 5um sections as their comparitor for advances despite clearing and multiphoton and other methods for seeing deep connections (see for example beautiful images produced by neurobiologists of neurites projecting and interweaving hundreds of microns into tissue.

The difficult part of this paper though is that, although some work seems of interest, it looks like a compendium of factoids and is hard to see a thread besides the imaging method. I also found many cases where I questioned the thresholding/surfacing in Imaris which can be tuned in many ways to smooth data and which I find difficult to accept as definitive.

I would kindly recommend the authors to submit this to a methods journal. The advance of putting Cytif together with clearing is hard to be excited about given the other excellent methods. With that, focusing on one or a few biological processes might make a study that is more compelling.

Thank you for this suggestion; we have reworked our manuscript for consideration at *Nature Methods*. We have also made significant improvements to the manuscript based on reviewer comments and added web resources with more methodological detail see protocols.io [<https://dx.doi.org/10.17504/protocols.io.261qe59m7g47/v1>].

Referee #3

(Remarks to the Author)

Yapp and Nirmal et al. demonstrates use of scanning confocal microscopy for cyclic immunofluorescence in archival human tissue samples of 30-40 µm range, which makes it possible to image 2-3 cell layers as opposed to the more standard and high-throughput 5 µm thin sections. It illustrates the potential of this approach for precise phenotyping of cellular and subcellular morphologies and cell-cell interactions.

As is, the paper very nicely underscores the potential of implementing multiplexing with higher resolution and 3D imaging. The generated (very rich) data that encompasses five tissue types and 11 datasets is impressive and can be analyzed in many different ways, however only three datasets are utilized for the current manuscript. For these 3 specimens, observational instances from the imaging data are highlighted in relation to 5 topics: (i) cell and organelle morphology, (ii) the microanatomy of cell clusters and structures (e.g. blood vessels), (iii) cell shape, (iv) tumour and immune cell lineages, and (v) cell-cell interactions (proximity analysis). Renderings are beautiful

and interesting observations are highlighted with potential functional implications based on literature. However, the paper reads almost anecdotal due to the main focus being on snapshots of different cases in the format of vignettes. The highlighted cases are definitely interesting, but the usability of the information is not immediately clear without quantitative analysis of more instances or backing up the observational statements with more objective metrics. For the broad Nature readership, the various highlighted instances may be rather specific cases, which would be of interest to multiple different expert groups, though in some cases they may be considered speculative due to lack of quantitative and comparative analysis. The proximity analysis is the most interesting usecase however, as is, it is only scratching the surface of the potential utility of the presented approach.

We thank the reviewer for these comments and agree that the manuscript as written is more suitable for a venue other than *Nature*; we are therefore pleased that the paper is under consideration for *Nature Methods*.

I think the paper could have come across very differently if the authors follow-up some of these vignettes, and quantify the significance of the highlighted observation at the level of whole tissue sections and compare conditions. Or alternatively do more rigorous technical assessments to more quantitatively show the impact/worth of their approach in terms of how it may alter the results of the analyses typically done to infer cell types/states/frequencies, spatial niches/neighborhoods etc. From a methods perspective, an important aspect of the manuscript would be to enable others to utilize a similar 3D approach by presenting the necessary optimizations and guidance on imaging parameters and the analysis pipelines for the different use cases presented. However, in the current format of the manuscript, this does not appear to be the main highlight of the paper either.

Overall, the manuscript can be considered as a mix of a methods paper and some potential biological insights for selected uses cases based on literature, though not sufficiently going in depth in either direction.

As described in greater detail below, the revised manuscript includes many more technical assessments of differences between 2D and 3D imaging. We have also significantly extended the method so it works with a wider range of tissue, and we have dramatically expanded the methods section.

More specific comments:

1. For better readability, the markers that are used to call the cell type (melanocyte, epithelial, tumour cell etc. should ideally be marked for each panel).

We thank the reviewer for this suggestion and have done our best to include this information on the figure panel or in the caption (when space does not allow). For the CD8 T cell subtypes discussed in **Figures 5 and 6**, we have included just the abbreviated calls (TPEX, TEFF, etc.) in the figure panels. Since each of these subtypes is defined by several markers, we felt it would be overwhelming to list all markers repeatedly. We have also referenced **Supplementary Figure 13** (our figure depicting how markers map to cell types) so that readers will easily know where to look for cell type calling information. It is challenging to accurately describe a 50-plex image in a compact manner.

2. The quantification data for the following statement should be included in the paper (for n > 1 FOV). Actually the more useful information would be the estimate of what percentage of a typical cell is captured (which could be even done for a few different cell types that the authors are highlighting, to further show cell-type dependent the bias): “This revealed that, in conventional 5 μm sections, nearly all nuclei (>90%) were incomplete along the Z (optical) axis whereas, in 30-40 μm sections, 60-80% of nuclear envelopes were complete along all three axes (Fig. 1a; Supplementary Video 1).”

We thank the reviewer for this excellent suggestion. We have added quantitative data regarding the percentage of cells captured within different tissue thicknesses (and the percentage of cell volume missing) to **Figure 1e** and **Extended Data Figure 1i-j**. We have also revised all of the thicknesses in the manuscript given our recent understanding of the effects of antigen retrieval and hydration on tissue thickness.

3. It would be great if the authors were actually able to do automated cell type inference from the 3D version 5 μm section data for the WSI and made a quantitative comparison of the cell types (like partially done in Extended Data Fig. 1b - but both for 3D and 2D images). “When we created sequential synthetic 5 μm serial ‘sections’ (2D maximum intensity projections; labelled I to V) along the Z axis, inferred cell types varied with the section and was often incorrect (Fig. 1d).”

We thank the reviewer for this good suggestion. Fully automated cell type calling in 3D is currently beyond our capability, but we have now provided in **Figure 1e** a quantitative comparison between virtual thin sections at 35-micron sections for polarised immune markers such as LAG3, MX1, and combinations thereof. We observed that over a third of cells would have been falsely designated negative for each of these markers if the imaging were to have been performed on thin sections. This is less pronounced for non-polarized markers (e.g., CD103+ and CD8+), however, we still observe 5-15% undercounting of these cell types with thin sections. Of note, SOX10 represents a least undercounted scenario, likely because SOX10 is a nuclear marker. We believe that this will substantially address the reviewer’s concern that 3D imaging of thick samples is needed for classifying immune markers that are sensitive to spot counting.

4. The showcase of the Lag3 almost feels like a missed opportunity, as there is a lot more analysis that can be done regarding this. For example clustering the cells based on other markers and then comparing the LAG3 spots per cell.

Thank you for the suggestion. We have these data and have now added them to **Extended Data Figure 4b**. This will allow us to compare spot distributions for LAG3 relative to other proximal cells.

5. “In the MIS as a whole, we found that B cells were the type of cell most likely to associate with collagen fibres in the dermis^{20,21} (Fig. 3b, c); they were often (n = 11 of 14 in the MIS) stretched into irregular shapes (this was not a feature of all B cells and those found in the stroma were often round; Extended Data Fig. 3d)”. The data that supports the second part of the statement is not featured or accessible in a quantitative manner, as no analysis result is shown. Fig.3c: n numbers are missing.

We thank the reviewer for pointing out these errors and omissions. We have quantified B cell sphericity and volume between the melanoma in-situ and invasive margin and provide numerical evidence for our assertion that B cells are more elongated in the melanoma in-situ. This can now be found in **Extended Data Figure 3e** (and pasted below). We have added numbers to **Figure 2c** (formerly **Figure 3c**).

6. “Cell shape and motility are regulated by the actomyosin cytoskeleton which was most prominent in dendritic and T cells (~2-fold more actin overall than melanocytic cells and 3-fold more than in keratinocytes; Fig. 3d).”

– Here again the quantitative data behind this statement and numbers is not shown anywhere. Similarly the rest of the statements for Fig. 3e-f do not include any quantification for the reader to evaluate the significance of the observations that are highlighted.

Thank you for this suggestion. We have chosen to remove this section to make room for more methodological detail and to conform to the word limit for Nature Methods.

However, we have calculated the mean intensity of b-actin across different cell types and have provided them here by way of addressing this question.

The mean intensity of b-actin for different cell types in the melanoma in-situ

In certain cases like Extended Data Fig. 4b, based on the color gradient it seems like a complex lookup table (like fire) was used instead of red in multicolor overlay images (without an explicit explanation). I cannot be sure if this is the case, as the legend only says red, but if this is the case it is very confusing for interpreting the images and actual overlaps.

(Furthermore, in certain cases almost the same colormap is used for different protein targets - Supplementary figure 2, row 2 column 1 for example. CD8 and PDL1 are colored using almost the same colormap.)

Thank you for catching this. We used a LUT for the overlay and forgot to define it appropriately.

However, we have chosen to remove this section to make room for more methodological detail and to conform to the word limit for Nature Methods.

7. Extended Fig. 5 Panel c - this would be good to also do on virtual thin sections, to see the differential impact of the 3D approach.

We are unsure precisely what the reviewer is proposing. **Extended Data Figure 5c** is intended to show that there are MART1+ tumour cells breaking through the DEJ - this is an observation that we would expect to appear on conventional thin sections, as it is typically observed during histopathology diagnosis.

We refer to panel 5c at the same time as panels 5b and d in the text, so the reviewer may be asking about whether virtual thin sections would change the quantification in Extended Data 5b, regarding the distance between MHC-1 expression and cells containing MX1 puncta. We expect that there would be a significant difference between 2D and 3D for this quantification. This is twofold – first, as shown in **Figure 1d** and **Supplemental Note 1**, lower resolution imaging and the inability to reject

out of focus light significantly reduces our ability to quantify punctate signals compared to the high-resolution approach. Additionally, as seen in our new **Figure 1e** and **Extended Data Figure 1i-j**, thin sections significantly reduce the portion of the cell captured within an image, and make it difficult to phenotype cells as MX1 positive.

8. Nonetheless, the MIS and underlying dermis were not highly proliferative, with only 1% of cells (n = 110) positive for the Ki67+ proliferation marker. Among Ki67+ cells, 34% were T cells while the remainder consisted of monocytes (28%) and endothelial cells (2.7%); only a single melanocytic cell was Ki67+. By contrast, in the invasive VGP melanoma domain from the same specimen, 11% of all cells were Ki67+ with melanoma tumour cells the most proliferative (45% Ki67+), followed by monocytes (44%). It is unclear which image data is the source of these statements as no figure is cited.

We thank the reviewer for pointing this out. The data refers to dataset 1. We have included the data as a graph in **Extended Data Figure 5a** (and pasted below).

9. "These data imply that melanocytic cells with features of early malignant transformation are subject to frequent changes in cell state (phenotypic plasticity) rather than progressive evolution from a single transformed or progenitor (stem-like) cell, as proposed for advanced invasive melanoma." -- The suggested implications is rather speculative as progressive evolution may happen at the genomic and epigenomic level without being apparent in the visual phenotype.

The reviewer is of course correct. We have modified the text to make clear that our conclusion is based on a limited set of markers, albeit ones that are widely used diagnostically. We believe the data support a conclusion of frequent change of state, but they say nothing about the origins of these changes (genetic or epigenetic). The new text reads "*These data imply that phenotypic plasticity is a feature of melanocytic cells undergoing the earliest stages of malignant transformation and that invasion into normal tissue can occur in the absence of the stem-like NGFR+ cells found in metastatic melanoma.*"

10. Fig. 4s: although the analysis performed here is interesting, based on n= 875 cells that is written in the main text (which should also be included in the legend, with the information that it is from one tissue section), It seems like cells in the different frequency bins were not counted exclusively (i.e. the total for the different columns add up to much more than this number) or the given n number is not correct.

We struggled with the presentation of this data. Our original form did not count cells exclusively, which may have been confusing to readers (it was also a concern of review 1) In the original plot, black represented “not specified” markers, i.e., the given marker could be either positive or negative. We have revised this plot (formerly **Figure 4s**, now **Extended Data Figure 5b**) to show specific positive/negative states for all six markers. We believe this is more useful for readers, as it better highlights the specific proportions of various states of interest. We have also redone the images to make the different states more visually intuitive. We trust that this helps.

11. The criteria for inferring IFN+ in Fig. 5a-b are not really clear, as the panel shows both correlated and uncorrelated expressions/distributions for the 3 markers (panel b). For panel a, can the authors rule out that these spatial domains are not connected in z (would imaging even thicker samples be necessary for this), since there appears to be smaller domains in between depending on how a domain is specified?

The reviewer makes a good point. It would be interesting to explore whether the IFN-positive domains are connected in three-dimensions. We have previously shown that other tumor features (e.g., mucin pools and tumor ‘buds’ in CRC are part of larger, interconnected 3D networks; Lin et al., Cell, 2023). As you suggest, this would require thicker specimens, which is a limitation of our current work. We have mentioned this in the discussion as an interesting question to explore in future efforts. We have toned back our language in this section to make it clearer that there is further work to be done to definitively characterise these IFN-positive domains.

12. Extended Data Fig. 5e - does this panel also include PCNA staining to show the PCNA negativity (according to the figure caption it should, but the label on the figure does not include it)?

We thank the reviewer for pointing out this error. PCNA was included in this panel and is negative. We did not previously include PCNA as we thought it would obscure other markers. We have added PCNA into **Extended Data Figure 6d** (formerly **Extended Data Figure 5e**).

13. T cell subpopulation/lineage and tumor proximity quantifications in Fig.5 and Extended Data Fig. 5 are some of the most useful results, as they provide more quantitative insights. A very relevant question here is how much of this information can be captured with typical thin sections, to see the impact of the 3D imaging on detailed cell type calling and quantification. If the authors had a dataset from a more typical 5 μm section with widefield imaging for doing this type of comparison from a consecutive section of the same sample that would have been the ideal way to show the comparative advantages of the approach in this paper over the more standard approach. As a worst case, the comparison can be done on virtual sections of the given dataset as in previous parts of the manuscript (which would also help to see section to section heterogeneity).

We thank the reviewer for this suggestion. We note that proximity in the current work is scored based on membrane-membrane association rather than nuclear location. We have in the past attempted to repeat distance/proximity metrics on membranes using the more common widefield imaging approach sampled with a 20x/0.75NA lens. The sampling is often too coarse, and the out-of-focus blur is too great to give confidence about cell contact. To address this point in another way, we have collected a proximate 5 μm section using conventional 2D imaging approaches and generated synthetic 5 μm sections. These are shown in **Extended Data Figure 1** and illustrate in a quantitative manner that the majority of cells will be incomplete in traditional 5-micron sections.

Moreover, using synthetic data we can prove that neighbourhood interactions are significantly undercounted in thinner sections.

14. The proximity analysis to infer cell-cell interaction modes is definitely an informative and interesting analysis that leverages the resolution of the data and a potential highlight of the paper. It actually creates a great opportunity to guide future studies on the imaging parameters that should be taken into account to be able to perform such analysis in a technically sound way, however relevant technical assessments and discussion is missing in the paper.

We thank the reviewer for this excellent suggestion. We have extensive detail on this point in the **Methods** with accompanying **Supplementary Figures 20-22**). We specifically examine the impact of signal-to-noise ratio, sampling rate (pixel size), numerical aperture, averaging along membranes, and Z-stacking on the ability to characterise cell membrane interactions. We also include line profiles and confocal images for each example.

Fig. 6 - The juxtaposition quantifications with distances going down to 60-70 nm, raises relevant questions about the technical error margin. What is the measured registration error margin across cycles and fluorescent channel in 2D and 3D- when potential imperfections are taken into account what is the confidence level for the subpixel scale quantifications?

Based on the analysis, the authors are encouraged to discuss what would be the minimum specs (pixel size, resolution, z-step size etc.) to generate the data that would be permissive to proximity analysis to infer all 3 interaction types.

How should the analysis be ideally done: on one or multiple single planes, on projection images? Is it necessary to look at the whole interaction surface in 3D or are the contacts only visible in a single plane? It is very difficult to understand and the line profiles done in Fig. 6 should at least be also performed along the z axis to reveal the distances in 3D, within the limit of the confocal microscopy and acquisition settings used.

These are all excellent questions. There is nothing unusual about inferring super-resolution features from images, but proximity calculations are dependent on signal-to-noise ratios and registration accuracy, as the reviewer points out. As stated above, we now include a detailed discussion in the methods and new **Supplementary Figures 20-22** on cell-cell interactions. The note compares these factors using experimental data and how they influence proximity calculations.

Axially, we sample at the Nyquist limit, but laterally, we have chosen to sample at about double the Nyquist limit (140 nm vs 80 nm) to reduce data size and increase throughput. In the revised paper, we demonstrate that under sampling laterally can still robustly detect cellular interactions down to ~50 nm (subpixel accuracy) using curve fitting approaches. This is assuming that signal to noise is optimal through the use of optical sectioning (confocality vs widefield) and using optimized antibodies. We also examined how, in noisy data, the use of different types of line profiles orthogonal to the membrane and projections across multiple z planes can have a beneficial impact on identifying membranes. Finally we cited: DOI: [10.1091/mbc.e17-01-0069](https://doi.org/10.1091/mbc.e17-01-0069) as a relevant citation for measuring distances.

Finally, we have manually identified regions of poor nuclei segmentation between each cycle by overlaying Hoechst channels and looking for overlap. We have excluded areas that have poor registration for proximity analysis as well as cell type calling. These errors affect a small minority of the data.

Methods:

15. The controls that were performed to find the optimal conditions and potential limitations are highly appreciated (Supplementary Fig. 13-17). However, these very important technical considerations including optimal section thickness, effect of fluorophore choice, necessary alterations of the sample preparation and CyclF protocol for thicker specimen as well as important processing details (registration, segmentation), which are some of the most challenging parts of dealing with this kind of data are buried in the Methods section and in the Supplementary. These should at least be discussed/mentioned and cited in the main text.

This is an excellent suggestion. In response, we have prepared a complete supplementary technical guide and on-line protocol at protocols.io

[<https://dx.doi.org/10.17504/protocols.io.261qe59m7q47/v1>]. We have also added to the methods and added additional supplementary figures in the hope that these can be more easily understood. Finally, we have developed a new alternative approach to 3D imaging that overcomes some of the limitations encountered when working with fragile tissues; this approach is shown in **Extended Figure 1**.

16. Regarding the antibody penetration heterogeneity between different antibodies and conjugates - previous works (for example: Murray et al, Cell 2015, doi: 10.1016/j.cell.2015.11.025) suggest that an important factor may be the distribution of targets along the diffusion direction (i.e. if the target is highly abundant in the surface layer, the effective concentration of the antibody may get lower or the better-binding conjugates may be depleted as it penetrates through the sample causing ineffective staining in the lower layers. And in that case slower/less efficient binding might allow better penetration and more consistent staining along z). That may also partially explain the observed differences between fluorophores (i.e. if a particular conjugate yields a lower ON rate, it may diffuse better before binding to the top layer very effectively).

As mentioned in our response to reviewer 1, we agree completely with this conclusion. We thank the reviewer for the citation and have included it in our revision. We have also expanded our discussion at this section of the methods and added a new **Supplementary Figure 14** that illustrates the antibody penetration over time.

17. Regarding the CyclF protocol, were any other changes introduced with wash times etc? What temperature was used for 8-10 h antibody incubations?

There were changes to the basic protocol and we have included all of this information in the online protocol, as mentioned in the response to point 15. Among the changes, we perform an 8-10-hour antibody incubation at room temperature. This time was determined experimentally. We now show a staining time lapse in **Supplementary Figure 14** to further explain this optimisation.

18. For the MATLAB functions, authors should include the parameters used, when applicable.

We thank the reviewer for pointing this out; we have added MATLAB code and parameters to the GitHub Repository (<https://github.com/labsyspharm/mel-3d-mis>).

19. The Supplementary Discussion section is very helpful. I would also recommend some more references to be included there for completeness (to the part: "While spectral unmixing, spillover

compensation, or the use of Raman dyes can increase this number to 8-10 channels^{71–73}, to date, sequential staining and high-plex high-resolution (sub-micrometer scale) imaging of thick sections has not been demonstrated.”). Although a direct demonstration of sequential staining and high-plex high-resolution (sub-micrometer scale) imaging of thick sections for FFPE samples has been missing, there are many efforts along this way, which should ideally be cited, including:

-Joha Park, Sarim Khan, Dae Hee Yun, Taeyun Ku, Katherine L. Villa, Jichen E. Lee, Qiangge Zhang, Juhyuk Park, Guoping Feng, Elly Nedivi, Kwanghun Chung (2021) Epitope-preserving magnified analysis of proteome (eMAP). *Science advances* 7.46 (2021): eabf6589.

- ExM- exchange-based multiplexing combination: Saka, S.K., Wang, Y., Kishi, J.Y. et al. Immuno-SABER enables highly multiplexed and amplified protein imaging in tissues. *Nat Biotechnol* 37, 1080–1090 (2019). <https://doi.org/10.1038/s41587-019-0207-y>

-Taeyun Ku, Justin Swaney, Jeong-Yoon Park, Alexandre Albanese, Evan Murray, Jae Hun Cho, Young-Gyun Park, Vamsi Mangena, Jiapeli Chen & Kwanghun Chung (2016) Multiplexed and scalable super-resolution imaging of three-dimensional protein localization in size-adjustable tissues. *Nature Biotechnology* 34:973–981.

We thank the reviewer for providing this excellent list of missing references. We have modified this supplementary discussion and added the new references to the revised manuscript.

20. Code availability: the repository currently contains only matlab code, although other (python) scripts are mentioned in the paper. The github link is currently broken.

Most python scripts were derived from other published packages and we have made sure they are correctly and clearly referenced. Python scripts related to cell type calling have now been added to the GitHub repository (<https://github.com/labsyspharm/mel-3d-mis>) along with all MATLAB scripts and parameters (which were primarily used to generate figures). We will ensure that all links work appropriately and these scripts will also be referenced in the new technical guide.

21. Some of the preprint references are not updated to the published versions.

We thank the reviewer for pointing this out. Our manuscript has been in review for an extended period of time, and this may explain some of these errors and omissions. We have carefully reviewed our references and believe they are now up to date.

RESPONSES TO REVIEW

NMETH-A57303: *Highly Multiplexed 3D Profiling of Cell States and Immune Niches in Human Tumours*

INDEX

FIRST ROUND OF REVIEW 4/7/2025 (NMETH-A57303-T)

Note that this manuscript was first submitted to Nature, but the response to review reflects a subsequent transfer to Nature Methods

Overview of the Response

Detailed Response to review

Reviewer 1

Reviewer 2

Reviewer 3

SECOND ROUND OF REVIEW 5/5/2025 (NMETH-A57303A)

Proposed revisions

Detailed Response to Review

Editorial Comments

Reviewer 1

Reviewer 2

RESPONSE TO REVIEW - 4/7/2025

NMETH-A57303-T

Highly Multiplexed 3D Profiling of Cell States and Immune Niches in Human Tumours
Yapp... Sorger et al.

OVERVIEW OF THE RESPONSE

We thank the reviewers for their thorough and detailed review of our manuscript. We believe that we have addressed all their concerns through revisions to the text, additional experiments and analysis, and the inclusion of expensive new data. To fit this new data into the manuscript we have removed several results relating to actin biology and the cytoskeleton; we do not believe that these are central to the story. We provide an overview of our response below, followed by a detailed point-by-point response.

1. New results and methods. As we worked through the quantitative analysis requested by the reviewers, we made several unexpected “discoveries” about tissue imaging that resulted in major improvements to the approach. Most notable among these is the fact that tissue sections processed for immunofluorescence expand ~ 1.5 fold in thickness when subjected to antigen retrieval and antibody staining. This hydration-dependent expansion is consistent with earlier literature, but to our knowledge, has not been quantified and incorporated into models of 3D tissue architecture. We have added to the manuscript a series of experiments that argue that hydrated tissue dimensions, and not the dimensions encountered in standard H&E images (which involve dehydrated specimens), are representative of actual tissue, at least at the level of cells and cell communities.

We have also undertaken a major effort to make thick section multiplexing more reliable across tissue types (as requested by reviewer 3). This has led to the development of two different approaches to imaging: a conventional one in which tissue is mounted to glass slides and an alternative one in which tissue is mounted to coverslips and overlaid with adhesive substrates (Matrigel) and physical grids to decrease tissue damage during cycling. We now demonstrate that these methods work well across a wide range of tissues, increasing the usability of our approach. All of these updates and new results are described in Figure 1 and Extended Figure 1.

2. Relationship to (and citation of) earlier work. Reviewer 1 pointed out a number of citations that we had missed in our first draft. We sincerely apologise, particularly since several of the missed manuscripts come from one of the leaders in multiplex imaging (the Germain group at NIH). We have corrected these omissions. At the same time, we believe that the reviewer has not fully recognized the difference between working with specimens from model organisms (such as mice) that can be frozen or alcohol fixed, and human specimens, which are almost always formaldehyde fixed and paraffin embedded (FFPE). The methods developed for tissues from model organisms in a research setting cannot be transferred directly to imaging FFPE specimens from the clinic. This is an important aspect of our work that we underemphasized in the original draft.

In addition, the reviewer describes as standard, microscopes and imaging conditions (e.g. 1.4 NA oil immersion objectives) that are commonly used by cell biologists working with cultured cells (and animal models); but are very different from the high-speed scanning microscopes used for tissue profiling in all of the recently popularised methods such as MIBI, MxIF, CODEX (Akoya), COMET (Lunaphore) and CyCIF. These latter approaches use lower resolution optics and single optical sections because the specimens are large, and the emphasis is on the efficiency of data collection: a whole-slide tissue image commonly comprises over 1,000 conventional fields of view (e.g. collected with a 0.5 to 0.75 NA dry (air) objectives). The resolution we achieve in this paper, and thus the ability to discriminate very fine features and cell membranes, is unprecedented for high-plex images of human tissue specimens using either thick or thin sections.

3. 2D vs 3D Imaging. Reviewer 1 makes the assertion that our data do not constitute true 3D imaging since the “3D” label only applies when specimens are thicker or imaged using fluorescence light sheet microscopy. This does not seem to us a defensible position: a 2D image has a single optical plane and a 3D image has multiple planes (in the case of our images, >170). The size of the specimen, and whether light sheet or confocal methods are used to acquire the data, is an entirely separate issue. Moreover, we believe one of the most interesting findings of our study is that after 75 years of immunofluorescence imaging of 5 μm tissue sections, we find that few if any intact cells have been examined.

We have nonetheless written a supplementary note on different approaches to 2D and 3D optical microscopy that we believe fully addresses the concerns of reviewer 1. We have also modified Figure 1 to better illustrate key features of 2D vs 3D imaging. These will be familiar to many experienced microscopists (like Reviewer 1) and we believe that the succinct “explanatory” text in the body of the manuscript is important for a spatial proteomics community that thinks less about imaging technology; the more detailed arguments find a home in the supplement.

Reviewer 2 takes an opposite approach from Reviewer 1 and suggests that we examine the implications of 2D vs 3D imaging more carefully and better quantify the impact of using thin tissue sections. This is an excellent idea that we have thoroughly explored. As described below, we now address this directly.

4. Organization of the manuscript. Both Reviewer 1 and 2 were concerned that our manuscript did not conform to the expected layout of a *Nature* manuscript. They correctly point out that our manuscript has a substantial methodological component followed by a series of illustrative findings that can be made using high resolution 3D images. They suggest that a more focused look at one of these topics with more specimens, or additional tissue types would make for a stronger “science” story. We have better focused the current study and we have modified the formatting to fit the *Nature Methods* style.

5. Request for additional detail of the method. Both reviewers ask for additional details with respect to the performance of the method. We have therefore prepared a detailed protocol for the on-line methods and in a protocols.io description. We have also built a web resource for the data and methods and we will keep this continuously updated as methods improve and additional reagents or tissue types are tested.

DETAILED RESPONSE TO REVIEW

REFEREE 1

In this submission from Yapp et al. entitled “Multiplexed 3D Analysis of Immune States and Niches in Human Tissue” the authors use 35µm thick sections of human melanomas to conduct what they term a “3D” analysis. This follows on from prior work of the Sorger lab involving 3D imaging using serial thin sections. The central claim in the present manuscript is that by extending the thickness of a section to 30-40 µm, it is possible to capture entire cells rather than cutting through most cells as would be the case for the more typical sections of 5-10 µm used for multiplex analysis. With this more complete image of intact cells, they conduct an analysis of the phenotype, surface protein distribution, intracellular organelle content and structure, and spatial relationships within the TME. As would be expected from two senior investigators with many key contributions to the generation and detailed analysis of optical imaging data (Sorger and Danuser), this study is replete with the use of advanced algorithms for image dissection and the rendered surface interpretations of the primary immunofluorescence (IF) data are elegant. Beyond the imaging itself, the authors draw a series of conclusions about the distribution of immune subsets and cancer cells in two different types of melanoma, as well as providing interpretations of the signaling events occurring in cells based on the patterns of surface protein distribution as well as the distance between opposing membranes.

Evaluation of this manuscript is difficult, as it represents a combination of biological claims and technical developments. There is substantial value in the technical details of how the authors both collected the primary data and in how it has been processed to provide multiscale information ranging from intracellular organization to local interactions, to overall representation of cell types within the TME. At the same time, many of the biological interpretations are not supported by adequate replicate experiments, either from the same tumor sample or from additional patient material. In addition, a number of the interpretations are of concern, several of the novelty claims on the technical side fail to properly credit published work from other groups, and some of the discussion in the supplement about achieving ‘real’ multiplex 3D imaging is incorrect based on recent findings from other laboratories.

As described in the overview above, we believe that our manuscript’s combination of “biological claims and technical developments” is a better fit to *Nature Methods*. We have extensively modified the manuscript throughout to address concerns raised in these two paragraphs regarding credit, biological interpretations, and novelty. These modifications are described more completely below in response to the reviewer’s numbered comments.

Technical issues:

1. The major premise of the current paper is that multiplex IF has not been applied to sections thicker than 5-10 µm and that such data even from these sections are at low resolution (“Nonetheless, almost all contemporary high-plex image-based profiling emphasizes relatively low resolution 2D imaging (0.6 to 2.0 µm lateral resolution) of conventional 5µm thick tissue sections to increase the speed and convenience of data acquisition.”). This is not accurate. Radtke et al. (Proc Natl Acad Sci U S A. 2020 Dec 29;117(52):33455-33465; Nat Protoc. 2022 Feb;17(2):378-401) have reported high multiplex IF using a method called IBEX with sections from diverse mouse and human tissues sectioned at 30 µm and imaged at a resolution comparable to that reported here (~250 x 250nm in XY). It is true that these investigators did not use the data for the deep study of individual cell features such as granule content or precise distribution of membrane proteins, but the imaging data needed for such analyses has been collected by others and the implication here that this is a new approach in multiplex imaging is not really true. It should also be noted that many of the observations about cell-cell interactions, T cell contact with dendrites, morphology of invading tumor cells and so on, have been published in papers using 2-photon intravital imaging by multiple groups (Cahalan, von Andrian, Mempel, Friedl, Germain, Bousso, Krummel, and others). These are

not high-plex images, being constrained by detector number to 4-6 parameters, but many of the observations made in static images here correspond to what is in the work from these other groups and it seems proper to credit such prior publications when discussing the present observations.

We thank the reviewer for drawing our attention to these papers and to a series of review articles by the same senior author; we have added these references to the manuscript. We agree that we should have mentioned the IBEX method (and now do - prominently) but we do not agree that it substantially overlaps with our paper. IBEX has been primarily applied to flash-frozen tissue, and only cursorily applied to FFPE tissues, which are more challenging to stain because of their low epitope abundance (as is mentioned in the Radtke 2020 paper). Additionally, we sample in higher plex, at finer resolution in both lateral and axial dimensions (2x and 3x more respectively), and at twice the bit depth. However, the IBEX method should have been mentioned, and we have corrected this error in our revised manuscript.

More significantly, we spend the great majority of the paper not on technical details, but rather on illustrating what can be achieved using high resolution, high sensitivity, 3D imaging. 2-photon intravital imaging is an important method, but it is restricted to 2-3 channels and performed exclusively in animal models. We do not believe that the current paper is the right place to review the totality of imaging methods applicable to tissue (although we make a stab at this in our new Supplementary Note 1), but we have tried to accommodate the reviewer's concern by adding a section on high-plex fixed cell and low-plex live cell intravital imaging to the conclusion. Finally, we submit that while the reviewer is correct that microscopists image tissues at a wide range of resolutions (particularly with model organisms), our statement that "*almost all contemporary high-plex [spatial proteomics] emphasizes [efficient but] relatively low resolution 2D imaging (0.6 to 2.0 μm lateral resolution*" remains true.

2. While I appreciate the difficulty of conducting >50 plex IF imaging with minimal noise and high reliability of all antibodies in all cycles, the current study only uses 3 sections of tissue for all the analyses. While this is adequate to show that they can imaging at 35 uM and obtain full cell shapes in such a data set, it is inadequate to provide quantitatively representative data even from the specific tumor samples giving rise to these images, no less as considering the data to be representative of the two melanoma disease states. Indeed, there are now a substantial number of high multiplex analyses of tumor tissue in the literature and these range from several dozen to several hundred samples each (see work from Angelo, Bodemiller, Nolan, and others). I think the authors should temper all their conclusions about tumor biology in the absence of further data; even better, they should conduct lower plex IF analysis of additional samples focused on their major conclusions about stem cells, invasion, cell interactions, and so on to see if their conclusions can be validated with these additional samples.

We respectfully disagree; comparison with Angelo, Bodemiller, and Nolan cannot be made due to dramatic difference in resolution and dimension of the specimens. Moreover, it is not true that the current study describes only 3 tissue sections; for simplicity, our analysis focuses on these tissues, but we provide many more in supplementary materials. The studies from Angelo, Bodemiller, Nolan, and others typically involve a similar number of 2D samples at 5-10 fold lower resolution, generating datasets 100-1000 fold smaller. Moreover, high-plex imaging of "several hundred samples" as described by the reviewer, has only been achieved with tissue microarrays, which represent very small pieces of tissue (typically <1 mm in diameter). The authors cited by the reviewer work largely with TMAs (or fields of view in whole slide images), which creates the impression of a rich data set, despite severe limitations in spatial power when dealing with such small specimens or FOVs (see DOI: 10.1016/j.cell.2022.12.028). Imaging hundreds of whole slide samples at high resolution on a laser scanning confocal is currently not feasible, but we do agree that it will be important to develop faster methods that balance throughput with resolution.

Overall, our manuscript includes 11 datasets, not only the three the reviewer identified. To reduce the complexity of our manuscript, we focus on 3 melanoma datasets for the body of the text. In

addition to the existing overview of each dataset in the supplement, we have now also made a high-resolution 2-D projection of each dataset using Minerva (an online visualisation software), where the reviewer can browse our images. Importantly, Minerva will allow readers to view full-resolution images and toggle channels on/off as they please. Links to the Minerva story for each dataset are available in **Supplementary Table 1**. We think this adds a significant value to readers, who can now browse each dataset easily online at full resolution - without download; however, the data are also freely downloadable for those who are able to manage such large image files.

3. With respect to the “3D” nature of the study, as they note, confocal imaging is inherently “3D” and even thin sections of 5-10uM contain ‘3D’ information on the contents of cells. 3D microscopic optical imaging has more conventionally been considered to be imaging at 100s of uM or even several mm dimensions that reveal the paths of tissue elements with ramifying distributions such as nerves, blood and lymphatic vessels, small airways, and ducts across sufficient tissue volume to understand their organization in space. The present work is far from such 3D analyses and it seem misleading to headline such a claim.

As described in the introduction to this response, there is no objective basis on which to make this claim. The 2D vs 3D distinction has nothing to do with the thickness of the specimen. Although there are many important observations that can be gleaned only from macroscale 3D analysis, our work provides an unprecedented “microscopic” 3-dimensional view of cellular interactions within native tumour context. We believe this to be a substantial advance from the current state of the art in imaging cells. By way of illustration, 3D optical deconvolution and superresolution microscopy of cultured cells, which are rarely thicker than ~10 um, has transformed cell biology over the past two decades. We also submit that the reviewer is ignoring the fact that nearly all of the high-profile spatial profiling papers reported over the last five years involve a single optical plane. Our new Supplementary Note outlines the merits of LSFM vs. confocal microscopy. The reviewer might be interested to know that we have developed an approach to cyclic light sheet fluorescence microscopy (LSFM) of mm thick FFPE human tissues and can indeed visualize “*nerves, blood and lymphatic vessels, small airways, and ducts.*” Of note, we also remount and resection these thick tissues for 3D confocal imaging. We are therefore confident that LSFM is not a substitute for the high resolution cell-level imaging shown here.

The authors might argue that with respect to multiplex IF, they are extending the usual work-flow from sections that lack most intact cells at 5-10uM to whole cells in 35uM sections and they have a point here. However, even at 35 uM, they will be missing portions of larger cells such as macrophages, dendritic cells, and stromal cells that often extend over 40-70uM. In addition, while they do capture and show a portion of a blood vessel with cells within the lumen, there are no data on the general organization of vessels within and across the tumor sample and no visualization of nerves.

The reviewer is correct. At any specific spatial scale it is difficult to discern features that have a larger characteristic dimension. The branching of blood vessels and nerves is better discerned using light sheet (LSFM) than confocal microscopy. However, this is not the purpose of our paper. Instead, we focus on showing that all or nearly all cells are incomplete in a typical 5 um tissue section but that in 35-40 um sections, >75% or more of all cells are intact (see new **Extended Data Figure 1i**). Some cell types will still be incomplete in 40 um sections, but the immune cells that are the focus of this paper are largely intact (excluding some dendritic cells). We carefully show why the switch to thick sections substantially impacts cell type calling, and the identification of intracellular structures. The reviewer does not mention the fact that LSFM is substantially (5-fold or more) lower in resolution than what we report, generally precluding a precise analysis of cell morphology (or processes such as extravasation - which we show for a blood vessel). LSFM does not currently extend beyond 5 channels (more typically 3). In sum then, the reviewer’s points are generally correct, but we submit that they are not germane to the work in our paper.

Furthermore, the speculation in this paper that confocal imaging cannot be used for high multiplex IF of thick tissue that does provide true 3D data is incorrect. Li et al (Nat Protoc. 2019 Jun;14(6):1708-1733; Proc Natl Acad Sci U S A. 2017 Aug 29;114(35):E7321-E7330) showed an entire mouse LN in 5-6 parameters and that same laboratory has stated in a review (Immunol Rev 2022 Mar;306(1):8-24) that they have achieved > 15-20 plex in > 300uM samples using confocal imaging. While the latter claim in a review in no way pre-empts this present submission, especially as the actual data are not available for evaluation and comparison, it does seem the authors should be more nuanced in how they describe their present data sets and also take into consideration the work of others doing larger scale multiplex 3D imaging.

With respect to these concerns, let us first state that we should have cited the C3D paper from the Germain laboratory and have now mentioned it in the discussion. We also prominently cite the IBEX work from the same lab and sincerely apologise that we missed these citations in our initial draft.

However, there is a very substantial difference between a single imaging cycle (which gives rise to 5-6 channels in Li et al) on a non-embedded murine tissue and high-plex (>50-channel) imaging of a human FFPE tissue. The latter approach introduces a host of problems with image registration, antibody staining, etc.

Importantly, the high-plex tissue imaging methods that have driven interest in spatial profiling enable a completely different approach to understanding tissue and tumour biology. These methods make it possible to distinguish cell types from each other and simultaneously assay cell state and cell-cell interactions. We were not aware of the Germain & Radtke 2021 review article, but our views are in line with the author's discussion about the in-progress 3D-IBEX method and we too believe that our manuscript highlights the value of high-resolution, highly-multiplexed confocal imaging for clarifying rare immune subtypes and revealing novel interactions within the tumour microenvironment. There is no precise threshold that distinguishes high-plex from conventional imaging (or transcript profiling) but high-dimensional data analysis methods generally become most effective over 20 channels, which is greater than what is claimed by the review article.

Unfortunately, we cannot compare our method to 3D-IBEX given that it is unpublished. However, we tried to accommodate the reviewer's concern by citing a 2024 meeting abstract from the Germain lab on 3D-IBEX (*Ce3D-IBEX: Achieving Multiplex 3-dimensional Imaging for Deep Phenotyping of Cells in Tissues*).

4. The authors largely ignore the entire field of stereology (e.g., METHODS 18, 493–507 (1999)) that has for some time worked on how thin 2D images mis-represent true tissue content. Such work should be cited as it provides methods for correcting what is seen in the thin sections. Of course, stereology does NOT provide data on the cell contents as nicely reported here, but the thin section-thick section issue is a key one of concern to this long-standing field and seems worthwhile to make readers aware of this literature.

This is a good point, we thank the reviewer for the suggestion. We have now added a brief section on stereology and suggested that our work should be integrated with previously developed methods for inferring 3D structures from 2D images.

5. The authors state "...cell segmentation methods rely on nuclei. Thus, accurate phenotyping of these three immune cells requires imaging whole cells to account for non-uniform protein distributions in cells and their overlap in Z." This is incorrect. In Li et al., Proc Natl Acad Sci U S A . 2017 Aug 29;114(35):E7321-E7330, the authors discuss segmentation in volumetric multiplex imaging and write "To circumvent these issues, we explored various image processing pipelines to accurately segment all imaged cells independent of nuclear staining.", followed by description of a pipeline for achieving such nucleus-free segmentation in 3D images.

Once again, we apologise for missing this paper from the Germain lab and specifically for missing the point that segmentation can be performed by summing multiple membrane markers together and applying watershed in mouse lymph nodes. We have tried such an approach ourselves and found that it is generally difficult to find such markers in the complex setting of a human tissue. A quick examination of the 50 or so papers published in the last five years to tackle the problem of cell segmentation in high-plex spatial profiles all use nuclei as the basis for the first step in segmentation. To our knowledge, this includes all of the commercial packages that make up the bulk of the published data in the field.

More importantly, we submit that the reviewer misses the point of this part of our manuscript in which we show what appears to be background signal in a thin section in fact represents foreground signal from a cell that lies above or below, but whose nucleus is not in the imaging plane (see **Figure 1d** and new quantification in **Figure 1e**). It seems highly improbable that any extant nuclei or *non-nuclei* based segmentation approach could accurately discern such signals in the complex environment of a human tissue or tumour. Whether to-be-developed machine learning approaches can address the issue remains to be determined, but for now we regard the non-assignable intensity to be a fundamental limitation of thin-section imaging which is nonetheless quite promising from the point of view of antibody specificity.

6. The authors then discuss slide scanner data, but most academic laboratories employing multiplex imaging conduct their data collection at high resolution; 40x 1.3 NA objectives are typical along with a high pixel count per image to yield data of ~250 x 250uM lateral resolution. Clinical labs using slide scanners are not conducting highly multiplex imaging of the type discussed here, so these comments are off the mark.

With respect, this is simply not true. As mentioned above, conventional low-plex tissue imaging (of model organisms and tissue culture cells) has long used high NA objectives but all of the work that has recently been published using CODEX, MxIF, CyCIF involves single optical planes acquired at resolutions closed to 600 nm laterally; in the case of MIBI and IMC, the resolution is as coarse as 1-5 um per pixel, whereas our study samples at a lateral resolution of 0.14 microns per pixel.

We submit however, that the reviewer has missed the point in these comments. The issue is not who has the better microscope, or smaller pixels, but rather what can be done at higher resolutions using 3D reconstruction. We believe that we have convincingly shown that with high-plex, high-resolution 3D imaging it is possible to discern subcellular structures (including those that drive tumour phenotypes and tumour-immune interaction) and juxtracrine signalling complexes that have not previously been described in either model organisms or human specimens.

7. For Fig 4i-s, they discuss the combinatorial representation of the proteins they interrogate and state that all possible mixtures were observed. As stated, this implies that many of the molecules studies are absent from many of the cells. This is of course dependent on the signal to noise ratio in the image, the extent of expression, and the thresholding they apply to the imaging data. It is quite possible that cells they categorize as negative for a given protein have a low amount of the molecule(s) in question; there is an important biological difference between a complete lack of protein in a cell and a low level of that molecule, perhaps only at the particular moment of fixation of the sample. The authors should be more cautious in their claims about expression data of this type, especially when it is the lack of detection that is driving the interpretation.

The reviewer is of course correct. We have added a note in this section that points readers to our methods section for more details about how channels were gated (see '*Single Cell Phenotyping*'). As stated above, we have also made Z projections of all images available online (see links in **Supplementary Table 1**) and have added a new **Supplementary Note 1** (described below) to address the role of signal to noise. Finally, we now explicitly described cases in which different marker intensities vary in a continuous rather than discrete manner.

Nonetheless, immunophenotyping using standard methods such as flow cytometry, primarily relies on positive and negative gating. Immune cells in particular are characterised based on a set of binary criteria such as CD3-positive, CD8-positive, CD4-negative, PD1-positive etc. Thus, the reviewer's comments pertain to the entire field of immunology. Also relevant to this discussion is the fact that high-resolution imaging has substantially better signal-to-noise (SNR) than low resolution imaging (and all extant approaches to high-plex spatial profiling) and that punctate signals can be much more accurately quantified. Thus, gating problems are less severe in the images we acquire than in most other lower resolution lower SNR methods.

8. They describe poor antibody penetration in S fig 15 and speculate about the origin of the problem. It is worth noting that they are seeing this problem with highly abundant targets such as SMA. Others doing volume imaging at varying specimen thickness have reported such a problem and it can be ascribed in at least some cases to rapid depletion of free antibody by the targets in the more superficial regions of the sample, leaving too little antibody to give adequate staining in deeper areas. This gradient effect is enhanced by the high intensity of the surface layer staining that leads the imager to attenuate the laser power to avoid signal saturation, thus reducing signal from whatever antibody actually makes it to deeper regions. There can be other explanations of course, including the fluorochrome issue discussed, but this removal of antibody from the staining mix is worth considering, especially if the antibodies are pre-titrated and diluted according to thin section controls.

We thank the reviewer for raising this point (which is also raised by reviewer 2) and agree completely. We have added this to the methods section, " Optimization of sample thickness and antibody staining protocol." We have also added a new **Supplementary Figure 17** that demonstrates how antibodies penetrate the tissue over time. As the reviewer may also have experienced, it is possible to improve penetration using various denaturing reagents. We are exploring these reagents and anticipate that it will be possible to use an increasingly wide range of antibodies with thick tissue sections.

9. In Fig 4o, I am confused – the magenta label says PRAME, but the cells in magenta largely correspond to those with SOX9 and SOX10 staining in panel m yet they do not appear in the conjoint green color assigned to these proteins in panel o. Perhaps I misunderstand how they have processed the data since some cells have all 3 markers.

We thank the reviewer for pointing this out. In retrospect, we realise that our labelling convention was confusing. The magenta is indeed PRAME and is consistent with the PRAME channel in Figure 4i and 4k. However, the green label indicates cells dually positive for SOX10 and SOX9. Likewise, the data in Figure 4i and 4k indicate cells that are positive for only SOX9 or SOX10. We apologise for the confusion and have clarified this in the label on Figure 4i, 4k, 4m, 4o, and 4p, and the corresponding figure legends.

Biology:

1. They describe Langerhans cells as "...the dendritic cells of skin". This is no longer considered to be true. These are embryologically derived macrophages – see Nat Rev Immunol 2014 Jun;14(6):392-404.

This was sloppy writing on our part. We have corrected this and thank the reviewer for pointing out this error.

2. In discussing the data in Figure 3, they make claims or suggestions about the direction of migration of T cells based on static data. In truth, the imaged cells could have been moving in any of many different directions within the TME and have just been caught in one place and one polarized orientation at the moment of fixation. Live imaging shows such diverse directions of movement of T cells within inflamed tissues and to a lesser degree tumors. I urge the authors to be

more cautious. As to the points about tension, the tumor core, and the margins, the section is oriented in only one direction across the full tumor volume and hence, it is unknown if these patterns would be seen in the orthogonal extremes of the tumor; further, this is a single sample from this particular tumor, so how representative the features are cannot be evaluated. Finally, details of tumor invasion have been visualized by Friedl and colleagues and should be cited (one recent review is *Trends Cell Biol* 2023 May;33(5):388-402; technical matters for such imaging are in *Elife*. 2022 Feb 15;11:e63776).

The reviewer is objecting to our use of migration to describe lymphocytes in tumour (as in “migrated into the tumour core”) when we should have used a more neutral and passive term such as “were infiltrated within.” We did not mean to imply that we could discern knowledge about migration patterns from a single image.

We have elected to remove these panels from Figure 3 in order to make room for more methodological detail and meet the word count limit of *Nature Methods*.

3. In the section on inflammatory neighborhoods, they describe responses to what they infer is local interferon production. Such measurements have been made in ref. 47, noted later in the text, but also in *Nat Cancer* 2020 Mar;1(3):302-314 and *Cancer* 2023 Jul;4(7):968-983 among other studies from Bouso and colleagues.

Thank you for the suggestion. We were aware of these papers and have collaborated directly with these authors; we have now added references to the Oyler-Yaniv paper (prior ref. 47) and *Nat Cancer* 2020 Mar;1(3):302-314 at this point in the text.

4. In the next section on T cell subsets, they conclude based on staining that there is substantial overlap at least in phenotype among Tpex, Tmem, and Teff CD8 T cells and that this is different from what is generally believed, but such overlapping marker protein expression is commonly seen and noted in scRNAseq data and especially flow cytometry in both tumor studies and the classic clone 13 LCMV model.

This is an interesting point, and we have revised our text and added some references. The clone 13 LCMV model is of course an acute infection model, and we agree with the general thrust of the reviewer’s comments here and below that tumour biologists are insufficiently aware of some of what is known about infection. More specifically, we have cited <https://www.nature.com/articles/s41590-020-0760-z> in this regard. Of note however, this paper and the others we have found on the LCMV model use flow cytometry and dissociative scRNAseq, not spatial immunoprofiling. Regardless, the reviewer is correct that we should have described these data without making any claim to novelty.

5. They state “Tpex, Tmem, and Teff cells were also highly proliferative (40-80% PCNA or Ki67 positive) demonstrating a high degree of activation and self-renewal.” This is an over-interpretation of the data. Other work has shown that the cells they define in these categories can/do have a lineage relationship and hence, if the Tpex divide, their progeny will have the PCNA or Ki67 staining they see even if these progeny cells are not continuing to divide. Suggesting that Teff have a capacity for self-renewal also runs against many observations in the literature that show these Teff to behave as terminally differentiated cells. Better phrasing and a more cautious interpretation of the data are called for here.

This is an interesting point and we have revised the text to state that cells expressing PCNA or Ki67 may have recently divided (as opposed to being about to divide). Of course, we agree entirely with the reviewer that the presence of a proliferation marker in a cell does not imply that it is poised for long-term proliferation as part of a self-renewing immune compartment. However, work has been shown that Tpex can be highly proliferative, and we now cite such work (see doi: 10.1016/j.cell.2023.02.021).

6. Likewise for the suggestions of three branches involved in T_{pex} giving rise to T_{mem} and T_{eff} cells. The data they show can easily fit the commonly accepted model of T_{pex} giving rise a stem-like daughter and what is called here a T_{mem} cell that can further differentiate into a T_{eff}. It is hard to see how the very limited data they provide on phenotype and position provide a more definitive answer to the lineage issue than scRNAseq velocity or pseudotime analyses or experiments with cells transfers of isolated T cell populations.

We agree completely with the reviewer on this point and did not intend to imply otherwise. We have reworded this section of the paper to emphasize interactions and spatial positioning rather than implying a novel discovery about lineage.

7. In the membrane to membrane section, they suggest they have observed relationships distinct from what is in the literature - "Type II interactions between CD4 and CD8 T cells were common across the MIS, without evidence of a nearby antigen presenting cell (e.g., dendritic cell), making this an atypical interaction with respect to current understanding of T cell biology." It is unclear to me why they make this statement. T cells can contact each other in a complex tissue environment but in the absence of a pMHC ligand for the TCR on at least one of the interacting T cells, upregulation of integrin affinity needed to make a firm (that is, tight) membrane-membrane association will not occur. Transient cell-cell contact can occur but the negative charge of a typical cell membrane will prevent close leaflet binding unless strong bonds are formed that for T cells typically involve LFA-1 and ICAM or VLA and VCAM. They also ignore the substantial evidence of antigen presentation by T cells to each other in such an environment, so considering this to be an interaction devoid of an antigen-presenting cells is simply incorrect. This is especially cogent with human tissues where the activated T cells can express MHC class II molecules in addition to MHC class I molecules.

Again, we have no disagreement with the reviewer on this point and are not sure what has raised their ire. We do see many tight CD4-CD8 associations without a proximate APC (as a statement of fact) but we **do not** mean to imply that this changes our view on well-established mechanisms of antigen presentation. We will have modified the manuscript to make clear that we are describing the organisation of the TME, not a mechanism of immune regulation.

8. The statement "One state missing from our primary melanomas but present in metastases is an NGFR-high state, which has been described as neural crest (stem-like) and tumour-initiating. Thus, high cell plasticity may precede the appearance of stem-like states rather than derive from them." is not supported by the data, given that only two sections were studied from a single tumor of this type. Without more data on additional sections and additional cases with the same tumor, such a conclusion cannot be drawn.

We have previously established this feature of early melanomas and have now cited that earlier evidence (see doi: 10.1158/2159-8290.CD-21-1357).

9. The authors write - "Moreover, we find that individual immune cells are often subject to opposing regulatory signals: a single cytotoxic GZMB+ CD8 T cell can be polarized toward a tumour cell, enveloped by filipodia from a CD4 helper cell, and repressed by a PDL1-expressing myeloid cell." This statement involves a misunderstanding of how PD-1 works. If a T cell has its immunological synapse containing the TCR polarized towards an antigen-bearing tumor cell, PD-1:PD-L1 interactions with a DC away from that synaptic region will actually decrease the inhibitory effect of PD-1 because such inhibition requires colocalization with the TCR to mediate a repressive effect on local phosphorylation events via SHP phosphatase recruitment. The cell interactions described would thus NOT include a repressive interaction with the DC for this reason.

We have collected new data on this topic and now illustrate cases in which CD8 T cells are directing granzyme B particles to tumour cells, with simultaneous PD1-PDL1 interactions occurring within a few microns of the point of membrane-membrane interaction. The CD8 T cells are also interacting

with helper T cells and dendritic cells. We think these types of multi-dentate interactions make our point, although we obviously cannot demonstrate the functional consequences of opposing regulatory interactions in fixed tissue. In multiple cases, these interactions co-occur within 3-5% of the membrane circumference. No our knowledge, there does not exist sufficient literature from functional (in vitro) studies to determine whether this constitutes a local interaction or not. Regardless, we have rephrased this section of the manuscript to make the conclusions more speculative.

REFEREE #2

(Remarks to the Author):

The authors use cyclic imaging (long established) in a tissue-clearing method using confocal microscopy. They use 5um sections as their comparator for advances despite clearing and multiphoton and other methods for seeing deep connections (see for example beautiful images produced by neurobiologists of neurites projecting and interweaving hundreds of microns into tissue.

The difficult part of this paper though is that, although some work seems of interest, it looks like a compendium of factoids and is hard to see a thread besides the imaging method. I also found many cases where I questioned the thresholding/surfacing in Imaris which can be tuned in many ways to smooth data and which I find difficult to accept as definitive.

I would kindly recommend the authors to submit this to a methods journal. The advance of putting Cytif together with clearing is hard to be excited about given the other excellent methods. With that, focusing on one or a few biological processes might make a study that is more compelling.

Thank you for this suggestion; we have reworked our manuscript for consideration at *Nature Methods*. We have also made significant improvements to the manuscript based on reviewer comments and added web resources with more methodological detail see protocols.io [<https://dx.doi.org/10.17504/protocols.io.261qe59m7g47/v1>].

REVIEWER #3

(Remarks to the Author)

Yapp and Nirmal et al. demonstrates use of scanning confocal microscopy for cyclic immunofluorescence in archival human tissue samples of 30-40 µm range, which makes it possible to image 2-3 cell layers as opposed to the more standard and high-throughput 5 µm thin sections. It illustrates the potential of this approach for precise phenotyping of cellular and subcellular morphologies and cell-cell interactions.

As is, the paper very nicely underscores the potential of implementing multiplexing with higher resolution and 3D imaging. The generated (very rich) data that encompasses five tissue types and 11 datasets is impressive and can be analyzed in many different ways, however only three datasets are utilized for the current manuscript. For these 3 specimens, observational instances from the imaging data are highlighted in relation to 5 topics: (i) cell and organelle morphology, (ii) the microanatomy of cell clusters and structures (e.g. blood vessels), (iii) cell shape, (iv) tumour and immune cell lineages, and (v) cell-cell interactions (proximity analysis). Renderings are beautiful and interesting observations are highlighted with potential functional implications based on literature. However, the paper reads almost anecdotal due to the main focus being on snapshots of different cases in the format of vignettes. The highlighted cases are definitely interesting, but the usability of the information is not immediately clear without quantitative analysis of more instances or backing up the observational statements with more objective metrics. For the broad Nature readership, the various highlighted instances may be rather specific cases, which would be of

interest to multiple different expert groups, though in some cases they may be considered speculative due to lack of quantitative and comparative analysis. The proximity analysis is the most interesting usecase however, as is, it is only scratching the surface of the potential utility of the presented approach.

We thank the reviewer for these comments and agree that the manuscript as written is more suitable for a venue other than *Nature*; we are therefore pleased that the paper is under consideration for *Nature Methods*.

I think the paper could have come across very differently if the authors follow-up some of these vignettes, and quantify the significance of the highlighted observation at the level of whole tissue sections and compare conditions. Or alternatively do more rigorous technical assessments to more quantitatively show the impact/worth of their approach in terms of how it may alter the results of the analyses typically done to infer cell types/states/frequencies, spatial niches/neighborhoods etc. From a methods perspective, an important aspect of the manuscript would be to enable others to utilize a similar 3D approach by presenting the necessary optimizations and guidance on imaging parameters and the analysis pipelines for the different use cases presented. However, in the current format of the manuscript, this does not appear to be the main highlight of the paper either.

Overall, the manuscript can be considered as a mix of a methods paper and some potential biological insights for selected uses cases based on literature, though not sufficiently going in depth in either direction.

As described in greater detail below, the revised manuscript includes many more technical assessments of differences between 2D and 3D imaging. We have also significantly extended the method so it works with a wider range of tissue, and we have dramatically expanded the methods section.

More specific comments:

1. For better readability, the markers that are used to call the cell type (melanocyte, epithelial, tumour cell etc. should ideally be marked for each panel).

We thank the reviewer for this suggestion and have done our best to include this information on the figure panel or in the caption (when space does not allow). For the CD8 T cell subtypes discussed in **Figures 5 and 6**, we have included just the abbreviated calls (TPEX, TEFF, etc.) in the figure panels. Since each of these subtypes is defined by several markers, we felt it would be overwhelming to list all markers repeatedly. We have also referenced **Supplementary Figure 13** (our figure depicting how markers map to cell types) so that readers will easily know where to look for cell type calling information. It is challenging to accurately describe a 50-plex image in a compact manner.

2. The quantification data for the following statement should be included in the paper (for $n > 1$ FOV). Actually the more useful information would be the estimate of what percentage of a typical cell is captured (which could be even done for a few different cell types that the authors are highlighting, to further show cell-type dependent the bias): "This revealed that, in conventional 5 μm sections, nearly all nuclei (>90%) were incomplete along the Z (optical) axis whereas, in 30-40 μm sections, 60-80% of nuclear envelopes were complete along all three axes (Fig. 1a; Supplementary Video 1)."

We thank the reviewer for this excellent suggestion. We have added quantitative data regarding the percentage of cells captured within different tissue thicknesses (and the percentage of cell volume missing) to **Figure 1e** and **Extended Data Figure 1i-j**. We have also revised all of the thicknesses in the manuscript given our recent understanding of the effects of antigen retrieval and hydration on tissue thickness.

3. It would be great if the authors were actually able to do automated cell type inference from the 3D version 5 μ m section data for the WSI and made a quantitative comparison of the cell types (like partially done in Extended Data Fig. 1b - but both for 3D and 2D images). “When we created sequential synthetic 5 μ m serial ‘sections’ (2D maximum intensity projections; labelled I to V) along the Z axis, inferred cell types varied with the section and was often incorrect (Fig. 1d).”

We thank the reviewer for this good suggestion. Fully automated cell type calling in 3D is currently beyond our capability, but we have now provided in **Figure 1e** a quantitative comparison between virtual thin sections at 35-micron sections for polarised immune markers such as LAG3, MX1, and combinations thereof. We observed that over a third of cells would have been falsely designated negative for each of these markers if the imaging were to have been performed on thin sections. This is less pronounced for non-polarized markers (e.g., CD103+ and CD8+), however, we still observe 5-15% undercounting of these cell types with thin sections. Of note, SOX10 represents a least undercounted scenario, likely because SOX10 is a nuclear marker. We believe that this will substantially address the reviewer’s concern that 3D imaging of thick samples is needed for classifying immune markers that are sensitive to spot counting.

e Percent of misclassified cells in virtual 9-micron thick section

4. The showcase of the Lag3 almost feels like a missed opportunity, as there is a lot more analysis that can be done regarding this. For example clustering the cells based on other markers and then comparing the LAG3 spots per cell.

Thank you for the suggestion. We have these data and have now added them to **Extended Data Figure 4b**. This will allow us to compare spot distributions for LAG3 relative to other proximal cells.

5. “In the MIS as a whole, we found that B cells were the type of cell most likely to associate with collagen fibres in the dermis^{20,21} (Fig. 3b, c); they were often (n = 11 of 14 in the MIS) stretched into irregular shapes (this was not a feature of all B cells and those found in the stroma were often round;

Extended Data Fig. 3d)”. The data that supports the second part of the statement is not featured or accessible in a quantitative manner, as no analysis result is shown. Fig.3c: n numbers are missing.

We thank the reviewer for pointing out these errors and omissions. We have quantified B cell sphericity and volume between the melanoma in-situ and invasive margin and provide numerical evidence for our assertion that B cells are more elongated in the melanoma in-situ. This can now be found in **Extended Data Figure 3e** (and pasted below). We have added numbers to **Figure 2c** (formerly **Figure 3c**).

e Sphericity and volume of B cells in MIS and VGP

6. “Cell shape and motility are regulated by the actomyosin cytoskeleton which was most prominent in

dendritic and T cells (~2-fold more actin overall than melanocytic cells and 3-fold more than in keratinocytes; Fig. 3d).”

– Here again the quantitative data behind this statement and numbers is not shown anywhere.

Similarly the rest of the statements for Fig. 3e-f do not include any quantification for the reader to evaluate the significance of the observations that are highlighted.

Thank you for this suggestion. We have chosen to remove this section to make room for more methodological detail and to conform to the word limit for Nature Methods.

However, we have calculated the mean intensity of b-actin across different cell types and have provided them here by way of addressing this question.

The mean intensity of b-actin for different cell types in the melanoma in-situ

In certain cases like Extended Data Fig. 4b, based on the color gradient it seems like a complex lookup table (like fire) was used instead of red in multicolor overlay images (without an explicit explanation). I cannot be sure if this is the case, as the legend only says red, but if this is the case it

is very confusing for interpreting the images and actual overlaps.

(Furthermore, in certain cases almost the same colormap is used for different protein targets - Supplementary figure 2, row 2 column 1 for example. CD8 and PDL1 are colored using almost the same colormap.)

Thank you for catching this. We used a LUT for the overlay and forgot to define it appropriately.

However, we have chosen to remove this section to make room for more methodological detail and to conform to the word limit for Nature Methods.

7. Extended Fig. 5 Panel c - this would be good to also do on virtual thin sections, to see the differential impact of the 3D approach.

We are unsure precisely what the reviewer is proposing. **Extended Data Figure 5c** (now **Extended Data Figure 6c**) is intended to show that there are MART1+ tumour cells breaking through the DEJ - this is an observation that we might expect to find on conventional thin sections.

We refer to panel 5c at the same time as panels 5b and d in the text, so the reviewer may be asking whether virtual thin sections would change the quantification in Extended Data 5b, regarding the distance between MHC-1 expression and cells containing MX1 puncta. We expect that there would be a significant difference between 2D and 3D for this quantification. This has two causes – first, as shown in **Figure 1d** and **Supplemental Note 1**, lower resolution imaging and the inability to reject out of focus light significantly reduces our ability to quantify punctate signals compared to the high-resolution approach. Additionally, as seen in our new **Figure 1e** and **Extended Data Figure 1i-j**, thin sections significantly reduce the portion of the cell captured within an image, and make it difficult to phenotype cells as MX1 positive.

8. Nonetheless, the MIS and underlying dermis were not highly proliferative, with only 1% of cells (n = 110) positive for the Ki67+ proliferation marker.

Among Ki67+ cells, 34% were T cells while the remainder consisted of monocytes (28%) and endothelial cells (2.7%); only a single melanocytic cell was Ki67+. By contrast, in the invasive VGP melanoma domain from the same specimen, 11% of all cells were Ki67+ with melanoma tumour cells the most proliferative (45% Ki67+), followed by monocytes (44%). "It is unclear which image data is the source of these statements as no figure is cited.

We thank the reviewer for pointing this out. The data refers to dataset 1. We have included the data as a graph in **Extended Data Figure 5a** (and pasted below).

9. "These data imply that melanocytic cells with features of early malignant transformation are subject to frequent changes in cell state (phenotypic plasticity) rather than progressive evolution from a single transformed or progenitor (stem-like) cell, as proposed for advanced invasive melanoma." -- The suggested implications is rather speculative as progressive evolution may happen at the genomic and epigenomic level without being apparent in the visual phenotype.

The reviewer is of course correct. We have modified the text to make clear that our conclusion is based on a limited set of markers, albeit ones that are widely used diagnostically. We believe the data support a conclusion of frequent change of state, but they say nothing about the origins of these changes (genetic or epigenetic). The new text reads “*These data imply that phenotypic plasticity is a feature of melanocytic cells undergoing the earliest stages of malignant transformation and that invasion into normal tissue can occur in the absence of the stem-like NGFR⁺ cells found in metastatic melanoma.*”

10. Fig. 4s: although the analysis performed here is interesting, based on n= 875 cells that is written in the main text (which should also be included in the legend, with the information that it is from one tissue section), It seems like cells in the different frequency bins were not counted exclusively (i.e. the total for the different columns add up to much more than this number) or the given n number is not correct.

We struggled with the presentation of this data. Our original form did not count cells exclusively, which may have been confusing to readers (it was also a concern of review 1). In the original plot, black represented “not specified” markers, i.e., the given marker could be either positive or negative. We have revised this plot (formerly **Figure 4s**, now **Extended Data Figure 5b**) to show specific positive/negative states for all six markers. We believe this is more useful for readers, as it better highlights the specific proportions of various states of interest. We have also redone the images to make the different states more visually intuitive. We trust that this helps.

11. The criteria for inferring IFN+ in Fig. 5a-b are not really clear, as the panel shows both correlated and uncorrelated expressions/distributions for the 3 markers (panel b). For panel a, can the authors rule out that these spatial domains are not connected in z (would imaging even thicker samples be necessary for this), since there appears to be smaller domains in between depending on how a domain is specified?

The reviewer makes a good point. It would be interesting to explore whether the IFN-positive domains are connected in three-dimensions. We have previously shown that other tumor features (e.g., mucin pools and tumor ‘buds’ in CRC are part of larger, interconnected 3D networks; Lin et al., Cell, 2023). As you suggest, this would require thicker specimens, which is a limitation of our current work. We have mentioned this in the discussion as an interesting question to explore in future efforts. We have toned back our language in this section to make it clearer that there is further work to be done to definitively characterise these IFN-positive domains.

12. Extended Data Fig. 5e - does this panel also include PCNA staining to show the PCNA negativity (according to the figure caption it should, but the label on the figure does not include it)?

We thank the reviewer for pointing out this error. PCNA was included in this panel and is negative. We did not previously include PCNA as we thought it would obscure other markers. We have added PCNA into **Extended Data Figure 6d** (formerly **Extended Data Figure 5e**).

13. T cell subpopulation/lineage and tumor proximity quantifications in Fig.5 and Extended Data Fig. 5 are some of the most useful results, as they provide more quantitative insights. A very relevant question here is how much of this information can be captured with typical thin sections, to see the impact of the 3D imaging on detailed cell type calling and quantification. If the authors had a dataset from a more typical 5 μm section with widefield imaging for doing this type of comparison from a consecutive section of the same sample that would have been the ideal way to show the comparative advantages of the approach in this paper over the more standard approach. As a worst case, the comparison can be done on virtual sections of the given dataset as in previous parts of the manuscript (which would also help to see section to section heterogeneity).

We thank the reviewer for this suggestion. We note that proximity in the current work is scored based on membrane-membrane association rather than nuclear location. We have in the past

attempted to repeat distance/proximity metrics on membranes using the more common widefield imaging approach sampled with a 20x/0.75NA lens. The sampling is often too coarse, and the out-of-focus blur is too great to give confidence about cell contact. To address this point in another way, we have collected a proximate 5 μm section using conventional 2D imaging approaches and generated synthetic 5 μm sections. These are shown in **Extended Data Figure 1** and illustrate in a quantitative manner that the majority of cells will be incomplete in traditional 5-micron sections. Moreover, using synthetic data we can prove that neighbourhood interactions are significantly undercounted in thinner sections.

14. The proximity analysis to infer cell-cell interaction modes is definitely an informative and interesting analysis that leverages the resolution of the data and a potential highlight of the paper. It actually creates a great opportunity to guide future studies on the imaging parameters that should be taken into account to be able to perform such analysis in a technically sound way, however relevant technical assessments and discussion is missing in the paper.

We thank the reviewer for this excellent suggestion. We have extensive detail on this point in the **Methods** with accompanying **Supplementary Note Figures 1.3-1.5**). We specifically examine the impact of signal-to-noise ratio, sampling rate (pixel size), numerical aperture, averaging along membranes, and Z-stacking on the ability to characterise cell membrane interactions. We also include line profiles and confocal images for each example.

Fig. 6 - The juxtaposition quantifications with distances going down to 60-70 nm, raises relevant questions about the technical error margin. What is the measured registration error margin across cycles and fluorescent channel in 2D and 3D- when potential imperfections are taken into account what is the confidence level for the subpixel scale quantifications?

Based on the analysis, the authors are encouraged to discuss what would be the minimum specs (pixel size, resolution, z-step size etc.) to generate the data that would be permissive to proximity analysis to infer all 3 interaction types.

How should the analysis be ideally done: on one or multiple single planes, on projection images? Is it necessary to look at the whole interaction surface in 3D or are the contacts only visible in a single plane? It is very difficult to understand and the line profiles done in Fig. 6 should at least be also performed along the z axis to reveal the distances in 3D, within the limit of the confocal microscopy and acquisition settings used.

These are all excellent questions. There is nothing unusual about inferring super-resolution features from images, but proximity calculations are dependent on signal-to-noise ratios and registration accuracy, as the reviewer points out. As stated above, we now include a detailed discussion in the methods and new **Supplementary Note Figures 1.3-1.5** on cell-cell interactions. The note compares these factors using experimental data and how they influence proximity calculations.

Axially, we sample at the Nyquist limit, but laterally, we have chosen to sample at about double the Nyquist limit (140 nm vs 80 nm) to reduce data size and increase throughput. In the revised paper, we demonstrate that under sampling laterally can still robustly detect cellular interactions down to ~50 nm (subpixel accuracy) using curve fitting approaches. This is assuming that signal to noise is optimal through the use of optical sectioning (confocality vs widefield) and using optimized antibodies. We also examined how, in noisy data, the use of different types of line profiles orthogonal to the membrane and projections across multiple z planes can have a beneficial impact on identifying membranes. Finally we cited: DOI: [10.1091/mbc.e17-01-0069](https://doi.org/10.1091/mbc.e17-01-0069) as a relevant citation for measuring distances.

Finally, we have manually identified regions of poor nuclei segmentation between each cycle by overlaying Hoechst channels and looking for overlap. We have excluded areas that have poor registration for proximity analysis as well as cell type calling. These errors affect a small minority of the data.

Methods:

15. The controls that were performed to find the optimal conditions and potential limitations are highly appreciated (Supplementary Fig. 13-17). However, these very important technical considerations including optimal section thickness, effect of fluorophore choice, necessary alterations of the sample preparation and CyclIF protocol for thicker specimen as well as important processing details (registration, segmentation), which are some of the most challenging parts of dealing with this kind of data are buried in the Methods section and in the Supplementary. These should at least be discussed/mentioned and cited in the main text.

This is an excellent suggestion. In response, we have prepared a complete supplementary technical guide and on-line protocol at [protocols.io](https://dx.doi.org/10.17504/protocols.io.261qe59m7g47/v1)

[<https://dx.doi.org/10.17504/protocols.io.261qe59m7g47/v1>]. We have also added to the methods and added additional Supplementary Note figures in the hope that these can be more easily understood. Finally, we have developed a new alternative approach to 3D imaging that overcomes some of the limitations encountered when working with fragile tissues; this approach is shown in **Extended Figure 1**.

16. Regarding the antibody penetration heterogeneity between different antibodies and conjugates - previous works (for example: Murray et al, Cell 2015, doi: 10.1016/j.cell.2015.11.025) suggest that an important factor may be the distribution of targets along the diffusion direction (i.e. if the target is highly abundant in the surface layer, the effective concentration of the antibody may get lower or the better-binding conjugates may be depleted as it penetrates through the sample causing ineffective staining in the lower layers. And in that case slower/less efficient binding might allow better penetration and more consistent staining along z). That may also partially explain the observed differences between fluorophores (i.e. if a particular conjugate yields a lower ON rate, it may diffuse better before binding to the top layer very effectively).

As mentioned in our response to reviewer 1, we agree completely with this conclusion. We thank the reviewer for the citation and have included it in our revision. We have also expanded our discussion at this section of the methods and added a new **Supplementary Figure 17** that illustrates the antibody penetration over time.

17. Regarding the CyclIF protocol, were any other changes introduced with wash times etc? What temperature was used for 8-10 h antibody incubations?

There were changes to the basic protocol and we have included all of this information in the online protocol, as mentioned in the response to point 15. Among the changes, we perform an 8-10-hour antibody incubation at room temperature. This time was determined experimentally. We now show a staining time lapse in **Supplementary Figure 17** to further explain this optimisation.

18. For the MATLAB functions, authors should include the parameters used, when applicable.

We thank the reviewer for pointing this out; we have added MATLAB code and parameters to the GitHub Repository (<https://github.com/labsyspharm/mel-3d-mis>).

19. The Supplementary Discussion section is very helpful. I would also recommend some more references to be included there for completeness (to the part: "While spectral unmixing, spillover compensation, or the use of Raman dyes can increase this number to 8-10 channels^{71–73}, to date, sequential staining and high-plex high-resolution (sub-micrometer scale) imaging of thick sections has not been demonstrated."). Although a direct demonstration of sequential staining and high-plex high-resolution (sub-micrometer scale) imaging of thick sections for FFPE samples has been missing, there are many efforts along this way, which should ideally be cited, including:

-Joha Park, Sarim Khan, Dae Hee Yun, Taeyun Ku, Katherine L. Villa, Jichen E. Lee, Qiangge Zhang, Juhyuk Park, Guoping Feng, Elly Nedivi, Kwanghun Chung (2021) Epitope-preserving magnified analysis of proteome (eMAP). Science advances 7.46 (2021): eabf6589.

- ExM- exchange-based multiplexing combination: Saka, S.K., Wang, Y., Kishi, J.Y. et al. Immuno-SABER enables highly multiplexed and amplified protein imaging in tissues. Nat Biotechnol 37, 1080–1090 (2019). <https://doi.org/10.1038/s41587-019-0207-y>
- Taeyun Ku, Justin Swaney, Jeong-Yoon Park, Alexandre Albanese, Evan Murray, Jae Hun Cho, Young-Gyun Park, Vamsi Mangena, Jiapei Chen & Kwanghun Chung (2016) Multiplexed and scalable super-resolution imaging of three-dimensional protein localization in size-adjustable tissues. Nature Biotechnology 34:973–981.

We thank the reviewer for providing this excellent list of missing references. We have modified this supplementary discussion and added the new references to the revised manuscript.

20. Code availability: the repository currently contains only matlab code, although other (python) scripts are mentioned in the paper. The github link is currently broken.

Most python scripts were derived from other published packages and we have made sure they are correctly and clearly referenced. Python scripts related to cell type calling have now been added to the GitHub repository (<https://github.com/labsyspharm/mel-3d-mis>) along with all MATLAB scripts and parameters (which were primarily used to generate figures). We will ensure that all links work appropriately and these scripts will also be referenced in the new technical guide.

21. Some of the preprint references are not updated to the published versions.

We thank the reviewer for pointing this out. Our manuscript has been in review for an extended period of time, and this may explain some of these errors and omissions. We have carefully reviewed our references and believe they are now up to date.

NMETH-A57303A

Second Round of Review – May 5, 2025

"Highly Multiplexed 3D Profiling of Cell States and Immune Niches in Human Tumours"

PROPOSED REVISIONS

To address reviewer's comments, particularly those from reviewer 1, we will make multiple changes to the text and figures and include a substantial amount of new data and analysis in extended and supplementary figures. We believe that these changes will address all of the concerns raised by the reviewers and also improve the manuscript.

Revisions in response to comments from Reviewer 1

1. We will address **Comment 4** by reworking the first part of the results and discussion to more clearly describe why we believe 30-50 micron sections represent a true optimal thickness for imaging human specimens and not just a part-way solution on the path to yet thicker specimens. Specifically, we will describe how 30-50 microns optimized high-plex whole cell reconstruction at high resolution with preservation of clinical specimens; we will describe how image quality declines as sections get thicker.
2. We will extensively update the *Supplementary Discussion on 3D Tissue Imaging (now Supplementary Note)* to include citations and information relevant to **Comments 1-4** on the strengths and weaknesses of different types of 3D tissue imaging; we will add the additional references on tissue clearing
3. Also in response to **Comment 4** will include an example of an ~400 micron thick tissue specimen of the type preferred by the reviewer to *Supplementary Note on 3D Tissue Imaging* and use it to discuss the strengths of this type of imaging as well as the substantial optical challenges involved.
4. We will create a new Supplementary Note Figure consolidating all of the data we have on the variability of PD1 and PDL1 staining to address **Comments 5 & 6**. This will demonstrate a full range of morphologies from punctate to diffuse for both proteins, depending on cell type and microenvironment.

5. We will create a new Supplementary Note Figure showing how different representations of PD1, PD-L1 and CD4 on a single cell change how it appears in a figure. This will include a single-section view, a maximum intensity projection of a 9 micron (5 micron as cut) virtual section, and several 3D renderings of the same data. This will address **Comments 5& 6** that had suggested that we have unnaturally clipped images to change the way the data looks.
6. We will generate supplementary figures with multiple image panels that cover all markers used for main figure panels of concern to the reviewer in **Comments 6 & 7**; this will demonstrate how cell types were assigned using markers not currently shown in the main panel.
7. We will requantify LAG3 expression levels (number of puncta) using different definitions of exhausted T cells (Tex) from the literature to address **Comment 7**. We will specifically address the point that previous reports do not stratify T cells based on proliferation, granzyme B status, exhaustion markers, and CD103 as we do in our manuscript.

Revisions in response to comments from Reviewer 2

1. We will update the sections of the methods and supplementary materials to better describe how tissue shrinkage and expansion due to dehydration and hydration affect specimen geometry,
2. We will update the description of the holder used for imaging specimens mounted to coverslips to clarify which components are important optically.
3. We will revise the readme file in our GitHub repo to make our code easier to understand and use.

DETAILED RESPONSE TO REVIEW EDITORIAL COMMENTS

1. Regarding the use of 3D, we appreciate both sides of this argument. I would ask that you avoid using the word volumetric, which I do think implies larger volumes beyond single slices, and clarify the type of 3D imaging being accomplished early on in the paper (abstract even).

We have made this change.

REVIEWER ONE

1 General Comments:

We appreciate the length to which Reviewer 1 has gone to refine and reiterate concerns raised in the initial round of review. In response to the first round of comments, we made many changes to the data and its presentation, and we completely rewrote the results. The additional issues raised by Reviewer 1 can be addressed with textual changes and additional supplementary data. Before going into a point-by-point response, we would like to make four overarching points.

First, the reviewer's comments incorrectly assert as "truths" - points about microscopy and immunological data - that are nuanced and subject to debate. Many of the specific points are true in some contexts and not others, for example, being applicable to mice but not humans; in some cases the critique seems to be based primarily on personal opinion.

Second, the comments represent a highly selective and prejudicial reading of our data, for example, by criticizing image panels depicting single focal planes of 3D objects that are shown elsewhere in 3D, as supplementary movies, and as data available for off-line viewing in free Imaris 3D software. We agree with the reviewer that it is not possible for a single 2D image (in a multi-panel figure) to adequately represent the full biology of a 3D tissue section, but we believe that our illustration of specific morphological features of biological interest is a strength rather than a weakness.

Third, the reviewer fundamentally misrepresents tradeoffs in doing deep 3D imaging (on sections hundreds of microns thick) and sections ~50 micron thick. We have ourselves performed this comparison (see below) and additional insight can be obtained from a Germain lab pre-print describing imaging of murine tissue (Hor... Germain et al; <https://doi.org/10.1101/2024.08.02.606241>). As tissues gets thicker, resolution decreases,

chromatic aberration increases, and the number of cycles falls (for example, from ~50 to ~15 in our work compared to Hor et al). There is more than enough demand in the field for both of these approaches.

Fourth, in each of the two rounds of review, a long set of minor interpretative and technical concerns are raised, each of which would require large-scale reorganization of the figures to address. We do not believe that these complex and time consuming reformatting exercises make the paper clearer or more impactful, although they will further delay publication. We respectfully submit that this type of repeated rewriting is just not productive.

2 Mice vs. human studies.

Reviewer one fails to account for a significant difference in technical approach and interpretation between studies in mice and in humans. In mice it is entirely reasonable to access (and consume) complete specimens, to use a variety of different fixatives (some more sparing of antibodies epitopes than formaldehyde) and to perform parallel flow cytometry and tetramer studies (to measure antigen-specific T cells). While a feature of the *Hor... Germain et al* preprint, none of this is possible in humans. Specimens from all but autopsy cases must be preserved for future diagnostic use, these specimens are almost always FFPE (with frozen specimens possible in some settings and research protocols) making it necessary to work within the limitations of this type of preparation. Moreover, the immune contexture of barrier tissues such as skin is vastly more complex in humans than in mouse bred in nearly germ-free environments.

We believe that understanding disease and immunology in humans, not just in mice, is a worthy goal. However, a much wider range of low-plex 3D methods exist for murine than for human samples. For example in *Hor... Germain et al* the fixative was quite mild (Cytoperm), specimens were mounted in agarose, and parallel specimens were prepared for flow. None of this would be feasible in a human setting except under highly specialized conditions. Of greater significance is the fact mentioned above: human barrier tissues contain a much larger number of resident immune cells than laboratory mice, perhaps explaining the reviewer's concerns about our detection of abundant CD103-positive T cells in the human skin. We stand by all of the results in our study, even if they do not conform in all details to previous expectations from murine studies. We have not identified any serious discrepancy between our findings and those in the human cancer biology literature where the two overlap although this literature is much less cut and dried than the reviewer's comments imply.

3 Optimal section thickness for human specimens.

The reviewer assumes that the thicker the tissue section the better and more novel the approach. As a result, they assert that our specimens are too thin (at ~50 microns) to be of interest in comparison to 300 micron thick specimens (such as those used by *Hor... Germain et al*). However, this assertion ignores the fact that in many cases, research studies are simply not permitted to use more than ~10-50 micron of a human specimen – particularly when this constitutes a substantial fraction of a primary cancer. The remainder must be preserved for future diagnostics needs.

Moreover, ~50 micron sections containing >70% intact cells represent an optimum in terms resolution (120-250 nm vs ~500-1000 nm), antibody plex (50-60 vs. 10-15), and data set size (<1TB) for examining local neighbourhoods and intracellular features - as we illustrate. There is an important role for very thick section imaging, to trace nerves and the vasculature for example, but we find that these two approaches are complementary not competing. Given the very high level of interest in high-plex 3D tissue imaging, it is our very strong opinion that both the methods in *Hor... Germain et al* and in our current manuscript merit publication in high impact journals.

DETAILED RESPONSE TO REVIEW

Reviewer #1:

In this manuscript, Yapp, Nirmal et al. present a technique that extends the highly multiplexed CyCIF method to thicker 35um tissue specimens. A key strength of this new implementation of the technique is that unlike traditional FFPE section imaging, often restricted to 5-10um, the axial expansion of the image dimension now enables whole cells (1-2 layers of cells) to be captured in full and even allows subcellular protein localization to be visualized when the specimens are imaged at high resolution. However, despite showing an impressive array of high quality images of cell-cell interactions in patient tumor samples, this manuscript nonetheless suffers from a number of major shortcomings that would need to be addressed, including such factors as lacking novel improvement over existing methodologies, a need to go beyond a series of “anecdotal” examples in terms of data, missing key baseline controls, and claims that appear to be an over-interpretation of the findings, given the small number of examples and an absence of further experimental validation

Comment 1A: *Although the authors have done an admirable job of staining and capturing high plex image datasets from a series of cancer samples, a clear technical advance from existing methodology is not apparent..... much of the work achieved in this manuscript involved characterization of antibody staining of thicker samples using a very minor modification of the existing CyCIF workflow, rather than demonstrating novel improvements over existing techniques, whether this entail significant innovation in terms of hardware/software/analytical methods, a new combination of existing technologies, or provision of complete software packages that could allow researchers to automate such tasks with ease, as just a few examples of where substantial advances could be made.*

Response: Respectfully, this is simply not the case. Development of the methods in this paper required two (and more recently three) experienced microscopists four years of work. The staining protocols, analytical approaches, and different ways of mounting the specimen (now described in the methods and an **Extended Data Figure 1a-c**) have all been challenging to implement. One measure of this is that addressing the reviewer’s previous comments about methodology took us nine months. As a consequence, no other published paper, or paper on a preprint server, achieves anything similar to what we describe in the current manuscript with respect to image quality and depth of 3D analysis (note that the reviewer’s point about depth of imaging is addressed above and below).

Comment 1B: *In terms of quantitative methods, the computational analyses are provided mostly in the form of a loose collection of Python and Matlab scripts on the Github repository, as opposed to a fully furnished software package that could assist users/researchers to perform/replicate the various quantitative measurements performed by the authors in this work.*

Response: Our GitHub repository includes code required to reproduce the specific results in this paper. The major computational advance in this work involves a 3D image segmentation method described briefly in the methods and which has recently been accepted for publication in *Nature Methods* (preprint doi: [10.1101/2024.05.03.592249](https://doi.org/10.1101/2024.05.03.592249)) and which has its own GitHub repo (<https://github.com/DanuserLab/u-Segment3D>). This repo contains demo data, software, instructions, bug fixes etc. We cited the segmentation paper in the Results and Methods (section “Segmentation of individual 3D cells with Cellpose” - ref #21) but in a revision we will better describe the key findings from that paper to make it clear that readers should refer to the primary segmentation publication for computation approaches.

Comment 2: *For all the advantage of extending beyond the standard z-axis range into 35um thick specimen such that whole cells can be visualized, the metric the authors use to ascertain that they are imaging “true positive” cells is the presence of an intact nucleus, which comprises only a portion of the entire cell volume. It is unclear if the authors also filter out cells touching the axial boundaries of the image volume that have more or less intact nuclei but still have their cytoplasmic compartments partially trimmed in the collected image. Based on Extended Data Fig. 1i, the proportion of incomplete cells with 35um thick sections is ~20%. If so, what were the methods used to quantify the cells whose entire volume were not captured, and do the authors have an automated solution*

(e.g. software) able to perform this task. Given that the adoption of this technique is supposed to improve visualization and analysis of clinical samples, this appears to be an important omission.

Response: We apologize if the reviewer has misunderstood our strategy for identifying partial cells. The analysis in the paper was performed on whole cells (not nuclei). In our analysis, we define “true positive” cells as those with an intact cell volume, excluding any cells that contact the boundaries of the image volume; this is described in full in the Methods section. Given that cell volume and nuclear size vary across cell types, we emphasize the importance of accurate identification of both cellular and nuclear dimensions in cell segmentation. More importantly, in our 35 μm tissue sections, the vast majority of nuclei remain intact, minimizing the risk of misclassifying true cellular signals as background due to missing nuclear information—a limitation commonly encountered in current segmentation methods that rely heavily on nuclear localization to define cell boundaries. By leveraging high-resolution 3D imaging of thicker (35 μm) sections, we show that it is possible to capture a substantially greater number of “true positive” cells and associated signals compared to standard 5 μm sections, which often underrepresent the full cellular architecture. We will further refine the Methods text to make this point even clearer.

Comment 3: *The use of the terminology “3D imaging” can appear misleading to the readers interested in spatial techniques. Although it is accurate to call confocal imaging a “3D imaging” technique due to its optical sectioning capability, as distinct from widefield microscopy, the advances in optical tissue clearing methods in recent years that render transparent tissues with hundreds of microns and even of millimeter thickness have, for the most part, redefined what “3D imaging” means to the field. Many readers would typically associate “3D imaging” with these thick volumetric imaging of whole tissues/animals. One suggestion is for the authors to call the technique “whole cell 3D thin section CyCIF imaging” (or similarly phrased names) and specify explicitly in the abstract and the opening paragraphs the axial dimension of 35 μm when first introducing the technique. This would help distinguish the technique from the volumetric 3D clearing methods and avoid undue confusion.*

Response: We addressed this issue extensively in our previous response to review, as well as in a revised “Supplementary Note” with three accompanying figures (**Supplementary Note Figures 1.3-1.5**). We will add the references suggested by the reviewer to this supplement. We do not want to make a fuss about a purely semantic issue, but this argument about the meaning of 3D seems to be at the heart of the reviewer’s concerns about novelty.

We are simply bewildered by the reviewer’s position: from a purely technical standpoint- an image stack that comprises ~ 180 optical planes (along the Z axis) is a 3D image. Wikipedia (and various reviews) suggest that 3D microscopy comprises (1) Confocal Microscopy (2) Super-Resolution Microscopy (STORM, PALM, and RESOLFT), (3) Quantitative Phase-Contrast Microscopy, and (4) Light Sheet Microscopy. A quick scan of a subset of $\sim 2,500$ “3D microscopy” papers on PubMed suggests that about half involve cultured cells that are $\sim 5 \mu\text{m}$ thick, for which imaging 10-20 optical planes is typically required.

For example, a recent Nature Communications manuscript (*Three-dimensional total-internal reflection fluorescence nanoscopy with nanometric axial resolution by photometric localization of single molecules* <https://doi.org/10.1038/s41467-020-20863-0>) describes 3D imaging of specimens 250 nm - thick, 100-fold less than our paper. We can find no evidence in the literature that a distinction is made between 2D and 3D on absolute specimen thickness other than in a small number of papers that the reviewer requested we add to the manuscript during the first round of revisions.

Using “thin section” in the description of our method would also seem to us a mistake since the usual tissue section is 5 μm and the ones in our paper are 35-50 μm , which we think of as “thick section.” Calling millimeter-thick samples as “thick sections”, as suggested by the reviewer, is not

quite right either since this is on the size order of an entire “block”. We propose that our supplementary note is the correct place to address these technical and doctrinal issues.

Comment 4: *Furthermore, optical tissue clearing methods that have made possible the collection of much thicker image volumes should be explicitly discussed in the Introduction/Discussion sections and the authors should justify how and why the 35um thickness approach as presented in this manuscript would be superior/preferable in certain cases. That is, what are the key use cases for this version of “3D” imaging.*

This point is similar to that in comment 3 above, namely that clearing methods used in deep confocal microscopy or LSFM are superior/preferable to what we describe here. In response, we will add these references to our Supplementary Note.

However, the reviewer's comments do not correctly represent the fundamental strengths and weaknesses of very deep tissue imaging with clearing. These methods allow deeper imaging: but with fewer channels and substantially lower resolution (commonly ~1 um). In contrast, our approach to thick section (35-50 micron) imaging achieves lateral resolutions of ~225 nm (or ~110 nm using super-resolution mode). This arises from a fundamental feature of optical microscopes: the highest resolution is achieved using oil-immersion high numerical aperture objective lenses (~1.4 NA) and these have a working depth of <200 microns (including the ~170 micron thickness of #1.5 coverslip).

To better understand these tradeoffs we have ourselves imaged human samples as thick as 400 microns using clearing and the same confocal microscopes used for 50 micron thick sections (see **Figure 1 below**). These data demonstrate a requirement for long working distance lenses having lower magnification and lower NA, resulting in lower sensitivity and 5-10 fold lower axial and lateral resolutions. This directly impacts the image features described in this manuscript, including the appearance of PDL1–PD1 contacts, a topic of concern to the reviewer (see below). Specialized multi-resolution LSFM microscopes have been described in the literature, but they too suffer from the tradeoffs in working distance, resolution, and throughput.

Finally, we and others have found that specimens subjected to tissue clearing (which is necessary to make very thick specimens optically transparent) can be stained with a much narrower range of antibodies than specimens processed in the usual manner (as in our paper). Thinner specimens enable passive diffusion for the majority of antibodies (custom conjugated or commercial) without additional hardware that may not be widely available. That may be why we achieve >50-plex imaging whereas *Hor... Germain et al* are limited to <15-plex. Higher plex is relevant in this setting because it determines the refinement of the biological analysis.

Stepping back, all of these considerations seem to us to be of primary interest to the small community of investigators developing specialized next-generation microscopes. This community is sufficiently sophisticated to understand the impact of specimen thickness on optical performance without our adding further complexity to the manuscript. We have nonetheless added extensive additional data to our Supplementary Note 1 to assist readers interested in these topics.

Our work is intentionally focused on the much larger “spatial proteomics” community that wants a practical way to look deeply into cells and cellular communities using existing instrumentation. With this in mind, the approach we describe has the following key advantages (i) higher resolution (ii) higher plex (iii) tissue sparing for precious clinical samples (iv) scalability to multiple specimens using standard antigen retrieval methods (v) compatibility with human and not just murine specimens.

Figure 1: 3D confocal image of a cleared 40 micron thick specimen of normal human colon imaged in our laboratory on an LSM980 confocal microscope using a 20x 0.75 NA objective lens and sampled at 283nm (in x,y) and 550nm (in z) across 804 Z-planes. Channels are DAPI (blue) and beta-III-tubulin (orange). We are happy to provide this image as comparator to our 50-micron images as part our supplementary note of 3D imaging should the reviewers conclude that this is useful.

Comment 5. In general, many of the interesting features revealed by this 3D CyCIF method were presented in a variegated fashion and lack focus/organization, often providing selective/anecdotal cases with 1-2 examples of cells without clear comparison with the rest of the cell population in the acquired image. This manuscript could really benefit from a substantive reorganization in its presentation, for example in Fig. 2 and 3, by first showing the segmented/cell type calling of the entire image, followed by close ups of the interesting features that the authors would want to focus on. This would also help the readers determine whether the presentation of the examples was a recurring feature across the tissue image, or sporadic cases randomly chosen by the authors.

We have organized the manuscript so that UMAPs and other traditional approaches to analysing intensity-based features in high-plex images are thoroughly described in the extended and supplemental figures and the main figures focus on morphology, the key advantage of high resolution imaging. To provide readers with an unbiased view of the data we have made it downloadable for viewing in the free Imaris Image Viewer and we have provided supplemental movies covering all of the multicellular communities we discuss in the text. In almost all cases we provide both an exemplary figure and a quantitative whole-slide analysis (see for example the images in Figure 1c and the quantitative analysis in Figure 1e and in Extended Figure 1). We have adopted this approach because it is simply not possible to put all of these data in the main figures. However, it is not the case that we show only anecdotes.

With respect to the ad hominem comment that we show “sporadic cases randomly chosen by the authors” we can assure the editors and we have performed the most thorough and careful analysis of which we are capable. On the whole, we have left many subtle features undescribed. Thus, it seems highly likely that development of better computational tools will allow more aspects of 3D images to be quantified and compared.

Comment 6. *i) The localization of certain proteins as visualized by this 3D CyCIF method is not convincing. In Fig. 3h-n, the authors showed co-localization of PD-1 and PD-L1 between a CD8+ T cell and its surrounding MART1+ tumor cells. Although co-localization of receptor-ligand molecules typically indicate protein-protein interactions, it seems unlikely that PD-L1 localization is almost completely concentrated within the few small specks/puncta along the T cell-tumor interface as shown in the images (Fig. 3i, 3m). T CD8+ T cells can express PD-L1, so what is the evidence that the PD-L1 signal belongs to the MART1+ tumor cells, and if so, where are the rest of the PD-L1 signals, especially among the adjacent tumor cells, that were not in direct contact with the T cell? Or did the PD-L1 signal belong to the CD3+ T cell that had interacted in cis with the PD-1 expressed by the same cell?*

We agree that it is challenging to represent intricate 3D communities of cells in 2D figure panels. As a result, we show (and describe) how we use maximum intensity projections or single plane views to examine specific features, which we then contextualize in whole-cell surface renderings. Key points (such as intramembrane spacing) are quantified directly from images, and are not subject to limitations in visualization. Finally, we provide all 3D data freely downloadable and accessible through the free ImarisViewer for any reader who wants to confirm our findings.

To further address the reviewer's concerns we show below the data in Figure 3h-n (below) in another orientation chosen to show all z-planes in the image. This confirms the localization of PD-L1 in small foci at the cell membrane (PD-L1 - green; CD3 - yellow; tumor - magenta; dendritic cells - blue). We suspect that this aspect of PD1 PDL1 interaction has not previously been described because no one has previously reported imaging of human (or murine) tissue at high resolution. Thus the reviewer is correct that a 2D slice through a 3D cell will underestimate the number of puncta in the whole cell, but is incorrect in saying that PD-L1 is not both diffuse and punctate in the primary data.

ii) Similarly, in Extended Fig. 4h, the authors showed an image of PD-1/PD-L1 co-localization between a CD8+ T cell and a CD11c+ dendritic cell, but those signals do not appear on MART1+ tumor cells at all. Were the tumor cells negative for PD-L1, and if so, where did the PD-L1 expression in Fig. 3i and 3m come from? The authors did not provide any evidence that a dendritic cell (that also expresses PD-L1) is involved in the interaction in these two images.

We are unable to show all 55 channels of each image in a single figure panel and must therefore make tissue type assignments based on channels that are not shown. These assignments are made by 3D image segmentation and quantification, as is common in all spatial profiling studies. To our knowledge, it is not generally required that all channels also be shown for every cell in the main channels. Instead, when the information is critical, we show it in supplementary figures.

The tumor cells of concern in this case were clearly positive for PD-L1 however at lower signal intensity than dendritic cells; as a result, the PD-L1 signal is masked by overlapping MART1 on tumor cells. This overlap can be resolved by toggling channels on and off or, as in Extended Fig. 4h, by rendering some channels as transparent surface objects (also shown in the screenshot above).

iii) This also appears later in Fig. 6a/c where PD-L1 supposedly on a tumor cell and DC were only present as sparse puncta, and did not appear to be detected anywhere else along the membranes of those cells or their adjacent cells. Clearly it would be unlikely for most/all of the PD-L1 molecules expressed on those cells to be concentrated within just a few spots at the interaction junction. It would be useful to show and quantify the localization of PD-1 and PD-L1 expressing cells that are not involved in cell-cell interaction as controls and compare them with the examples as shown in this manuscript.

We are not sure why the reviewer thinks “*clearly it would be unlikely*” that PD-L1 is punctate since this is exactly what our data show. The panels in **Figure 4 (below)** show further examples of this data spanning several hundreds of microns of tissue. We also now address the complex issue of data visualization in the *Supplementary Note* and demonstrate that PD-L1 is punctate based on a variety of metrics.

iv) Otherwise these examples constitute “anecdotes” without a baseline for comparison, nor would they demonstrate that the imaging technique as presented is able to visualize membrane bound proteins (e.g. PD-1 and PD-L1) when they were not localized as discrete clusters/puncta. It is important to note in this regard that the spatial concentration of photon emission from such a puncta containing many antibody-bound molecules can lead to a non-negligible signal at the detector whereas a single molecule along the membrane would be bound only by a single labeled antibody and potentially produce too few photons per unit of the detector to be discerned from background, especially if the imaging parameters are adjusted to give sub-saturating signals from the puncta.

The reviewer is misrepresenting our findings:. We show that both PD1 and PD-L1 also exhibit diffuse staining across multiple specimens and we devote multiple panels in the main text and the supplemental note to describing differences between punctate and diffuse PD-L1 staining, which we speculate are biologically meaningful. The reviewer’s comments about how antibodies work in

immunofluorescence are true in general, but pertain no more to our work than any other antibody-based tissue profiling study.

Comment 7. The quantification of Lag-3 expression also does not line up with data from existing literature. In Extended Data Fig. 4b (lines 235-238, 337-338), quantification of LAG3 puncta distribution showed that TEX (exhausted T cells) have the lowest level of LAG3 among other subsets. Does the quantification in the plot represents LAG3 expression on CD4, CD8 or all T cells? This was not clearly stated in the text and the figure legend. Second, how does the diverging Lag-3 level between subsets here fit with the literature that Lag-3 has the highest expression on exhausted phenotype (Baitsch 2011, 10.1172/JCI46102, Ruffo 2019, 10.1016/j.smim.2019.101305)? Perhaps performing flow cytometry on similar patient samples could help clarify whether these discrepancies observed between imaging and flow data should be a cause of concern when it comes to interpreting the imaging data revealed by 3D CyCIF.

The reviewer is misrepresenting both our data and the literature: we do not perceive any difference between what we have found by quantifying the number of LAG3 puncta across all T cells and the general understanding that LAG3 is more abundant in activated/exhausted cells. In *Extended Data Fig. 4b* we bin T cells based in part on expression levels of CD103, an integrin whose expression distinguishes tissue resident memory T cells from other T cells. These memory cells are abundant in human skin, as a result of previous encounters with immunogenic antigens, but virtually absent in mice.

While noting that the reviewer is drawing attention to a distinction that we did not discuss in the text, we have examined the data in question again, we found that different ways of binning cells generate different ways of looking at LAG3 levels in

T cell populations. As described in the text, we imposed a very strict definition on a Tex as being PCNA and KI67 (negative) GZMB (negative) TIM3_LAG3 (positive) CD103 (negative) TCF1

Figure 4: Volumetric 3D rendering of metastatic melanoma showing PD-L1 (cyan - top) and composite of PD-L1 and MART1 (cyan and grey respectively - middle). Bottom - composite of PD-L1 (cyan) and MART1 (red) for better visualization PD-L1 in tumor cells. In this region of tissue, PD-L1 appears to be largely punctate.

(negative) CD8 (positive) and PD1 (negative or positive). Other definitions are possible, and common when fewer markers are available. The beauty of these data are that they are all tabular and provided in the supplement, so interested reviewers can impose any definition on exhaustion that they choose.

It is not possible to perform flow cytometry on primary human melanomas, another difference

Figure 5. Image showing typical data used for quantifying LAG3 puncta (turquoise). CD8 – magenta, CD103 – yellow. In our analysis we discriminated between CD103 negative Tex and CD103 positive Tmem cells although not all papers make this distinction, LAG3 puncta are plentiful in both populations.

between work on mice and humans that is not accounted for in the reviewer comments. However, even where flow cytometry is feasible, it would not resolve a question about the number of puncta in a cell as opposed to the total expression level.

Comment 8. *In lines 298-300, the authors' claim that "these data imply that melanocytic cells with features of early malignant transformation are subject to frequent changes in cell state (phenotypic plasticity) rather than progressive evolution from a single transformed or progenitor (stem-like) cell, as proposed for advanced invasive melanoma" should be made with care. Since the "3D images" as shown here are still derived from thin cross-sections of 1-2 cell thickness, the distribution of the diverse states of melanocytic cells across larger volumes (as one would obtain with tissue clearing methods) remains very much limited. Sectioning multiple adjacent 35um slices and reconstructing them in 3D could yield a clearer picture of the actual distribution pattern of those cell states, and quantitative analysis applied to detect non-uniform spatial patterning would be useful.*

The reviewer might not be aware that the melanoma in situ shown here is quite limited in spatial extent. To our knowledge, our images depict the first high resolution high-plex data on a melanoma pre-cancer ever collected. We respectfully disagree with reviewers' assertion that we should adopt a different imaging approach - one favored by the reviewer throughout their comments. Based on our own work with very thick images (see Figure 1 above) and results in *Hor... Germain et al* we think it highly unlikely that the resolution in 3D volumetric samples will be sufficient to discriminate between

adjacent keratinocytes and melanocytes, the primary concern with these data, and that the necessary plex (~30-plex) can be achieved. We have also looked at multiple images of this specimen and many others in 5 micron serial section reconstructions and stand by our conclusions. At the same time, we can all agree that having more data will provide additional insight.

Comment 9. *In lines 313-317, the authors claim that “Within these domains, melanocytic cells had started to pass through the DEJ and were in contact with immune cells (Fig. 5b-c). Thus, our data provide direct evidence for restricted and recurrent spatial niches, defined by the simultaneous presence of an IFN response, melanocyte-immune cell contact, and melanocytes crossing the DEJ (the first step in invasion).” How did the authors determine the DEJ, shown as white dashed lines in Fig. 5b-c? Did the authors stain for keratinocytes (e.g., cytokeratin) to mark the dermal-epidermal boundary? The white arrows in Fig. 5b supposedly show the invasion of melanocytic cells into the dermis, but the arrows point to what appears to be MHC-I+ cells without also showing their tumor antigen (e.g. MART1). Similarly, based on the lower resolution image in Fig. 5a highlighting the close-up region of Fig. 5b, there is no indication of MART1+ cells entering the dermal layer.*

Response. As described in the text, this 55-plex image includes a stain for cytokeratin, which can be used to precisely identify the position of the DEJ. In addition, as described in the text, this feature was also identified by a trained pathologist working on an adjacent H&E image. At our first mention of the DEJ, we will now explicitly referenced that cytokeratin staining & pathologist review were used to identify the DEJ.

As discussed above, it is not possible for us to show all 55 channels in each single image. Instead, we have made a 2D projection of each image available for download-free browsing online using MINERVA software (<https://www.tissue-atlas.org/atlas-datasets/yapp-nirmal-2023/>). MINERVA enables panning and zooming across 11 datasets while turning on and off channels; MINERVA works in-browser and requires no data or software download. We are also releasing a full set of processed and unprocessed data upon acceptance. As we state in *Data Availability*, the images are very large, and impractical for most viewers to download and view on their personal computers. Thus, most viewers will want to use the MINEVA versions.

Comment 10. *What is the basis of defining CD103+ TPEX in Fig. 5d-k? As far as this reviewer understands this issue, there is no evidence from the literature that indicates CD103 as being a marker expressed by TCF-1+ TPEX cells, and as the authors correctly noted in the text, it is typically associated with (but not all) TRM and exhausted T cells. Note that TCF-1 is also a marker for naïve T cells, and at least in mouse, CD103 is known to be expressed among circulating naïve CD8+ T cells (Gavil 2024, 10.1016/j.immuni.2024.06.017). Judging from the images shown in Fig. 5g-j, it appears that the CD103+ TCF1+ cell labeled as TPEX had only minimal level of PD-1 and Tim-3 (Fig. 5g), whereas Lag-3 (Fig. 5h) has been found to be present as intracellularly stored proteins within peripheral naïve CD8+ T cells of a subset of cancer patients, including melanoma patients (Somasundaram 2022, 10.1158/2326-6066.CIR-20-0736). Thus, while this reviewer concurs that immune subset phenotyping in cancer tissues continues to detect increasingly complex and diverse cell states, it is advisable to exercise caution when making such claims without further experimental evidence. Additionally, it is surprising to find that the TEX subset is marked as granzyme B negative and having no GZMB+ cells (0%) in the TEX compartment (Fig. 5k-l). This does not conform with the literature where granzyme B has been identified to be strongly expressed in exhausted T cells in tumor samples (Miller 2019, 10.1038/s41590-019-0312-6). The measured distance of the TEX subset as defined by the authors to tumor cells was also, surprisingly, further than TMEM (Extended Fig. 6i) when exhausted T cells engaging with persistent antigen presentation in the tumor should be in a closer proximity to the tumor cells.*

We also discuss this issue in response to comment 5 above. We also precisely describe how we define different T cells populations in the text, based on careful reading (and referencing) of the biology of human skin and extensive discussion with multiplexed experts in the field of T cell biology. The reviewer references several studies in mice, and one review, but in carefully reading

these papers we detect no discrepancy with our studies. The reviewer also fails to mention that there exists a wide range of ways to subset T cells, depending on the number of markers used, the organism, and the tissue, and that these definitions change from one paper to the next. We very carefully reworded this section of the manuscript in response to the reviewer's previous comments and now make clear that ours is only one way of logically subdividing the complex patterns of marker expression described in our specimens; the reviewer is more than free to come up with other classification schemes and we make no claims of novelty or uniqueness of description.

Importantly, regardless of the precise definition, the key conclusion of this section of the manuscript remains the same: T cell subsets classically thought to comprise different lineages and activity states (cytotoxic, exhausted, memory) extensively overlap in tissue, and they actively engage tumor cells in both memory and exhausted states.

REVIEWER #2:

Remarks to the Author:

I think the authors did a great job to answer all of the major concerns from the initial review in Nature. I appreciate they included a lot more information on the methodic details and shared protocols and I sincerely think that Nature Methods is the right home for the paper.

I do think that even seemingly simple and rather technical figures like Supp. Figures 12-22 are helpful for readers to understand potential limitations and know what to watch out for to perform these kind of experiments. The level of complexity in creating well-vetted panels for 3D multiplexed imaging is very high, yet is often underestimated by many scientists who are just starting with multiplexed antibody stainings or rely on commercial platforms/products without performing important technical controls and optimization. I hope seeing the hurdles documented in the papers provides a more critical thinking perspective to the readers and users of these methods. The last set of supplementary figures are great to evaluate the affect of the imaging settings on the analyses and to get a sense what is adequate and what is overkill for particular questions.

I have just two minor notes:

- 1. The new observation and the associated quantification of the tissue shrinkage/expansion due to the hydration conditions are interesting. I was just curious if the observed change is isotropic or is just happening in the z dimension because the tissue is sticking to the support at the bottom at the time of hydration (or could be happening unevenly from the bottom to the top, where the tissue would expand not into a rectangle prism but to a trapezoid one). The reason I was asking is because I do not think that this is typically taken into account by most when doing registration of H&E stained sections with multiplexed IF or imaging-based spatial transcriptomics assays, but the effect might be minor laterally on thin sections. If the authors have relevant observations or data/quantifications, some comments on this would be useful for that readers that also perform other types of tissue assays.*

Response:

We thank the reviewer for this question. We have evidence of some lateral shrinking/expansion in addition to changes in thickness along the Z dimension, but its magnitude may depend on how well the tissue is adhered to the slide. The orthogonal views shown in Supplementary Figure 18 for a human FFPE melanoma tissue reveal an undulating pattern of tissue-slide interaction, suggesting that the tissue expanded laterally and ultimately peeled off the glass slide. We know that other parts of this tissue sample remained fully stuck to the slide, implying that they were constrained. In other tissue specimens, such as the one we describe in greater detail in the manuscript (Dataset 1 and 2), the tissue remained firmly flat along the slide.

We will revise the text to make these points clearer and to suggest that further analysis of drying and shrinkage effects is necessary as we aspire to build accurate 3D representations of tissue.

Figure 7. A thick-section image in an oblique view and end-on view showing a case in which the tissue has detached from the slide and formed a series of corrugations. This suggests that rehydration causes lateral expansion of tissues but that this expansion is commonly constrained by binding to the rigid slide substrate.

2. *For the alternative mounting with the matrigel, can the authors clarify - is the top coverslip fixed in place somehow (if so how?) and is this then manually displaced to do the washes and put back on each imaging cycle?*

Thank you for the question. The top coverslip is placed loosely on the chamber so as to reduce evaporation or prevent dust from entering the sample. It is lifted off and replaced between wash/bleach/stain cycles. We opted to use a glass coverslip so that visual inspection of the sample by eye or transmitted light was possible. Since the coverslip is not for imaging, any grade (0, 1.5, 2) is fine. We will make these points clear in the revised manuscript.

Remarks on code availability:

I have browsed through the deposited code, but was not able to do an extensive review to see how it runs as I do not have MATLAB.

As far as I can see the readme only covers the cell interaction analysis and demo. A bit more documentation/instructions on the other parts could be helpful (some useful information is fortunately provided in the Methods section).

Response:

Thank you for these suggestions. We will make sure the repo is more self-explanatory and that we also direct readers to the Danuser repo (<https://github.com/DanuserLab/u-Segment3D>) for the segmentation code.